



# Heterogeneous glacier thinning patterns over the last 40
# years in Langtang Himal
**S. Ragettli[1,2], T. Bolch[2,3] and F. Pellicciotti[1,4]**
[1]{Institute of Environmental Engineering, ETH Zürich, Switzerland}
[2]{University of Zurich, Department of Geography, Zurich, Switzerland}
[3]{Institute for Cartography ,Technische Universität Dresden, Dresden, Germany}
[4]{Northumbria University, Department of Geography, Newcastle upon Tyne, UK}
Correspondence to: S. Ragettli (ragettli@ifu.baug.ethz.ch)
**Abstract**
Himalayan glaciers are losing mass at rates similar to glaciers elsewhere, but heavily debris-
covered glaciers are receding less than debris-free glaciers or even have stable fronts. There is
a need for multi-temporal mass balance data to determine if glacier wastage of debris-covered
glaciers is accelerating. Here, we present glacier volume and mass changes of seven glaciers
(5 partially debris-covered, 2 debris-free) in the upper Langtang catchment in Nepal of 28
different periods between 1974 and 2015 based on 8 digital elevation models (DEMs) derived
from high-resolution stereo satellite imagery. We show that glacier volume decreased during
all periods between 2006 and 2015 (2006 - 2015: $-0.60 \pm 0.34$ m $a^{-1}$) and at higher rates than
between 1974 and 2006 ($-0.28 \pm 0.42$ m $a^{-1}$). However, the behavior of glaciers in the study
area was highly heterogeneous, and the presence of debris itself does not seem to be a good
predictor for mass balance trends. Debris-covered tongues have highly non-linear thinning
profiles, and we show that local accelerations in thinning correlate with complex thinning
patterns characteristic of areas with a high concentration of supraglacial cliffs and lakes. At
stagnating glacier area near the glacier front, on the other hand, thinning rates may even
decrease over time. We conclude that trends of glacier mass loss rates in this part of the
Himalaya cannot be generalized, neither for debris-covered nor for debris-free glaciers.



## 1 Introduction

Global warming has caused widespread recent glacier thinning and retreat in the Himalayan region (Bolch et al., 2012). The impact of current and future glacier changes on Himalayan hydrology and downstream water supply strongly depends on the rate of such changes. However, planimetric and volumetric glacier response rates are not easy to characterize due to limited data availability, and many recent studies have highlighted the spatially heterogeneous distribution of glacier wastage in the Himalayas (Fujita and Nuimura, 2011; Bolch et al., 2012; Kääb et al., 2012). Prominent examples of current-day regional differences in glacier evolution across the Hindu Kush–Karakoram–Himalaya (HKH) are the reported positive glacier mass balances in the Pamir and Karakoram (Kääb et al., 2012; Gardelle et al., 2013). Across regional scales systematic differences in recent glacier evolution can often be associated to differences in climatic regimes (Fujita, 2008), particularly to the varying influence of the south Asian monsoon and westerly disturbances (Yao et al., 2012). However, also within the same climatic region the rate of glacier changes can be highly heterogeneous (Scherler et al., 2011b). As such, the main focus of current scientific debate concerns the differences in glacier response to climate in function of varying surface characteristics caused by supraglacial debris. Thick debris cover is a common feature in the HKH (Scherler et al., 2011b; Racoviteanu et al., 2015) and effectively reduces melt rates of underlying ice (e.g. Östrem, 1959; Mattson et al., 1993). Debris layers usually thicken in downstream direction due to the emergence of englacial debris through melting out of ice and rockfalls on the valley sides (Zhang et al., 2011; Banerjee and Shankar, 2013). Decreasing melt rates with increasing debris thickness counteract the effect on melt of air temperature increase at lower elevations. Consequently, debris-covered glaciers often form low reaching tongues with low surface velocities and low melt rates near the terminus (Benn et al., 2012). It is known that such quasi-stagnating tongues often have nearly stationary fronts also if the overall glacier mass balance is negative (Banerjee and Shankar, 2013). Glacier area loss is therefore not a good indicator to characterize the response of debris-covered glaciers to a warming climate (Scherler et al., 2011b). The characterization of debris-covered glacier response to climate is further complicated by the frequent occurrence of ice cliffs and supraglacial lakes. At exposed cliffs, melt rates are much higher compared to the ice covered by a thick debris mantle (Sakai et al., 1998, 2002; Immerzeel et al., 2014; Steiner et al., 2015; Buri et al., 2016), and also at supraglacial ponds energy absorption is several times larger than that at the surrounding debris-covered surface (Sakai et al., 2000; Miles et al., 2016a). The water in the ponds warms





and may cause large englacial voids created by the drainage of warm water from the ponds,
which may collapse and generate new depressions or ice cliffs (Benn et al., 2012). Debris-
covered glaciers are also often avalanche nourished (Scherler et al., 2011a), which means that
downslope conditions may be influenced more quickly by changes in high-altitude
precipitation (Hewitt, 2005). To document the response of debris-covered glaciers to a
warming climate, estimations of glacier-scale mass changes are therefore required (Cogley,
2012). Recent large scale geodetic studies based on remote sensing have provided evidence
that the present-day lowering rates of debris-covered areas in the HKH might be similar to
those of debris-free areas even within the same altitudinal range (Kääb et al., 2012; Nuimura
et al., 2012; Gardelle et al., 2013), and surmise this could be due to enhanced melt from
exposed ice cliffs and near supraglacial lakes. Several detailed modelling studies on the other
hand have demonstrated the melt reducing effect of debris on the glacier scale (e.g. Juen et al.,
2014; Ragettli et al., 2015), and have shown how supraglacial debris prolongs the response of
the glacier to warming (Rowan et al., 2015). Discrepancies between the different conclusions
may be associated to glacier samples that are not comparable, to the discrepancy between
thinning and melt due to glacier emergence velocity and to model uncertainties particularly
regarding the representation of the effect on melt caused by supraglacial cliffs and lakes.
Programs to monitor debris-covered glaciers by direct observations have been recently
initiated in the Karakorum (e.g. Mayer et al., 2006; Mihalcea et al., 2006, 2008) or in the
Central Himalaya (e.g. Pratap et al., 2015; Ragettli et al., 2015) which will increase the
knowledge on debris-covered glacier processes and improve their representations in models.
However, due to the logistical and financial constraints, long term mass balance measurement
programs are basically inexistent in the HKH. To document changes in debris-covered glacier
thinning rates over time, declassified high resolution reconnaissance satellite data available
from the 1960s and 1970s are an important source of information. However, only few studies
used these data and employed multi-temporal digital elevation models (DEMs) extracted from
stereo satellite imagery to study changes in thinning rates of Himalayan glaciers over time. In
the Khumbu region in the Nepalese Himalaya, Bolch et al. (2008, 2011) have calculated
multi-decadal mass loss of glaciers since 1962. They found that volume loss has possibly
increased in recent years (e.g. volume loss rates of Khumbu glacier 1970-2007: -0.30 $\pm$
0.09 m $a^{-1}$, 2002-2007: -0.50 $\pm$ 0.52 m $a^{-1}$). Similar conclusions were drawn from a study by
Nuimura et al. (2012) who calculated accelerated thinning rates in the same study region





comparing the two periods 1992-2008 (e.g. Khumbu glacier: -0.35 ± 0.20 m a$^{-1}$) and 2000-
2008 (-0.76 ± 0.52 m a$^{-1}$).
The aim of this study is to calculate multi-decadal surface elevation changes of selected
glaciers in the upper Langtang catchment in Nepal, for different periods between November
1974 and October 2015. Eight different high resolution DEMs, all extracted from stereo or tri-
stereo satellite imagery, allow gaining more detailed insights in spatial and temporal changes
in glacier thinning patterns than any geodetic study before in this part of the Himalaya. For
each of the seven glaciers in the sample (five debris-covered and two debris-free glaciers), the
analysis is constrained to those DEM combinations which are least affected by uncertainties
stemming from errors in the DEM, ambiguous outlier definitions, the filling of missing data
or DEM adjustment errors. The resulting 30 m resolution dataset of thinning rates allows
addressing three main research questions. First, we assess if overall thinning of glaciers in the
region has accelerated in recent years. Second, we determine if spatial thinning patterns have
changed over time. To explain changes in thinning rates we derive a number of glacier surface
properties and glacier surface velocities. Third, we assess if there are major differences
between the response of debris-covered and debris-free glaciers in the sample. Finally, we
also look at the immediate cryospheric impact of the April 2015 earthquake that devastated
large parts of the Langtang catchment by triggering large avalanches. By comparing pre-
earthquake DEMs from April 2014 and February 2015 with post-earthquake DEMs from May
and October 2015 we can quantify the impact of singular avalanche events on the mass
balance of debris-covered glacier tongues.
**2   Study Site**
We analyze the seven largest glaciers in the Langtang valley (Langtang, Langshisha,
Shalbachum, Lirung, Ghanna, Yala, Kimoshung), located in the monsoon-dominated Central
Himalaya in Nepal, approximately 50 km north of Kathmandu and 100 km west of the
Everest region. While Yala and Kimoshung glaciers are debris-free glaciers, all other studied
glaciers have tongues that are entirely covered by supraglacial debris (Figure 1). Langtang
Glacier the largest glacier in the valley with an area of 46.5 km$^2$ in 2006 (Table 1) and a total
length of approximately 18 km. The smallest glacier in our sample is Ghanna Glacier with an
area of 1.4 km$^2$.
Critical debris thicknesses leading to a reduction of melt rates are exceeded often throughout
the entire ablation areas of debris-covered glaciers the upper Langtang catchment (Ragettli et





al., 2015). Relatively shallow layers of debris appear only at the transition zone between accumulation and ablation area. However, at Lirung, Shalbachum, Ghanna and Langshisha Glaciers the upper margins of debris-covered sections are located at the foot of steep cirques or icefalls, and transition zones are therefore very short. Only at Langtang Glacier englacial debris emerges gradually just below the equilibrium line altitude, which in this part of the Himalaya is located between 5400 and 5500 m asl (Sugiyama et al., 2013; Ragettli et al., 2015). In addition to spatially variable debris thicknesses due to gradual emergence of debris and rockfalls on the valley sides, ice cliffs and supraglacial ponds further increase the heterogeneity of glacier surface characteristics in the Langtang valley (Pellicciotti et al., 2015).

The ablation season of glaciers in the Langtang valley lasts from pre-monsoon (April – mid June) to post-monsoon (October – November), whereas the ablation season generally lasts longer at the low-lying debris-covered tongues. The monsoon season (mid June – September) is at the same time the warmest and the wettest period of the year. Snow cover at the lower elevation ranges of debris-covered glaciers is common only in winter (December – March). However, outside the monsoon period precipitation is limited and winters are rather dry (Collier and Immerzeel, 2015).

## 3 Data and methods

### 3.1 Satellite imagery and DEM generation

Multitemporal high-resolution data from different sensors are applied to assess glacier change in the upper Langtang catchment. Each type of remote sensing data employed to calculate glacier elevation changes is listed below. Spatial and radiometric resolutions and base to height (b/h) ratios are provided in Table 2.

- The oldest data originate from Hexagon KH-9 stereo satellite images from November 1974 (Surazakov and Aizen, 2010; Pieczonka et al., 2013; Maurer and Rupper, 2015). These are declassified images from an U.S. reconnaissance satellite program. The Hexagon DEM used here was generated for the study by Pellicciotti et al. (2015). We therefore refer to this study for further technical details regarding the Hexagon DEM.

- Cartosat-1 is a remote sensing satellite built by the Indian Space Research Organisation (Tiwari et al., 2008). We purchased radiometrically corrected along-track stereo imagery (processed at level 'ortho-kit') of the upper Langtang catchment from



October 2006 and November 2009. Cartosat-1 data have been previously used for
DEM generation e.g. in the Khumbu region in the Nepal Himalaya by Bolch et al.
(2011) and by Pieczonka et al. (2011).
• ALOS-PRISM (Advanced Land Observing Satellite - Panchromatic Remote-Sensing
Instrument for Stereo Mapping) was an optical sensor mounted on a Japanese satellite
system which operated from January 2006 to April 2011 (Bignone and Umakawa,
2008; Tadono and Shimada, 2009; Lamsal et al., 2011; Holzer et al., 2015). We
purchased a radiometrically calibrated along-track triplet mode scene from December

9     2010.

• SPOT6/7 (Système pour l'Oberservation de la Terre) along-track tri-stereo images are
available for April 2014, May 2015 and October 2015. SPOT6 and 7 are the newest
satellites of the SPOT series which have been frequently used for geodetic glacier
mass balance studies (e.g. Berthier et al., 2007, 2014; Pieczonka et al., 2013). We
acquired stereoscopic images in panchromatic mode corrected for radiometric and
sensor distortions. Two of the three SPOT6/7 scenes used in this study were acquired
in April/May which means that limited amounts of winter snow still is present on the
images. However, the imagery has a high radiometric depth of 12bit (Table 2) which
leads to good correlation results also over snowy parts.
• The WorldView DEMs used in this study are 8m posting Digital Elevation Models
(DEMs) produced using the Surface Extraction with TIN-based Search-space
Minimization (SETSM) by Noh and Howat (2015) and were downloaded from
http://www.pgc.umn.edu/elevation. The DEMs are constructed from overlapping pairs
of high-resolution images acquired by the WorldView-2 and 3 satellites in February

24     2014.

The WorldView DEMs rely on the satellite positioning model to locate the surface in space,
while all other DEMs used in this study have been extracted with ground control. The basis
for the georectification were six differential GPS (dGPS) points collected on Lirung Glacier
23 October 2014 (Brun et al., 2016). Because glacier motion and ablation have to be
accounted for when using on-glacier dGPS points, we first generated a DEM from an across-
track Pléiades stereo image pair from 1 and 9 November 2014 using the available dGPS
points as ground-control points (GCPs). Glacier changes between 23 October and the
acquisition dates of the Pléiades scenes are negligible due to the low temperatures during this



period. Subsequently, we determined 17 GCPs on the basis of the Pléiades scene which were
then used to derive a DEM from the SPOT6 April 2014 tri-stereo scene. The Pléiades DEM
itself in the following is not used to calculate glacier elevation changes since only low
correlation scores could be achieved for the upper parts of glaciers because of snowfall onset
between 1 and 9 November 2014. To guarantee high quality GCPs, only pixels with
correlation scores higher than 0.7 were considered for GCPs. Since the Pléiades scene covers
only about one fourth of the upper Langtang catchment, an additional 60 GCPs were
determined on the basis of the April 2014 SPOT6 scene for the DEM extraction from the
Cartosat-1, ALOS Prism and SPOT7 scenes. All SPOT6/7, Cartosat-1 and ALOS Prism
DEMs used for this study were generated using the OrthoEngine module of PCI Geomatica
2015 and approximately 100 tie points for each scene. We were using the same parameters for
DEM generation as proposed by Berthier et al. (2014) except setting the parameter 'DEM
detail' to 'very high' instead of 'low', which provided better results for the rugged debris-
covered glacier surfaces.
In addition to the DEMs listed above, the 2000 SRTM (Shuttle Radar Topography Mission)
1 Arc-Second Global DEM (30m spatial resolution) was used to calculate slopes and
accumulation area ratios (AARs) of glaciers (Table 1). However, for DEM differencing the
SRTM DEM was not used because of the uncertainty regarding the penetration depth of the
radar signal into snow and ice (Gardelle et al., 2013; Kääb et al., 2015; Pellicciotti et al.,
2015). Only DEMs extracted from optical stereo imagery are therefore employed to calculate
elevation changes in this study.

## 3.2  Co-registration and DEM differencing

To measure the glacier elevation changes we use all possible DEM pairs. The number of
possible two-fold combinations of $n$ DEMs ($N_{\Delta t}$) is
$$N_{\Delta t} = \sum_{k=1}^{n-1} k \, . \tag{1}$$
Since the two available WorldView scenes were acquired only 20 days apart and are adjacent
to each other (the Worldview-2 DEM covers the western part of the study catchment and the
WorldView-3 DEM the eastern part), for each part of the catchment eight DEMs are available
and elevation differences over $N_{\Delta t} = 28$ different time periods can therefore be calculated. Co-
registration of each DEM-pair is applied in order to minimize the errors associated with shifts.



Systematic errors due to tectonic uplift which could be relevant after the April 2015 Nepal
earthquake are also corrected with the co-registration. For this purpose we exclude from each
DEM the non-stable terrain such as glaciers and in general all off-glacier area at elevations
higher than 5400 m asl (which is the estimated height of the equilibrium line altitude (Ragettli
et al., 2015)). The correlation score maps from the DEM extraction are used to exclude all
DEM grid cells with a correlation score below 0.5. Then, horizontal shifts are determined by
minimizing the aspect-dependent bias of elevation differences (cf. Nuth and Kääb, 2011)
between each DEM pair. All terrain below a slope of 10° is excluded to allow for the slope
dependency of the method. The slave DEM (always the 'older' DEM with earlier acquisition
date) is then resampled (bilinear interpolation) according to the determined horizontal shift. In
a second step the vertical DEM shifts and possible tilts are corrected using second order trend
surfaces fitted to all gently inclined (≤15°) stable terrain (Bolch et al., 2008; Pieczonka et al.,
2011; Pieczonka and Bolch, 2015).
We resample all DEMs bilinearly to the grid size of the coarsest DEM (30 m) in order to
reduce the effect of different resolutions. Elevation differences are calculated by subtracting
the older from the younger DEM (such that glacier thickening values are positive) and are
converted to elevation change rates by dividing by the number of ablation seasons between
the acquisition dates. Area-average glacier elevation change rates are calculated using always
the maximum glacier extent between two acquisition dates.

## 20   3.3  Delineation of glaciers and debris-covered areas, proxies for supraglacial
## 21       cliffs/lakes

The glacier outlines were manually delineated. We used the orthorectified satellite images
with the least snow cover (the Cartosat-1 2006 and 2009 scenes) to delineate the accumulation
areas, and assumed no changes in the accumulation area over time. The tongues of the seven
studied glaciers and debris extents were re-delineated for every year for which satellite images
are available (1974, 2006, 2009, 2010, 2014 and 2015), using the corresponding orthorectified
satellite images. A first operator delineated the outlines and a second operator provided
feedback in order to improve delineation accuracy. The four largest glaciers in the valley were
already delineated manually by Pellicciotti et al. (2015) for the years 1974 and 2000.
However, we decided not to use those outlines because of the considerably higher resolution
of the images that are available for this study and for consistency in the procedure applied for
different outlines. We also re-delineated the catchment boundaries using the SRTM 30 m



DEM and an automated flow accumulation process to accurately define the upper limit of the
glacier accumulation areas. As a result, the calculated glacier areas (Table 1) changed
considerably with respect to Pellicciotti et al. (2015). The 1974 glacier area of Langshisha
Glacier changed by -40.4%, mostly due to clipping with the catchment mask which reduced
the extent of the accumulation areas. The 1974 areas of Langtang, Shalbachum and Lirung
changed by -8.7%, -9.5% and +8.0%, respectively.
For the co-registration of the DEMs and for stable terrain accuracy assessments we
furthermore use the GAMDAM glacier inventory (Nuimura et al., 2015) to mask out also the
smaller glaciers from off-glacier data.
Since supraglacial cliffs are difficult to identify on the orthorectified satellite images, we use
two statistical proxies to characterize debris-covered glacier sections. The first proxy is the
standard deviation of $\Delta h/\Delta t$ values ($\sigma \Delta h/\Delta t$). Second, we consider the difference between the
50% and 10% quantile of $\Delta h/\Delta t$ values per elevation band ($\Delta h/\Delta t_{Q50-Q10}$). Both proxies
identify rugged, heterogeneous surfaces with a strong spatial variability in melt rates,
characteristic of unstable debris layers where supraglacial cliffs and lakes appear (e.g. Sakai et
al., 2000, 2002; Immerzeel et al., 2014; Buri et al., 2016). The proxies are calculated per 50 m
elevation band of each debris-covered tongue (excluding tributary branches and large
avalanche cones). The two uppermost elevation bands of each glacier are not considered due
to intermittent, irregular debris cover at the transition between debris-free and debris-covered
ice. Because glacier movement and changes in surface topography over time smooth out
heterogeneous thinning patterns, short time intervals are required to identify cliff and lake
areas on surface elevation change maps. $\Delta h/\Delta t_{2006-09}$ (Figure 2a) is thus used to calculate the
cliff and lake proxies, because this map represents elevation changes over a relatively short
time interval and is not particularly affected by outliers (Table S1).
**3.4  Outlier definitions**
The relatively large dataset of 28 elevation change rate ($\Delta h/\Delta t$) maps allows for a rigorous
definition of outliers in order to restrain the subsequent analysis only to those $\Delta h/\Delta t$ signals
which are least affected by various sources of uncertainty. For this purpose we defined several
criteria to filter out outliers at different scales (Table 3).



### 3.4.1  Outlier detection at the grid scale

Pixel values identified as outliers according to the criteria defined below are removed from Δh/Δt maps and filled using inverse distance weighting (IDW), considering the remaining glacier grid cells.

Correlation scores

PCI Geomatica provides the stereo matching score for each extracted DEM pixel. A threshold of 0.5 is applied to exclude elevations of poor accuracy. The correlation score of 0.5 is the lower bound of matching scores denominated as 'fair' in Pieczonka et al. (2011). Note that this criterion cannot be applied to the Hexagon and WorldView DEMs, since the correlation scores for these DEMs are not available.

Δh outliers

Pixels of debris-free glacier area are defined as outliers when the elevation differences differ by more than two standard deviations (2σ) from the mean elevation difference of all debris-free glacier area. The same criterion is applied to debris-covered glacier area but with a threshold of three standard deviations (3σ). The higher threshold applied to debris-covered terrain is justified by two reasons. First, the DEM accuracy is generally higher for debris-covered terrain (Figure S1b) due to more shallow slopes and high contrast. Second, the spatial variability in thinning rates can be very high over debris, due to heterogeneous surface characteristics such as variable debris thickness or supraglacial cliffs (Immerzeel et al., 2014). By applying a higher Δh outlier threshold for supraglacial debris we reduce the risk of misclassifying areas of high local thinning as outliers. The most extreme outliers (exceeding ±150 m) are not considered to calculate the standard deviations, in order to guarantee a better comparability with Δh maps where such artefacts occur more often (especially the Δh maps involving Hexagon or WorldView DEMs, where outlier correction based on matching scores is not possible).

### 3.4.2  Outlier detection at the glacier scale

Outliers at the glacier scale concern systematic errors which affect area-averaged Δh/Δt values per glacier. The glacier scale outliers are removed from the dataset, which means that from Δh/Δt maps all pixel values of concerned glaciers are removed, while the data from other glaciers can still be used.





Data availability
If for individual glaciers and $\Delta h/\Delta t$ maps less than 50% of all pixel values remain after outlier
removal at the grid scale, these are not considered for subsequent analysis.
Outlier correction uncertainty
Since no unambiguous criterion exists to identify outliers, but results might be sensitive to
their definition, we apply different $\sigma$ thresholds to the $\Delta h/\Delta t$ maps ($\pm 1\sigma$). Each time all data
gaps are filled using IDW and mean $\Delta h/\Delta t$ values per glacier are calculated. Then we compare
the resulting mean $\Delta h/\Delta t$ values per glacier corresponding to a more strict outlier definition
($-1\sigma$) with the mean $\Delta h/\Delta t$ values corresponding to a more tolerant outlier definition ($+1\sigma$). If
the absolute difference between the two values exceeds the mean thinning rates of the total
glacier area calculated between October 2006 and October 2015, it is assumed that the signal
to noise ratio is below a critical level. Therefore all $\Delta h/\Delta t$ pixel values corresponding to a
glacier where this criterion is exceeded are not considered subsequently, since the uncertainty
which is due to outlier correction cannot be constrained sufficiently. This outlier criterion
generally causes the exclusion of glaciers from the analysis which are affected by many
outliers, because of factors that increase the DEM uncertainty such as steep slopes or low
image contrast due to snow or shadows.
DEM adjustment uncertainty
Co-registration of DEMs is important because a small horizontal offset between two DEMs
can produce a large elevation error where the topographic slope is steep (Berthier et al.,
2004). However, co-registration procedures rely on curve and surface fitting functions which
may be sensitive to outliers or which themselves might not describe vertical or horizontal
shifts accurately due to tilts or distortions in the DEM. Assuming three acquisition dates $t_1$, $t_2$
and $t_3$, the elevation differences calculated between $t_1$ and $t_3$ ($\Delta h_{t1,t3}$) should be equal to
$\Delta h_{t1,t2} + \Delta h_{t2,t3}$ if the DEMs are adjusted perfectly. However, this is rarely the case, and
triangulation residuals therefore provide an estimate of the co-registration accuracy (Nuth
and Kääb, 2011; Paul et al., 2015). Having $n$ DEMs available, differential DEMs can be
derived $C_{k-1}$ times by employing $k$ DEMs (and therefore $k-1$ DEM differencing steps):
$$C_{k-1} = \frac{\binom{n}{k}}{2} N_{\Delta t}^{-1} \tag{2}$$





$N_{\Delta t}$ is the number of possible two-fold combinations of $n$ DEMs (equation 1). According to
equation (2), having $n=8$ DEMs available, the difference between two DEMs can be
determined six times by adding or subtracting the $\Delta h$ values using a third DEM ($k=3$). Using
$k$ equal to 2, $C_l$ is equal to 1 according to equation (2). The DEM adjustment uncertainty
($U_{adj}$) is then calculated as follows:
$$U_{adj} = (abs(\Delta h_{C1} - median(\Delta h_{C2}))) / \Delta t , \tag{3}$$
where $\Delta h_{C1}$ and $\Delta h_{C2}$ are the elevation differences calculated $C_{k-1}$ times for a given period $\Delta t$.
To quantify the DEM adjustment uncertainty of area-average $\Delta h$ values per glacier, we
therefore first calculate mean glacier elevation differences corresponding to each of the 28 $\Delta h$
maps, considering only the common minimum glacier extent. Then we combine each of the
values six times as described above and calculate the DEM adjustment uncertainty according
to equation (3). As a threshold for the acceptable DEM adjustment uncertainty we use again
the mean thinning rates calculated between October 2006 and October 2015 of the total
glacier area.
Note that gap filled $\Delta h$ maps are required for the calculation of the DEM adjustment
uncertainty. The gap filling itself causes uncertainty which increases $U_{adj}$. Therefore, the
DEM adjustment uncertainty as defined here is used to estimate the uncertainty which stems
both from the co-registration procedure and from the gap filling procedure.
### 3.4.3 Outlier detection at the catchment scale
Catchment scale outliers are $\Delta h/\Delta t$ maps which are affected by systematic errors that lead to
significant off-glacier elevation differences. Those $\Delta h/\Delta t$ maps are removed entirely from the
dataset, which means that the final analysis may be conducted using less than 28 $\Delta h/\Delta t$ maps.
At the catchment scale we define outliers by the off-glacier mean elevation difference
($MED_{noglac}$) and standard deviation ($\sigma_{noglac}$). $MED_{noglac}$ and $\sigma_{noglac}$ are calculated excluding the
steepest slopes where glaciers are unlikely to occur. This threshold slope is defined as the 95[th]
percentile of the slope of all glacier grid cells ($Q95\ s_g$) and is equal to 45°.
To identify off-glacier elevation difference outliers we also use a map of monsoon snow-
cover frequency (Figure 1) which is based on Landsat 1999 to 2013 land cover classifications
(Miles et al., 2016b). Since the monsoon period is the warmest period of the year we assume
that a monsoon snow-cover frequency higher than 20% represents terrain which is frequently



snow covered. For a second estimate of the off-glacier mean elevation difference (MED2$_{noglac}$)
and standard deviation (σ2$_{noglac}$) we therefore mask out also those areas since the surface
elevation of snow covered terrain might change over time.
Finally a third estimate is provided by masking out also intermediate slopes which can occur
on glaciers but which do not appear on debris-covered glacier area. The threshold slope is
defined as the 95$^{th}$ percentile of the slope of all debris-covered glacier grid cells (Q95 s$_d$) and
is equal to 18°. Since the DEM uncertainties generally increase with steeper slopes (Nuimura
et al., 2012) and lower image contrast such as over snow, it can be assumed that this third
estimate leads to the lowest mean elevation difference (MED3$_{noglac}$) and standard deviation
(σ3$_{noglac}$).
In order to effectively minimize the uncertainty in the ensemble, outliers are defined if they
are larger than $q_3 + 1.5*(q_3 - q_1)$ or smaller than $q_1 - 1.5*(q_3 - q_1)$, where $q_1$ and $q_3$ are the
25th and 75th percentiles, respectively, of all MED$_{noglac}$ and σ$_{noglac}$ values in the ensemble.
Note that off-glacier outliers at the pixel scale are removed prior to the calculation of mean
elevation differences and standard deviations analogous to the outlier correction for
glacierized areas (Section 3.4.1). For off-glacier area with a monsoon snow frequency > 20%
we use the same 2σ threshold as for debris-free glacier area and for off-glacier area with a
monsoon snow frequency ≤ 20% we use a 3σ threshold as for debris-covered glacier area.
**3.5   Uncertainty quantification**
The uncertainty of the elevation change rates of the glacierised areas is quantified based on
the individual stable terrain elevation differences. Since it is known that the distribution of
uncertainty strongly depends on terrain characteristics such as slope, deep shadows, and
snowfields with low contrast or the non-uniform distribution of the GCPs in altitudes
(Berthier et al., 2004), we first quantify the uncertainties separately for each 50 m elevation
band. It can be expected that the mentioned sources of uncertainty become more abundant at
higher altitudes (Nuimura et al., 2012). The SRTM 30 m DEM is used as a basis to delineate
50 m elevation bands. Both the standard error of the mean (SE) and the mean elevation
difference (MED) are considered for the uncertainty estimates. The standard error quantifies
the effect of random errors on uncertainty according to the standard principles of error
propagation:





$SE = \dfrac{\sigma_{\Delta h, noglac}}{\sqrt{n}}$ ,            (4)
where $\sigma_{\Delta h, noglac}$ is the standard deviation of the mean elevation change of non-glacierized
terrain per elevation band, and n is the number of pixels per elevation band. To calculate the
uncertainty (*unc*) per elevation band SE and MED are summed quadratically:
$unc = \sqrt{SE^2 + MED^2}$ ,            (5)
Usually, *n* in equation 4 is reduced to the number of independent pixels within the averaging
area (e.g Bolch et al., 2011; Gardelle et al., 2013) to take into account the spatial
autocorrelation in the dataset. However, we conservatively assume 100% dependence of the
uncertainty estimates for each elevation band (i.e. assuming no error compensation across
elevation bands). Furthermore, to account for spatially non-uniform distribution of
uncertainty, uncertainties per elevation band are weighted by the altitudinal distribution of a
given area when calculating area-average $\Delta h/\Delta t$ uncertainties. $\Delta h/\Delta t$ uncertainty estimates for
each individual glacier therefore differ depending on glacier hypsometry.
For overall mass budget uncertainties we assume an ice density of 850 kg/m$^3$ to convert the
volume change into mass balance (Huss, 2013) and consider at once the elevation change rate
uncertainties and an ice density uncertainty of 60 kg/m$^3$.
**3.6   Surface velocities**
To assist with the interpretation of the volumetric changes, we use glacier velocities
determined with the COSI-Corr cross-correlation feature-tracking algorithm (Leprince et al.,
2007) and the available satellite imagery. Since the cross-correlations can be best determined
if the period between the acquisition dates of images is short, we use the orthorectified
Cartosat-1 Nov 2009 and ALOS-PRISM Dec 2010 images for this purpose. Other image pairs
were not considered due to longer periods between acquisitions or the presence of snow
patches at lower elevations (SPOT6 April 2014, SPOT7 May 2015). The selected
orthorectified images (5 m resolution) are adjusted according to the shifts determined by co-
registration (Section 3.2). Since the window size must be large enough to avoid correlating
only noise but small enough to not degrade the output resolution (Dehecq et al., 2015), we
tested several configurations. The best results for the COSI-Corr multiscale correlation
analysis were achieved using a window size of 128 down to 32 pixels, as also proposed by





Scherler et al. (2008). To post-process the velocity data we removed pixels with x- or y-
velocity values greater than 40 m/a, since these were identified as errors by manually
measuring the surface displacement on the basis of the orthorectified images and prominent
features. We then ran a median filter on the data to remove areas which show a local reversal
in x or y directions. Missing values are then filled with the mean of the adjacent 8 values.
Finally, the velocity map is resampled to 30 m resolution with a bicubic algorithm. To
discriminate moving ice from quasi-stagnant ice we use a threshold of 2.5 m a$^{-1}$ following
Scherler et al. (2011b).
**4    Outliers and uncertainty assessment**
The definition of outliers lead to a considerable reduction of the whole dataset from 196
glacier Δh/Δt maps (7 glaciers x 28 DEM difference maps) to 104 maps. 92 maps (46.9%,
Table 4) were therefore removed from the dataset because they did not fulfil one or more
outlier criteria at the glacier or catchment scale (Table S1).
Glacier Δh/Δt maps which involve the ALOS-PRISM Dec 2010 DEM for the DEM
differencing were most often rejected (85.7% rejected, Table 4), followed by maps involving
the WorldView Feb 2015 DEM (63.3%) and the SPOT6 April 2014 DEMs (59.2%). The
ALOS-PRISM sensor has a radiometric resolution of 8-bit, which means that in comparison
to a 12-bit image (SPOT6/7, Table 2), $2^4$=16 times less information is provided per
panchromatic image pixel. The image contrast over snow and also over debris-free glacier
area is therefore lower which leads to the more frequent occurrence of outliers. The high
rejection rate for the WorldView DEM can be explained by the fact that this composite DEM
was generated with an automatic algorithm using only the sensor RPCs and no ground control
(Noh and Howat, 2015), but also due to an abundance of snow at lower elevations and low
contrasts in February 2015. The presence of continuous snow surfaces down to ~5000 m asl
in April 2014 also lowered the matching scores of the SPOT6 April DEM. However, the
relatively high number of rejected maps involving the SPOT6 April DEM is mostly due to the
incomplete representation of Langtang Glacier on the SPOT6 scene, which does not cover the
area north of 28°19'N (Figure 1).
The glacier-wise outlier evaluation led to an uneven distribution of rejection rates per glacier.
37 of the rejected maps (40.2% of all rejected maps) concern Kimoshung and Lirung Glaciers.
Lirung Glacier is the steepest of all glaciers (Table 1) which leads to low matching scores
(Figure S1a) and missing data (Figure 5b) due to deep shading and to higher DEM adjustment





uncertainties (Figure S2a). Kimoshung Glacier has a very steep tongue and the accumulation
area is located on a high plateau above 5400 m asl, representing 86% of its area (Table 1),
which is frequently covered by a continuous snow layer. These topographic characteristics
lead to significantly lower matching scores than for other glaciers (Figure S1a), more outliers
and therefore higher outlier correction uncertainties (Figure S3a).
We also applied all outlier criteria (Table 3) separately to the $\Delta h/\Delta t$ maps of debris-covered
tongues. Considering only shallow slopes below 18°, representative of the slopes of debris-
covered tongues, the off-glacier standard deviation decreases ($\sigma 3_{noglac}$, Figure 3b) and we
identify significantly less outliers in mean off-glacier elevation differences than if also steeper
slopes are considered (MED3$_{noglac}$, Figure 3a). The matching scores for debris-covered area
are high (Figure S1b) which leads to only few data gaps (Figure 5b) and very low outlier
correction uncertainties (Figure S3a). Overall, only 17.1% of all $\Delta h/\Delta t$ maps of debris-
covered tongues are removed from the dataset after outlier cleaning (Table 4).
It is important to note that most of the rejected $\Delta h/\Delta t$ maps of glaciers and of debris-covered
tongues correspond to short time intervals between DEM-pairs. The median period length of
all rejected $\Delta h/\Delta t$ maps is three years for glacier $\Delta h/\Delta t$ maps and one year for debris-covered
tongue $\Delta h/\Delta t$ maps (Table 4). Outliers are more likely to occur when the intervals are short,
since errors in the DEMs in this case lead to lower signal to noise ratio. This also explains
why only very few outliers concern DEM pairs involving the Hexagon 1974 DEM (Table 4),
in spite of the lower spatial and radiometric resolution of the Hexagon KH-9 imagery (Table

21 2).

The removal of $\Delta h/\Delta t$ maps identified as outliers on the basis of the off-glacier mean
elevation difference and standard deviation led to a reduction of the ensemble uncertainty
range throughout all elevation bands (Figure 4). At the lower elevations below 5300 m asl,
where the debris-covered areas are located, the remaining uncertainties are very low with
magnitudes of a few centimeters per year. However, considerable uncertainties at high
altitudes remain. We therefore calculate higher uncertainties for glaciers with large areas at
high altitudes after weighting the uncertainties associated to each elevation band with the
hypsometry of each glacier. This explains the overall higher uncertainty estimates for Lirung
and Kimoshung Glacier, and the lower uncertainty estimates for debris-covered areas (Figure
**5**a). The individual uncertainty estimates for each glacier reflect their topographical
characteristics and correlate well with the number of identified outliers per glacier (Table S1).





Although the uncertainties discussed above are representative of calculated elevation changes
and not of DEM accuracy, we can characterize relative DEM accuracies by comparing the
average uncertainty of all Δh/Δt maps generated with a given DEM (Table 4). Of all Δh/Δt
maps which were not rejected, the mean off-glacier uncertainty weighted by the hypsometry
of the total glacier area is 0.27 m/a. The highest uncertainties are attributed to DEM pairs
involving the ALOS-PRISM DEM (0.4 m/a) and the lowest to those involving the SPOT7
Oct 2015 DEM (0.21 m/a). The mean uncertainty estimates correlate with the glacier Δh/Δt
map rejection rates (also provided by Table 4). An exception is represented by the
Hexagon1974 DEM with low rejection rates but a relatively high average uncertainty of 0.34
m/a. The Nov 1974 - Oct 2006 Δh/Δt map (Figure 6a) reveals an irregular and unrealistic
distribution of Δh/Δt values at high altitudes, which can be likely associated to errors in the
Hexagon 1974 DEM. Outlier correction can attenuate the effect of such errors, but cannot
completely eliminate them. The higher uncertainties calculated for Δh/Δt maps involving the
Hexagon 1974 DEM seem therefore justified. However, the uncertainties at lower elevations
are not higher for Δh/Δt maps involving the Hexagon 1974 DEM than for other maps (see the
average uncertainties representative of the debris-covered tongues, Table 4). The average
uncertainty of all Δh/Δt maps representative of the debris-covered areas (0.06 m/a) is
substantially lower than the corresponding value representative of all glacierized areas (0.27
m/a). This difference reflects the extreme altitudinal range of glaciers in the study region and
topographical characteristics at high altitudes that reduce DEM accuracy.
**5   Results**
**5.1   Mean glacier surface elevation changes**
The ensemble of mean elevation changes consistently indicates an increase in mean thinning
rates between 2006 and 2015 in comparison to the periods starting in 1974, both at the total
glacier area and at debris-covered glacier area (Figure 7a and b). From the debris-covered
glaciers, Langtang and Langshisha Glaciers seem to undergo stronger thinning during the
recent decade than before the turn of the century (Figure 7e and g). For Shalbachum and
Ghanna Glaciers (Figure 7i and m), the scatter in the values is such that no clear trends in
mean thinning rates can be identified, but a majority of values suggest that thinning rates
remained approximately constant after 2006 in comparison to 1974-2006.



The ensemble of values helps to distinguish between trends that should be classified as
uncertain (Shalbachum and Ghanna Glaciers) from trends that are consistent within the
ensemble (Langtang and Langshisha Glaciers). Differences in values between largely
overlapping periods should be attributed to uncertainty, as suggested by the uncertainty
bounds (Figure 7). For Lirung Glacier an ensemble representation of values for the recent
periods is not possible, since a majority of values did not fulfil the outlier criteria. The
remaining values suggest slightly higher thinning rates in recent years with respect to the
period 1974-2009 (Figure 7k).
The scatter in mean $\Delta h/\Delta t$ values of overlapping recent periods is much lower for the debris-
covered tongues (Figure 7b) than at the whole glacier scale (Figure 7a). The temporal trends
in thinning rates indicated for the debris-covered parts of glaciers are consistent within the
ensemble. This result corresponds well to the low uncertainty estimates for debris-covered
tongues (Figure 5a). A gradual acceleration of thinning within the last decade is suggested by
the ensemble results for Langtang, Langshisha and Shalbachum tongues (Figure 7f, h and j).
At Lirung tongue, the ensemble of values indicates significantly higher thinning rates in
recent periods than before the turn of the century, but recent trends are less clear. We assume
that the very high thinning rates calculated for the period Nov 2009 to April 2014 at Lirung
tongue (-2.2 m/a, orange line Figure 7l) are due to systematic errors in the differential DEM.
At Ghanna tongue, the values suggest a slight deceleration of thinning or constant thinning
rates prior/after 2006 (Figure 7n). The mean thinning rates calculated for Ghanna tongue
therefore clearly follow a different trend than mean thinning rates of other debris-covered
glacier areas.
Regarding the thinning trends of debris-free glaciers (Figure 7c and d) the interpretation is
again complicated by uncertainty. This is especially true for Kimoshung Glacier, where the
uncertainties in mean $\Delta h/\Delta t$ values are highest (Figure 5a). For this glacier the ensemble
consistently indicates close to zero elevation changes after 2006, while the 1974-2006 and
1974-2009 mean $\Delta h/\Delta t$ values are much more negative (-0.55 and -0.47 m/a, respectively).
The opposite behaviour is suggested by the results for Yala Glacier, where thinning rates
seem to gradually increase over time (Figure 7d).
The glaciers for which the most negative average elevation differences are calculated for the
period 1974-2006 are Shalbachum (-0.63 ± 0.38 ma[-1], Table 5), Kimoshung (-0.55 ±
0.73 ma[-1]) and Ghanna Glacier (-0.51 ± 0.13 ma[-1]). The least negative values were calculated





1. for Langshisha (0.02 ± 0.50 ma⁻¹), Lirung (values only available for 1974-2009: -0.14 ±

2. 0.51 ma⁻¹) and Langtang Glacier (-0.27 ± 0.41 ma⁻¹). Comparing the two periods 1974-2006

3. and 2006-Oct 2015, the strongest acceleration of thinning took place at Yala Glacier (from

4. -0.40 ± 0.25 ma⁻¹ to -1.00 ± 0.26 ma⁻¹, Table 5), which for the period 2006-Oct 2015 was the

5. glacier with the highest thinning rates. On average, glacier thinning rates increased by more

6. than 100% between the periods 1974-2006 (-0.28 ± 0.42 ma⁻¹) and 2006-Oct 2015 (-0.62 ±

7. 0.34 ma⁻¹). Only Kimoshung thinning rates decreased by 0.5 ma⁻¹ to -0.05 ± 0.53 ma⁻¹.

8. Thinning of debris-covered areas also increased on average (from -0.77 ± 0.04 ma⁻¹ to -1.01 ±

9. 0.06 ma⁻¹, Table 5). The most important differences in mean $\Delta h/\Delta t$ values are determined for

10. Lirung (difference between $\Delta h/\Delta t_{1974-2006}$ and $\Delta h/\Delta t_{2006-Feb2015}$: -0.55 ma⁻¹), Shalbachum

11. (-0.49 ma⁻¹) and Langshisha tongue (-0.36 ma⁻¹), while thinning of Langtang tongue between

12. the same two periods only increased moderately (-0.10 ma⁻¹) and decelerated at Ghanna

13. tongue (+0.05 ma⁻¹).

## 5.2 Temporal and spatial patterns

15. A visual inspection of $\Delta h/\Delta t$ values of debris-covered areas in Figure 6 suggests that areas of

16. very strong thinning (< -2 ma⁻¹) seem to have become more common in the past nine years

17. (Oct 2006-Oct 2015, Figure 6b) in comparison to the 32-year period between Nov 1974 and

18. Oct 2006 (Figure 6a). However, glacier movement and changes in surface topography smooth

19. out heterogeneous elevation change patterns over time. To assess if area-average thinning

20. rates have changed over time we compare $\Delta h/\Delta t$ values averaged over 50 m elevation bands

21. of individual glaciers.

22. The altitudinal distributions of mean elevation changes clearly show that the thinning patterns

23. of all debris-covered tongues have changed over time (Figure 8). Areas with a clear increase

24. in thinning rates in recent years with respect to the earlier periods can be identified at

25. Langtang Glacier 4950-5200 m asl, at Langshisha Glacier 4600-5100 m asl, at Shalbachum

26. Glacier 4400-4800 m asl and at Lirung Glacier above 4250 m asl. Thinning rates mostly have

27. remained approximately constant over time near the terminus, in the lower third of elevation

28. ranges (Langtang, Shalbachum and Lirung Glaciers). At Ghanna Glacier, unambiguously

29. thinning rates have recently declined near the glacier terminus (Figure 8e). At Langshisha

30. Glacier (Figure 8b) the patterns near the glacier terminus are more ambiguous. Here, the

31. glacier tongue became very narrow in the last decade and ultimately a small part below

32. 4500 m asl disconnected from the main tongue (Figure 1) after 2010. The narrowing and





eventual fragmentation of the tongue leads to mean thinning rates close to zero if the relative
weight of disappearing glacier area is high, since always the maximal (initial) glacier area is
considered to calculate mean thinning rates per elevation band. This seems to have been the
case at Langshisha Glacier for periods starting in 2006 near 4500 m asl.
Small differences between profiles of overlapping periods from the last decade can be
attributed to uncertainty (uncertainty bounds in Figure 8). At the uppermost elevation bands
the profiles of overlapping periods diverge because the uncertainty increases. This is nicely
reflected by the uncertainty bounds for Langtang Glacier (Figure 8a), while at Lirung or
Ghanna Glacier (Figure 8d and e) the local error is likely underestimated by the uncertainty
bounds. At altitudes higher than the debris-covered tongues, the altitudinal $\Delta h/\Delta t$ profiles
diverge further (Figure 9). Above 5500 m asl it is impossible to separate uncertainty from
actual differences in thinning rates.
Yala Glacier experiences more rapid thinning over almost its entire elevation range in recent
periods (Figure 9d). This is in clear contrast to the altitudinal thinning profiles of debris-
covered glaciers, which present much less uniform patterns (Figure 9a-c). At Yala Glacier
maximal thinning takes place at the terminus and then decreases nearly linearly with altitude
until it reaches values close to zero (Figure 9d). At debris-covered glaciers, the elevation
corresponding to the maximum thinning rates is different from glacier to glacier. At
Shalbachum and Lirung Glaciers the maximum is reached somewhere close to the upper end
of the tongue (4650 – 4750 m asl and 4300 – 4400 m asl, respectively, Figure 8c and d), at
Langtang and Ghanna Glaciers more in the middle part (4950 – 5150 m asl and 4900 – 5000
m asl, respectively, Figure 8a and e) and at Langshisha Glacier closer to the terminus (4450 –
4700 m asl, Figure 8b). At the large debris-covered glaciers, areas of maximum thinning seem
to have shifted and extended to higher elevations only at Langtang Glacier, where during the
period 1974-2006 maximum thinning occurred between 4850 and 4950 m asl (Figure 8a). At
Langtang and Shalbachum Glaciers the difference between thinning near the terminus and
maximum thinning became much more pronounced in recent periods, but at Shalbachum
Glacier maximum thinning during the period 1974-2006 occurred slightly higher up at 4750 –
4800 m asl (Figure 8c).
Note that the altitudinal $\Delta h/\Delta t$ profiles (Figure 8, Figure 9) always refer to the same position
in space, since 50 m elevation bands were delimited only once on the basis of the SRTM
1 Arc-Second Global DEM. To account for the up-valley movement of on-glacier elevation





bands over time due to surface lowering, profiles would have to be slightly shifted relative to
each other. However, given the maximum thinning rates of 1-1.5 ma$^{-1}$ in early periods, the
maximum relative adjustment of values in Figure 8 and Figure 9 would never exceed one 50
m elevation band. Accounting for the shifting of elevation bands over time would therefore
not lead to different conclusions regarding changes in spatial Δh/Δt patterns.

### 5.2.1 Explanatory variables

We determine overall negative correlations between changes in Δh/Δt values over time and
σ Δh/Δt (r=-0.52, Figure 10a) and between changes in Δh/Δt values and Δh/Δt $_{Q50-Q10}$ (r=-0.55,
Figure 10b), respectively. This indicates a link between accelerated thinning and the presence
of supraglacial lakes and cliffs, but also between reduced thinning (e.g. Ghanna tongue,
Figure 8e) and homogeneous layers of debris. At debris-free glacier area, for comparison, we
calculate high positive correlations between spatial variability in elevation change values and
changes in mean thinning rates (r=0.91, Figure 10a; r=0.97, Figure 10b). This result can be
explained by increasing variability due to increasing uncertainty with altitude and at steeper
slopes (Figure 4, Figure 10c and d). Indeed, changes in thinning rates at debris-free glacier
area are essentially altitude dependent (Figure 9d, Figure 10c), whereas at debris-covered
glacier area the correlation with altitude is close to zero (Figure 10c).
The correlation between median surface velocity per elevation band and changes in Δh/Δt
values over time is rather low (r=0.31, Figure 10e). However, the comparability of surface
velocity fields across several glaciers is limited, since flow dynamics of debris-covered
tongues in the catchment are differing due to the diversity in glacier shapes. The uncertainty
in velocity estimates is also rather high, especially at narrow glacier tongues of smaller
glaciers where cross-correlation windows are likely to overlap with glacier borders. However,
at large debris-covered tongues the general patterns in the velocity fields indicate a clear
interdependence of ice velocities and thinning rates. We consistently find low velocities
below 2.5 m/a near the termini of debris-covered glaciers and higher velocities up to 25 m/a
in the upper reaches of large debris-covered tongues (Figure 11). The pattern of down-glacier
velocity decay agrees with the tendency of lower thinning rates and more homogeneous
thinning patterns near the glacier termini (Figure 6, Figure 2). Indeed, 77% of all elevation
bands where thinning accelerated (Δh/Δt$_{1974-06}$ - Δh/Δt$_{2006-15}$ < -0.2 m/a) are not stagnating
(Figure 10e), and in 72% of all elevation bands where thinning rates remained constant or
declined we observe stagnant conditions with velocities below 2.5 m/a (Figure 10e).





## 5.3 Impacts of the April 2014 earthquake

On 25 April 2015 the study area was struck by a 7.8 magnitude earthquake with an epicentre approximately 80 km west of the Langtang Valley. The earthquake triggered a large number of geohazards in Nepal and China such as landslides and avalanches (Kargel et al., 2016). Also in the upper Langtang catchment earthquake-induced avalanches occurred on Lirung, Langtang, Shalbachum and Langshisha Glaciers. The availability of two post-earthquake DEMs, one acquired less than two weeks after the earthquake on 7 May 2015 (Table 2), allows quantifying the impact of this singular event on debris-covered tongues. For this purpose we use the April 2014 - May 2015 Δh map (which is less affected by outliers than the Feb 2015 - May 2015 Δh map, Table S1) to quantify the accumulated volumes immediately after the earthquake, and the April 2014 - Oct 2015 Δh map to quantify the remaining volumes after one ablation season. To identify glacier area where avalanche material accumulated we consider all glacier grid cells with positive elevation changes by >5 m, which is approximately two times the standard deviation of off-glacier elevation differences calculated for the April 2014 - May 2015 Δh map. Approximately 7.9% (1.9 km$^2$) of all debris-covered areas were affected by avalanches according to this definition. To calculate the deposited volumes we first estimate the volume loss between April 2014 and April 2015 (pre-earthquake), considering the mean annual thinning rates of the identified avalanche affected areas between Oct 2006 and Feb 2015. We then sum these volumes with the volume change measured by DEM differencing between 21 April 2014 and 7 May 2015 to obtain accumulated avalanche material volumes.

We calculate a total volume of avalanche material that accumulated on debris-covered area of 2.49*10$^7$ m$^3$, which is equivalent to a cube length of 292 m. 40% of the avalanche material remained until 6 Oct 2015 (Table 6). The two glaciers which were most affected by avalanches were Langtang Glacier (receiving 58% of the total volume) and Lirung Glacier (29%). Figure 12 shows that the avalanche cone at Lirung Glacier piled up to a height of nearly 60 m, while the avalanche material at Langtang Glacier was more spread. Consequently, more material remained until 6 Oct 2015 at Lirung Glacier (57%), while at Langtang Glacier only 31% remained (Table 6). Field visits at the end of October 2015 revealed that a smooth debris layer melted out of the avalanche material and covered the surface uniformly with a thickness of a few centimeters (P. Buri an P. Egli, personal communication).





Considering the calculated volumes of avalanche deposits divided by the total debris-covered
area we can compare the deposited volumes to average annual volume loss. The avalanche
deposits remaining on 6 Oct 2015 are equivalent to an average surface elevation change by
+0.52 m (Table 6). Given the average $\Delta h/\Delta t$ rates between October 2006 and February 2015
of -1.01 ± 0.06 m/a (Table 5), the avalanches after the earthquake compensated by about 50%
the volume loss of one average year. Over periods of several years, the effect of the post-
earthquake avalanches on the altitudinal thinning profiles such as presented in Figure 8 is
therefore only minor. It is best visible at Lirung Glacier at 4350 m asl (dark red and orange
lines in Figure 8d), and slightly at Langtang tongue at about 4650 m asl (Figure 8a).
**7    Discussion**
**7.1    Spatial and temporal elevation change patterns**
Elevation change rates of debris-covered area which are not primarily elevation dependent
(Figure 8, Figure 10c) have already been identified in previous studies for glaciers in the
Langtang catchment (Pellicciotti et al., 2015) or elsewhere in high-mountain Asia (e.g. Bolch
et al., 2011; Dobhal et al., 2013; Pieczonka et al., 2013; Pieczonka and Bolch, 2015; Ye et al.,
2015). Such patterns have usually been explained by downward-increasing debris thickness
and by ablation associated with supraglacial lakes and exposed ice cliffs. Our analysis shows
that the highest thinning rates and the strongest increase in thinning rates can be associated to
areas with patchy, spatially highly variable elevation change patterns, characteristic of areas
with a high concentration of ice cliffs and supraglacial ponds (Figure 2, Figure 10). While
previous studies have pointed out that debris-covered areas with a large presence of
supraglacial cliffs and lakes make a disproportionately large contribution to ablation (e.g.
Bolch et al., 2011; Zhang et al., 2011; Juen et al., 2014; Reid and Brock, 2014; Buri et al.,
2016; Miles et al., 2016a), this is the first study which shows the correlation between complex
thinning patterns and accelerations in volume loss rates at the scale of multiple glacier
tongues.
Accelerated thinning of debris-covered area in the Upper Langtang catchment does not take
place at stagnating parts of the tongues, but in contrary at areas where debris-covered glacier
area is dynamically active (Figure 10e), and where the transition between the active and the
stagnant ice can be expected. Compressive stresses in the down-glacier direction associated
with flow deceleration may initiate fracturing (Benn et al., 2009). Such stresses are usually





not large enough to initiate open surface crevasses, but in combination with elevated water
pressure below the margins of supraglacial lakes lead to hydrologically driven fracture
propagation (hydrofracturing) and englacial conduit formation (Benn et al., 2009). The
collapse of large englacial voids destabilizes the debris layers and may lead to the formation
of new ice cliffs. This explains why high values of σ Δh/Δt and thus thinning accelerations are
associated to active glacier dynamics. Higher ablation rates up-glacier than at the terminus
cause a reduction of the glacier surface gradient, which in turn leads to further glacier
slowdown and stagnation due to reduced driving stresses (Quincey et al., 2009; Jouvet et al.,
2011; Benn et al., 2012). It is therefore likely that reduced ice fluxes and enhanced melt at
supraglacial cliffs/lakes both contribute to the observed thinning accelerations at debris-
covered tongues. Reduced ice flux could explain why the areas of maximum thinning
migrated to slightly higher elevations on Langtang Glacier (Figure 8a), and why a new local
maxima at 4650 m - 4750 m asl emerged on Shalbachum Glacier (Figure 8c). In order to
assess which of the two factors contribute most to the observed accelerations in thinning, it
would be necessary to quantify changes in ice flux over time (e.g. Nuimura et al., 2011;
Berthier and Vincent, 2012; Nuth et al., 2012). For such an assessment information about the
evolution of surface velocities over long time periods would be required, which our dataset
cannot provide. However, given the usually very slow dynamical response of debris-covered
glaciers to changes in the local temperature (Banerjee and Shankar, 2013) it can be assumed
that reductions in glacier uplift are not the primary factor that cause the observed thinning
accelerations. Over the timescales considered in this study, on the other hand, high warming
rates have been identified in this part of the Himalaya (Shrestha et al., 1999; Lau et al., 2010).
The rise in air temperatures directly impacts glacier melt rates, and explains rapid acceleration
of thinning where ice is not insulated from warming by thick debris.
Previous studies have provided numerical evidence that ablation rates of debris-covered ice
may decrease over time as a consequence of thickening debris cover, in spite of rising air-
temperatures (Banerjee and Shankar, 2013; Rowan et al., 2015). This insulating effect might
even lead to terminus advance during warmer climatic periods (Kellerer-Pirklbauer et al.,
2008). Due to long response times this somewhat counterintuitive response of debris-covered
glaciers is difficult to observe. However, near the snout of Ghanna Glacier a deceleration in
thinning rates can be clearly identified, and also in the lower ablation areas of Lirung,
Langtang and Shalbachum Glaciers the ensemble of thinning rates point to a slowly
decreasing trend (Figure 8). Terminus advances, on the other hand, have not been observed





(Table 7) and are unlikely to occur at the five studied debris-covered glaciers due to frontal
ablation (evident from higher thinning rates at the lowest elevation band at each profile in
Figure 8).
Other authors have suggested that slope can be used as a proxy for debris-covered glacier
sections where ice cliffs are prone to form, favoring increases in ice losses (Nuimura et al.,
2012; Pellicciotti et al., 2015). Most of the elevation bands of debris-covered sections are
gently sloped (74% have a mean slope of less than 10°). However, we do not find a
correlation between slope and thinning acceleration (Figure 10d). We explain this by the low
variability in slopes at the scale of glacier tongues, whereas other studies have analyzed the
connection between slope and elevation change at larger scales (Nuimura et al., 2012) or
across the entire elevation range of debris-covered glaciers (Pellicciotti et al., 2015).

### 7.1.1 Differences between individual debris-free glaciers

Debris-free Yala Glacier experienced by far the strongest increase in relative annual area loss
of all studied glaciers (1974-2006: -0.43% $a^{-1}$, 2006-2015: -1.77% $a^{-1}$, Table 7). Of all glaciers
in our sample, this is also the glacier for which the strongest acceleration in mean thinning is
identified (Table 5). Areal changes of debris-free Kimoshung Glacier correlate with the
average surface height changes only for the recent period 2006-2015. During this period the
glacier area of Kimoshung Glacier decreased at a rate of 0.05% per year, which is 35-times
less than at Yala Glacier. Accordingly, Kimoshung Glacier experienced much less thinning
(-0.05 $ma^{-1}$, Table 5) than Yala Glacier (-1.00 $ma^{-1}$) during the same period. However, the
retreat rates of Kimoshung Glacier were relatively low also during the period 1974-2006
(-0.08% $a^{-1}$, Table 7), while the calculated average elevation change rates for the same period
are significantly more negative during this period (-0.55 $ma^{-1}$, Table 5). Likely, the identified
average $\Delta h/\Delta t$ values for Kimoshung Glacier for earlier periods (Figure 7c) overestimate the
actual thinning rates. Presumably, unrealistically high thinning rates at high altitudes due to
errors in the Hexagon 1974 DEM led to this result (Figure 6a). The actual $\Delta h/\Delta t$ values still
can be expected within the indicated uncertainty bounds but closer to steady-state conditions.
By the differences in mean elevation change rates between Kimoshung and Yala Glacier we
can demonstrate how the response of debris-free glaciers strongly depends on glacier
hypsometry. Almost balanced mass budgets in recent years can be associated to high
accumulation area ratios such as characteristic of Kimoshung Glacier, although thinning of





debris-free glacier area below the equilibrium line altitude is accelerating rapidly (Figure 9d).
Only a small fraction of area of Kimoshung Glacier is exposed to rising temperatures above
freezing level, and due to its steep tongue the accumulation area ratio (AAR) is not sensitive
to changes in the 0°C isotherm altitude. The balanced conditions of Kimoshung Glacier
therefore indicate that precipitation in recent decades remained approximately stable, which
agrees with the findings of studies on precipitation trends in this part of the Himalaya
(Shrestha et al., 2000; Immerzeel, 2008; Singh et al., 2008). Yala Glacier, on the other hand,
is sensitive to fluctuations in temperature and is therefore thinning rapidly due to recent
warming.

### 10   7.1.2  Differences between individual debris-covered glaciers

In comparison to the current retreat rates of Yala Glacier, all debris-covered glaciers are
shrinking at a much slower pace, with retreat rates between -0.04% $a^{-1}$ and -0.40% $a^{-1}$ (Table
7). It is interesting to note that the debris-covered glacier for which we currently observe the
highest annual volume loss (Shalbachum Glacier, -0.70 ± 0.31 m/a, Table 5) has an almost
stationary front (area loss -0.04% $a^{-1}$, Table 7). Ghanna Glacier in contrast is retreating at the
highest pace of all debris-covered glaciers (-0.40% $a^{-1}$, Table 7) although the thinning rates
have significantly declined near the terminus in recent periods (Figure 8e). Ghanna Glacier is
also the only debris-covered glacier where average annual volume loss at the tongue did not
accelerate in recent periods (Table 5, Figure 7n), although relative area loss seem to have
increased slightly (from -0.33% $a^{-1}$ to -0.40% $a^{-1}$, Table 7).
Banerjee and Shankar (2013) numerically investigated the response of extensively debris-
covered glaciers to rising air-temperatures and describe the dynamical response as follows:
during an initial period the fronts remain almost stationary and in the ablation region a slow-
flowing quasi-stagnant tongue develops. During this period, which may last more than 100
years, glaciers loose volume by thinning. After this initial period glaciers start to retreat with a
higher rate, while annual volume loss decreases because of thickening debris layers. The
response time of such glaciers depend on local climate and geometrical properties (e.g. slope,
length), but smaller glaciers with thinner initial glacier tongues and lower flow speeds can be
generally assumed to pass the initial period with stationary lengths faster than large valley
glaciers. Our observations therefore suggest that Ghanna Glacier, which is the smallest glacier
in the sample, is already entering the second period, since annual volume loss of the tongue is
decreasing but retreat rates are increasing. All large debris-covered glaciers in the valley



(Langtang, Langhsisha, Shalbachum) are still responding to increasing temperatures by
accelerated thinning rather than retreat. Since thinning near the glacier fronts has not yet
started to substantially decrease (Figure 8) and the glacier tongues are still dynamically active
(Figure 11) it can be assumed that the quasi-stationary length period will persist in the near
future. The model of Banerjee and Shankar (2013) does not account for supraglacial cliffs and
lakes, which likely cause an acceleration in thinning (Figure 10a and b). However, we have
shown that they primarily appear on parts of the glacier tongues which are still dynamically
active. It can thus be assumed that they become less abundant with decreasing flow, such as it
is already the case in the lower section of Langtang Glacier (Figure 2b). The presence of cliffs
and lakes therefore does not interfere with the dynamical response of debris-covered glaciers
as described by Banerjee and Shankar (2013).

### 7.1.3 Differences between debris-free and debris-covered glaciers

The response of debris-covered and debris-free glaciers to a warming climate is substantially
different, as described in the two sections above and exemplified by the altitudinal elevation
change profiles in Figure 9. Our observations do not support the conclusion of previous
studies (Kääb et al., 2012; Nuimura et al., 2012; Gardelle et al., 2013) that the present-day
lowering rates of debris-covered glacier areas in high mountain Asia might be similar to those
of debris-free areas at the same elevation. In the Upper Langtang catchment this might be the
case only very locally, e.g. comparing the elevation change rates of large debris-covered
tongues and Kimoshung tongue (Figure 6b). However, Kimoshung Glacier is rather unique,
since the very high AAR of 86% (Table 1) leads to a high ice flux, which also explains why
its tongue reaches unusually low elevations (terminus at 4385 m asl, Table 1). Yala Glacier
has an AAR of 40% (Table 1), which is a more common value in the HKH region (Kulkarni,
1992; Kulkarni et al., 2004). The glacier reaches a minimum elevation of about 5150 m asl,
which is the altitude where maximum thinning occurs on Langtang Glacier (Figure 8a, Figure
9a). However, maximum thinning per elevation band does not exceed 1.3 - 1.5 m/a at the
main tongue of Langtang Glacier (Figure 8a), while at the same altitude Yala Glacier is
thinning by about 1.8 - 2.3 m/a in recent periods (Figure 9d). This means that in spite of
enhanced melt from supraglacial cliffs and at supraglacial lakes (Figure 2), within the same
altitudinal range thinning rates of debris-covered glaciers do not exceed 65% - 75% of the
thinning rates at Yala Glacier. In addition, a comparison of thinning rates at the same
altitudinal range between debris-free and debris-covered glaciers is not indicative of their





climate responses, since debris-covered glaciers usually reach much lower elevations than
debris-free glaciers and are often thinning less near the glacier terminus (Figure 8).
The comparison of mean surface elevation change rates 1974-2006 and 2006-2015 (Table 5)
reveals that both the debris-free and the debris-covered glaciers show a heterogeneous
response to climate. Considering the values in Table 5, average thinning of all glaciers has
increased in the recent period (except at Kimoshung Glacier, but which likely derives from
errors in the Hexagon 1974 DEM, see Section 7.1.1). However, as discussed in the two
sections above, there are examples for both types of glaciers where thinning seems to increase
rapidly or where thinning remains approximately constant. A main difference between debris-
free and debris-covered glaciers in our sample cannot be identified with regard to average
thinning in recent periods. In our sample, the best predictor for average thinning between
2006 and 2015 seems to be the altitude distributions of glaciers, since glaciers with a high
AAR (Kimoshung) or which reach the highest elevations (Lirung) have the most balanced
mass budgets (Table 5), whereas glaciers with low AAR and which span over the relatively
small elevation ranges (Yala, Ghanna) are thinning most rapidly.
Note that lower thinning rates at debris-covered tongues are not in contradiction to the overall
mass balances trends, where no major difference between glacier types can be identified.
Debris-covered glaciers have low-lying tongues with shallow slopes, whereas the tongues of
debris-free glaciers are shorter and often much steeper, which means that smaller areas are
exposed to the high air-temperatures which cause rapid downwasting.
**7.2   Comparison with other studies**
The four largest debris-covered glaciers in the valley (Langtang, Langshisha, Shalbachum,
Lirung) have been the focus of a recent geodetic mass balance study by Pellicciotti et al.
(2015), who reconstructed elevation and mass changes using the 1974 Hexagon DEM which
is also used in this study (spatial resolution 30 m) and the 2000 SRTM3 DEM (90 m). They
found that all four glaciers were losing mass over the study period but with different rates (on
average -0.32 ± 0.18 m w.e. a$^{-1}$). We find an overall glacier mass balance for the period 1974-
2006 of the four glaciers which is only slightly less negative (-0.24 ± 0.35 m w.e. a$^{-1}$).
Correction of the Pellicciotti et al. (2015) results by a larger penetration estimate of the SRTM
radar signal, as suggested by Kääb et al. (2015), reconciles their results with ours. Differences
in the mass balance of Langtang, Lirung and Shalbachum Glacier are within uncertainty





bounds and can be attributed to differences in used glacier masks, study period, outlier
correction approaches and density assumptions. However, for Langshisha Glacier we
calculate a mass balance which is substantially less negative than in Pellicciotti et al. (2015).
While we identify almost balanced conditions for the period 1974-2006 (0.01 ± 0.39 m w.e. a$^{-1}$
, Table 5), the mass balance indicated by Pellicciotti et al. (2015) is very negative (-0.79 ±
0.18 m w.e. a$^{-1}$). The main reason for this are uncertainties in the Hexagon DEM which are
accounted for differently. In the present study, volume gains in the accumulation area of
Langshisha Glacier are not per-se identified as outliers, whereas in Pellicciotti et al. (2015)
those areas are completely masked out (Fig. 3 in Pellicciotti et al. (2015)). Likely, the actual
mass balance of Langshisha Glacier is somewhere between the two average values. Note that
the higher uncertainty estimates for the present study are due to differences in the uncertainty
quantification approach, and cannot be related to differences in data quality. At Langshisha
Glacier the uncertainty estimates of the present study are certainly more realistic.
Yala Glacier has been frequently visited for field measurements in the last 25 years.
Sugiyama et al. (2013) calculated mean thinning rates of Yala Glacier for the periods 1982-
1996 (–0.69 ± 0.25 m a$^{-1}$) and 1996-2009 (–0.75 ± 0.24 m a$^{-1}$) on the basis of GPS surveys.
The values suggest a more moderate acceleration of volume loss rates than presented by our
study (Table 5, -0.40 ± 0.25 m a$^{-1}$ 1974-2006 to -1.00 ± 0.26 m a$^{-1}$ 2006-2015). However,
similarly to our study Sugiyama et al. (2013) identified a rapid acceleration of thinning rates
at the lowest elevations. Our ensemble of average elevation change rates representative of
recent periods also suggests especially strong increases in thinning during the last couple of
years (Figure 7d).
Ragettli et al. (2015) used a glacio-hydrological model to calculate the mass balances of all
glaciers in the upper Langtang catchment for the hydrological year 2012/2013. They used
glaciological and meteorological field data from Lirung and Yala Glacier to calibrate the melt
parameters taking into account the effect of variable debris thickness and spatio-temporal
changes in surface albedo. The calculated average mass balance of glaciers in the valley was
-0.24 m w.e. Here we identify mass balances which were substantially more negative during
recent periods (-0.51 ± 0.30 m w.e. a$^{-1}$, Table 5). However, the hydrological year 2012/2013
was one of the wettest years since 1990 (Ragettli et al., 2015), which explains the less
negative mass balances.





The acceleration in mass loss in recent periods identified by this study agrees with other
studies from the Nepalese Himalaya which assess multi-temporal elevation changes (Bolch et
al., 2011; Nuimura et al., 2012). Bolch et al. (2011) identify an increase in mass loss rates by
0.47  m w.e. a$^{-1}$ comparing the two periods 1970-2007 (-0.32 ± 0.08 m w.e. a$^{-1}$) and 2002-
2007 (-0.79 ± 0.52 m w.e. a$^{-1}$). Nuimura et al. (2012) calculate increasing mass losses in the
same study region between 1992-2008 (-0.26 ± 0.24 m w.e. a$^{-1}$)  and 2000-2008 (-0.45 ±
0.60 m w.e. a$^{-1}$). However, the uncertainties in the mass loss estimates by Bolch et al. (2011)
and Nuimura et al. (2012) are higher than the identified acceleration in glacier thinning.
Moreover, the mass loss estimates of Gardelle et al. (2013) for Khumbu region and the period
2001-2011 (average of -0.41 ± 0.21 m w.e. a$^{-1}$) are in the same order as calculated by Bolch et
al. (2011) for 1970-2007. The ensemble approach of this study can therefore substantially
strengthen previous conclusions that mass loss of glaciers in the Central Himalaya is
accelerating. Although the uncertainty bounds of average elevation changes calculated for
different periods overlap, volume changes calculated over a multitude of periods between
1974 and 2015 consistently indicate that glacier thinning is indeed accelerating (Figure 7a).
**8   Conclusions**
This study presents glacier volume changes in the upper Langtang catchment in Nepal of 28
different periods between November 1974 and October 2015 based on 8DEMs derived from
high-resolution stereo satellite imagery. The large dataset of elevation change maps was
systematically checked for outliers in order to consider only those elevation change maps
which are least affected by uncertainty. The ensemble of remaining maps provides one of the
most rigorous documentations to date of glacier response to climate change over the last 40
years in the Himalaya.
To analyze spatial and temporal patterns of thinning we addressed three main points. First, we
assessed if thinning rates of glaciers in the region have accelerated in recent years. Second, we
determined if spatial thinning patterns have changed over time and third, we addressed if
there are major differences between the response of debris-covered and debris-free glaciers in
the sample. Regarding the first point, we could constrain the findings of previous studies
(Bolch et al., 2011; Nuimura et al., 2012) that glacier wastage in the Central Himalaya is
accelerating. Glacier volume decreased during all periods between 2006 and 2015 (2006 -
2015: -0.60 ± 0.34 m a$^{-1}$) and at higher rates than between 1974 and 2006 (-0.28 ± 0.42 m a$^{-1}$).
However, whereas a majority of glaciers in the study region are thinning rapidly, glaciers with




a high percentage of glacier area at very high elevations have almost balanced mass budgets
in recent years and experienced no or only minor accelerations in thinning.
Regarding the spatial thinning patterns, the focus was on the extensively debris-covered
tongues of five glaciers in the study region. In the upper reaches of the tongues, thinning has
mostly accelerated in recent years, while the nearly stagnant areas near the terminus show
constant or decreasing thinning rates. The highest thinning rates and the strongest increase in
thinning rates can be associated to areas with complex, spatially heterogeneous elevation
change patterns, characteristic of areas with a high concentration of ice cliffs and supraglacial
ponds. Constant to clearly decelerating thinning rates can be associated to areas with
relatively homogeneous debris layers near the termini of glaciers. We conclude that the
response of extensively debris-covered glaciers to global warming is largely determined by
feedback processes associated to different surface characteristics.
Finally, regarding the third objective of this study, we point out the importance of
differentiating between spatial patterns and temporal patterns. Regarding the temporal
patterns and therefore the trends in average elevation change rates, no clear difference
between debris-free and debris-covered glaciers can be identified. In this respect, the behavior
of glaciers in the study area is highly heterogeneous, and the presence of debris itself is not a
good predictor for mass balance trends. However, the spatial thinning patterns on debris-
covered glaciers are fundamentally different than on debris-free glaciers. While on debris-free
glaciers thinning rates are linearly dependent on elevation, debris-covered glaciers have
highly non-linear altitudinal elevation change profiles. Still, throughout the entire elevation
range the thinning rates of debris-covered tongues are lower than at corresponding altitudes of
debris-free glaciers. Our observations do therefore not provide evidence for the existence of a
so-called debris-cover anomaly, where the insulating effect of thick supraglacial debris is
compensated by enhanced melt from exposed ice cliffs or due to high energy absorption at
supraglacial ponds.
Future work should be devoted to look at larger glacier samples to compare the response of
debris-free and debris-covered glaciers. Large-scale datasets of elevation change
measurements (e.g. Kääb et al., 2012; Gardelle et al., 2013) have been used to compare the
response of debris-free and debris-covered glaciers, but should consider also the differences in
elevation distribution and the non-linearity of altitudinal elevation change profiles. Finally,
for an in-depth analysis of debris-free and debris-covered glacier response to climate, also



glacier uplift and therefore changes in ice flux over time should be quantified, which would
enable the calculation of ablation rates instead of only thinning rates. A better knowledge of
melt rates, particularly in presence of supraglacial cliffs and lakes, would substantially
advance our understanding of debris-covered glacier response to climate.
On 25 April 2015, triggered by a magnitude 7.8 earthquake, a large avalanche went down on
Lirung Glacier and caused a strong pressure blast that devastated the trekking village of
Kjangjing consisting of about 30 houses. Further down-valley co-seismic snow and ice
avalanches and rockfalls destroyed Langtang Village and killed or left missing at least 350
people (Kargel et al., 2016). On the basis of two post-earthquake DEMs we quantified the
avalanche impacts on the mass balance of debris-covered tongues. At the end of the 2015
ablation season, avalanche deposits outweighed by 50% the average annual volume loss of
debris-covered glacier area during the last decade. We conclude that the impact of the
earthquake on the cryosphere is almost as disproportional to the impact of global warming on
glaciers in this region as it was disproportional to the impact on human lives.

**15  Acknowledgements**

This study is funded mainly by the Swiss National Science Foundation (SNF) project
UNCOMUN (Understanding Contrasts in High Mountain Hydrology in Asia). T. Bolch
acknowledges funding through German Research Foundation (DFG, code BO 3199/2-1) and
European Space Agency (Glaciers_cci project, code 400010177810IAM). We thank Evan
Miles for helping with the glacier delineations and the post-processing of surface velocity
data. Jakob Steiner and Pascal Buri created the map of cliffs and lakes on Langtang Glacier,
which is gratefully acknowledged. We thank Etienne Berthier for the Pléiades image data, for
his useful comments regarding the DEM extraction and Fanny Brun for her help with the
identification of GCPs. DigitalGlobe imagery was used to produce the WorldView-1 and 2
digital elevation models.



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

| | Name | Area | Debris cover | Mean slope | Mean slope glacier tongue* | AAR** | Elevation range |
|---|---|---|---|---|---|---|---|
| | | km² | km² | % | % | | m asl. |
| 1 | Langtang | 46.5 | 15.5 | 17.1 | 7.2 | 52% | 4479-6615 |
| 2 | Langshisha | 16.3 | 4.5 | 17.7 | 7.5 | 55% | 4415-6771 |
| 3 | Shalbachum | 10.2 | 2.6 | 16.9 | 9.1 | 52% | 4231-6458 |
| 4 | Lirung | 6.5 | 1.1 | 34.0 | 9.9 | 49% | 4044-7120 |
| 5 | Ghanna | 1.4 | 0.7 | 20.9 | 15.5 | 15% | 4721-5881 |
| 6 | Kimoshung | 4.4 | - | 24.4 | 32.1 | 86% | 4385-6648 |
| 7 | Yala | 1.9 | - | 22.7 | 20.3 | 40% | 5122-5676 |

*Here we consider the debris-covered area for glaciers with debris-covered tongues and all glacier area below
5400 m asl. for debris-free glaciers.

**Assuming an equilibrium line altitude of 5400 m asl (Sugiyama et al., 2013; Ragettli et al., 2015)

3    Table 2. Remote-sensing data used

| Sensor | Date of acquisition | Stereo mode (b/h-ratio) | Spatial/radiometric Resolution | Role |
|---|---|---|---|---|
| Hexagon KH-9 | 23 Nov 1974 | Stereo (0.4) | 6-9m/8-bits | DEM differencing, glacier outlines |
| Cartosat-1 | 15 Oct 2006 | Stereo (0.62) | 2.5m/10-bits | DEM differencing, glacier outlines |
| Cartosat-1 | 9 Nov 2009 | Stereo (0.62) | 2.5m/10-bits | DEM differencing, velocities, glacier outlines |
| ALOS PRISM | 3 Dec 2010 | Tri-stereo (0.5) | 2.5m/8-bits | DEM differencing, velocities, glacier outlines |
| SPOT6 | 21 Apr 2014 | Tri-stereo (0.5) | 1.5m/12-bits | DEM differencing, glacier outlines |
| WorldView-2 | 2 Feb 2015 | Stereo (0.5) | 0.46m/11-bits | DEM differencing |
| WorldView-3 | 22 Feb 2015 | Stereo (0.5) | 0.31m/11-bits | DEM differencing |
| SPOT7 | 7 Mai 2015 | Tri-stereo (0.64) | 1.5m/12-bits | DEM differencing, glacier outlines |
| SPOT7 | 6 Oct 2015 | Tri-stereo (0.68) | 1.5m/12-bits | DEM differencing |
| Pléiades | 1 and 9 Nov 2014 | Across track stereo (0.4) | 0.5m/12-bits | Basis for georectification |





1   Table 3. Outlier criteria applied to filter glacier elevation change (Δh) signals.

| Criteria | Threshold |
|---|---|
| **Pixel scale (Section 3.4.1)** | |
| Correlation score | 0.5 |
| Δh outliers, debris free glacier area | 2σ* |
| Δh outliers, debris-covered glacier area | 3σ* |
| **Glacier scale (Section 3.4.2)** | |
| Data availability | 50% of all glacier grid cells |
| Outlier correction uncertainty | 0.61 m** |
| DEM adjustment uncertainty | 0.61 m** |
| **Catchment scale (Section 3.4.3)** | |
| Mean Δh/Δt stable terrain | $Q_{25}-1.5(Q_{75}-Q_{25}) < \Delta h/\Delta t < Q_{75}+1.5(Q_{75}-Q_{25})$ *** |
| σ Δh/Δt stable terrain | $Q_{75}+1.5(Q_{75}-Q_{25})$ *** |

\* σ is the standard deviation calculated separately for Δh values corresponding to debris-free glacier area and debris-covered glacier area, respectively.

\** The threshold value of 0.61 m is equivalent to the mean thinning of all considered glacier areas between October 2006 and October 2015.

\*** $Q_n$ is the n th percentile of all values in the ensemble (one value per Δh map).

2   Table 4. Summary of outliers and uncertainties per DEM.

| | | Glaciers | | | Debris-covered tongues | | |
|---|---|---|---|---|---|---|---|
| | | % rejected | MPL [a] | Mean *unc* [m/a] | % rejected | MPL [a] | Mean *unc* [m/a] |
| | All Δh/Δt maps | 46.9 | 3 | 0.27 | 17.1 | 1 | 0.06 |
| **ID** | **DEM** | | | | | | |
| 0 | Hexagon Nov 1974 | 6.1 | 39 | 0.34 | 0.0 | - | 0.04 |
| a | Cartosat-1 Oct 2006 | 32.7 | 6 | 0.22 | 0.0 | - | 0.04 |
| b | Cartosat-1 Nov 2009 | 28.6 | 2 | 0.22 | 14.3 | 1 | 0.06 |
| c | ALOS-PRISM Dec 2010 | 85.7 | 4 | 0.40 | 14.3 | 1 | 0.03 |
| d | SPOT6 April 2014 | 59.2 | 2 | 0.25 | 22.9 | 1 | 0.07 |
| e | WorldView Feb 2015 | 63.3 | 1 | 0.30 | 42.9 | 1 | 0.07 |
| f | SPOT7 May 2015 | 49.0 | 1 | 0.29 | 20.0 | 1 | 0.10 |
| g | SPOT7 October 2015 | 51.0 | 2 | 0.21 | 22.9 | 1 | 0.09 |

Rejection rates (in %) indicate the percentage of Δh/Δt maps which are classified as outliers according to at least one criterion listed in Table 3. Rejection rates per DEM are calculated considering only the 7 Δh/Δt maps in which a given DEM is involved. The total number of all glacier Δh/Δt maps is 196 (7 glaciers x 28 DEM difference maps), and the total number of debris-cover Δh/Δt maps is 140 (5 debris-covered tongues x 28 DEM difference maps).

*MPL* is the median period length corresponding to rejected Δh/Δt maps. Mean *unc* characterizes the DEM accuracies by averaging the uncertainties attributed to non-rejected Δh/Δt maps (Figure 5a).




Table 5. Glacier volume and mass changes 1974-2006, 2006-Feb 2015

| | Average elevation differences (m a$^{-1}$) | | Average mass balance (m w.e. a$^{-1}$) | |
|---|---|---|---|---|
| | Nov1974-Oct2006 | Oct2006-Feb2015 | Nov1974-Oct2006 | Oct2006-Feb2015 |
| **Glaciers** | | | | |
| Langtang | -0.27± 0.41 | -0.59± 0.33 | -0.23± 0.34 | -0.50± 0.30 |
| Langshisha | 0.02± 0.50 | -0.51± 0.36 | 0.01± 0.39 | -0.43± 0.32 |
| Shalbachum | -0.63± 0.38 | -0.70± 0.31 | -0.54± 0.34 | -0.59± 0.29 |
| Lirung | -0.14± 0.51* | -0.25± 0.14* | -0.12± 0.41* | -0.21± 0.12* |
| Kimoshung | -0.55± 0.73 | -0.05± 0.53 | -0.47± 0.61 | -0.04± 0.42 |
| Yala | -0.40± 0.25 | -1.00± 0.26 | -0.34± 0.22 | -0.85± 0.26 |
| Ghanna | -0.51± 0.13 | -0.61± 0.15 | -0.44± 0.14 | -0.52± 0.15 |
| Average | -0.28± 0.42 | -0.62± 0.34 | -0.24± 0.35 | -0.52± 0.30 |
| **Debris-covered areas** | | | | |
| Langtang | -0.79± 0.04 | -0.89± 0.06 | -0.68± 0.08 | -0.75± 0.10 |
| Langshisha | -0.69± 0.03 | -1.05± 0.06 | -0.59± 0.06 | -0.89± 0.11 |
| Shalbachum | -0.78± 0.04 | -1.27± 0.05 | -0.66± 0.08 | -1.08± 0.11 |
| Lirung | -1.03± 0.08 | -1.58± 0.03 | -0.87± 0.13 | -1.34± 0.12 |
| Ghanna | -0.57± 0.02 | -0.52± 0.04 | -0.48± 0.05 | -0.44± 0.07 |
| Average | -0.77± 0.04 | -1.01± 0.06 | -0.65± 0.08 | -0.86± 0.11 |

*Periods Nov 1974 - Nov 2009 and Nov 2009 - Oct 2015 (Lirung Glacier is not well represented on the Oct 2006 DEM, Table S1).

Table 6. Volume changes of debris-covered glacier tongues due to avalanches triggered by the
Nepal earthquake on 25 April 2015. The first three data columns provide the volume changes
of avalanche affected area divided by the total debris-cover area (Table 1).

| | 21 Apr 2014-25 Apr 2015* (m) | 25 Apr 2015-7 May 2015 (m) | 25 Apr 2015-6 Oct 2015 (m) | 6 Oct 2015, volume remaining (%) |
|---|---|---|---|---|
| Langtang** | -0.10 ±0.03 | 1.33 ±0.13 | 0.42 ±0.12 | 31.3% |
| Langshisha | -0.04 ±0.03 | 0.32 ±0.11 | 0.10 ±0.13 | 31.6% |
| Shalbachum | -0.11 ±0.02 | 0.74 ±0.18 | 0.31 ±0.04 | 42.5% |
| Lirung | -0.87 ±0.02 | 6.79 ±0.23 | 3.87 ±0.05 | 57.0% |
| Average | -0.13 ±0.03 | 1.31 ±0.13 | 0.52 ±0.09 | 39.5% |

*Estimation based on average annual melt Oct 2006 – Apr 2014
**Only lower part (south of 28°19'N), upper part not on April 2014 scene



Table 7. Glacier area changes over the periods 1974-2006 and 2006-2015.

| ID | Glacier name | 1974-2006 | | 2006-2015 | |
|---|---|---|---|---|---|
| | | km$^2$ | % a$^{-1}$ | km$^2$ | % a$^{-1}$ |
| 1 | Langtang | -2.65 | -0.17 | -0.45 | -0.11 |
| 2 | Langshisha | -0.48 | -0.09 | -0.13 | -0.09 |
| 3 | Shalbachum | -0.28 | -0.08 | -0.03 | -0.04 |
| 4 | Lirung | -0.45 | -0.20 | -0.05 | -0.08 |
| 5 | Ghanna | -0.16 | -0.33 | -0.05 | -0.40 |
| 6 | Kimoshung | -0.11 | -0.08 | -0.02 | -0.05 |
| 7 | Yala | -0.31 | -0.43 | -0.31 | -1.77 |

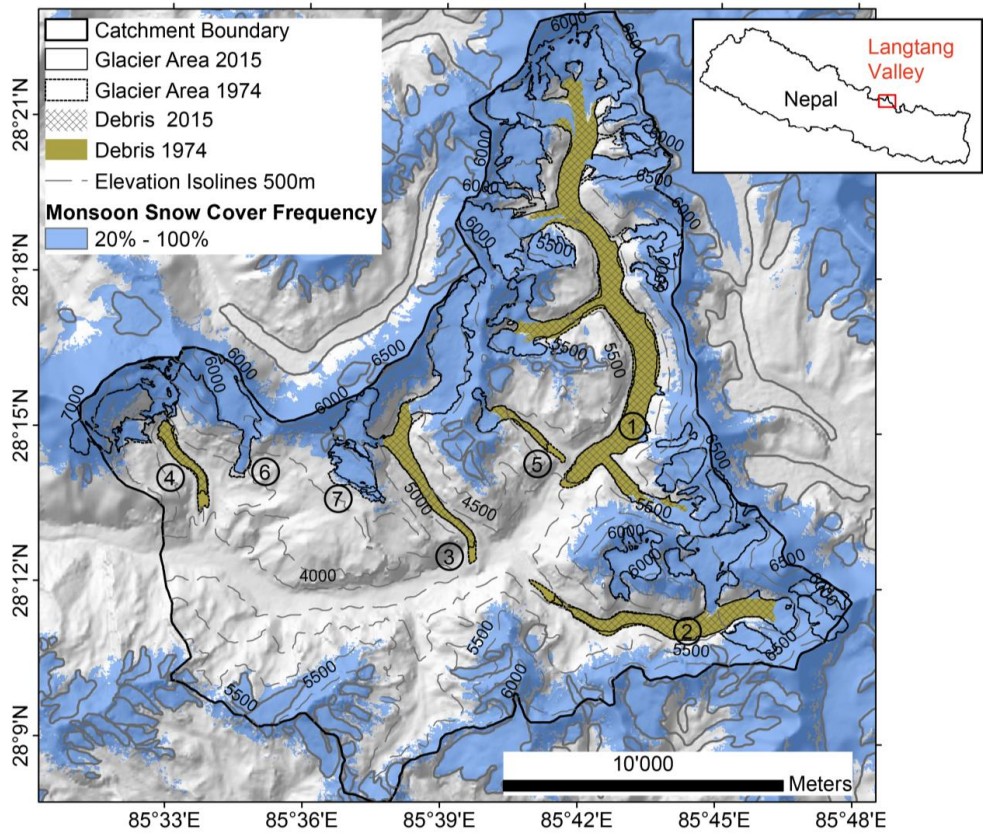

Figure 1. Map of the upper Langtang catchment. The numbers on the map correspond to the glaciers listed in Table 1. Monsoon snow-cover frequency is based on Landsat 1999 to 2013 land cover classifications (Miles et al., 2016b). 1974 glacier area (dotted lines) is shown for the seven studied glaciers only.





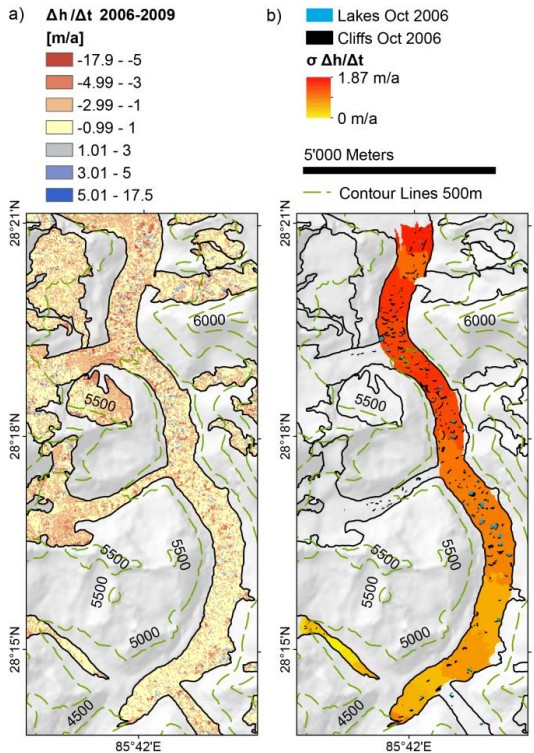

Figure 2: (a) Δh/Δt map of Langtang Glacier tongue, Oct 2006 – Nov 2009 (outlier corrected and missing data filled with inverse distance weighting), (b) standard deviation in Δh/Δt values calculated per 50m elevation band. The cliffs and lakes indicated in (b) were delineated manually on the basis of the Oct 2006 orthorectified satellite image.





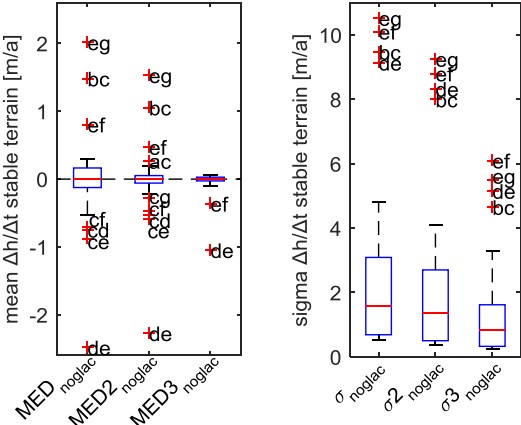

Figure 3. a) Off-glacier mean elevation differences and b) standard deviations ($\sigma$). On each box, the central mark is the median and the edges of the box are the 25th and 75th percentiles. The whiskers extend to the most extreme data points not considered outliers. Outliers are plotted individually and labelled by the corresponding DEM IDs (Table 4). $MED_{noglac}$ and $\sigma_{noglac}$ are calculated excluding the steepest slopes (slope<45°). For $MED2_{noglac}$ and $\sigma2_{noglac}$ areas with a monsoon snow-cover frequency higher than 20% (Figure 1) are masked out (slope<45°, sc≤20%). For $MED3_{noglac}$ and $\sigma3_{noglac}$ the threshold slope is defined as the 95th percentile of debris-covered glacier slope (slope<18°, sc≤20%).





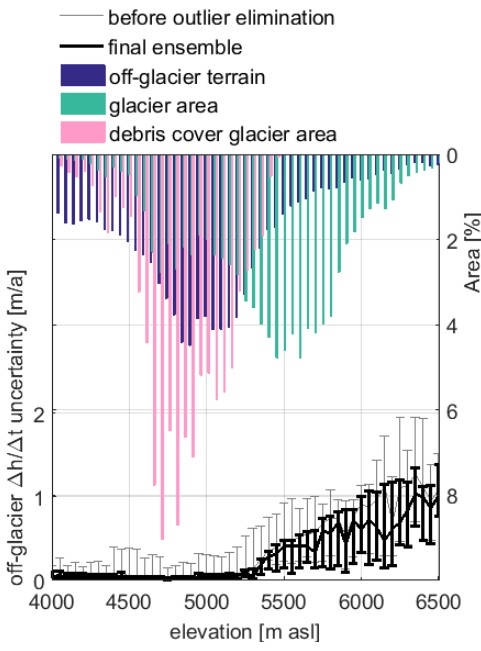

Figure 4. Off-glacier elevation change rate (Δh/Δt) uncertainty in function of elevation, prior
(grey line) and after (black line) outlier elimination (Figure 3). The lines represent the median
uncertainty (*unc*, eq. 5) per 50-m elevation band of all Δh/Δt maps in the ensemble. Error bars
represent the 50% confidence interval.





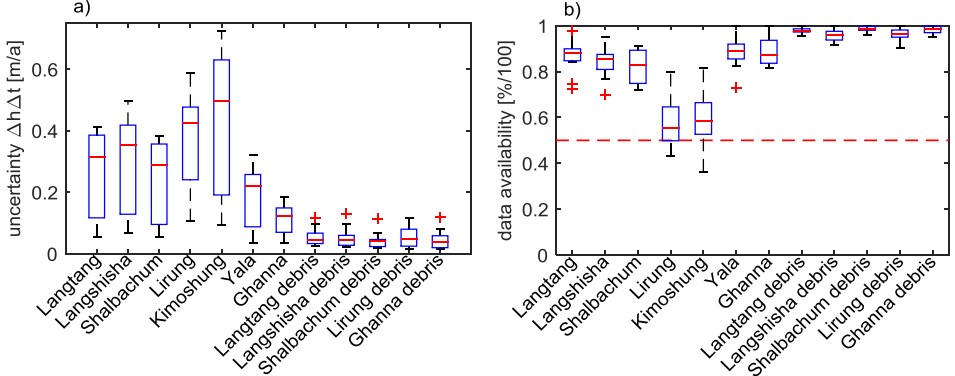

Figure 5. a) Uncertainty estimates of average elevation change rates (Δh/Δt) per individual glacier and per debris-covered tongue. b) Fraction of glacier pixels remaining after removing low stereo matching scores (Figure S1) and outliers. On each box, the central mark is the median of all Δh/Δt maps and the edges of the box are the 25th and 75th percentiles. The dashed red line in b) represents the threshold set for outlier elimination (50%).

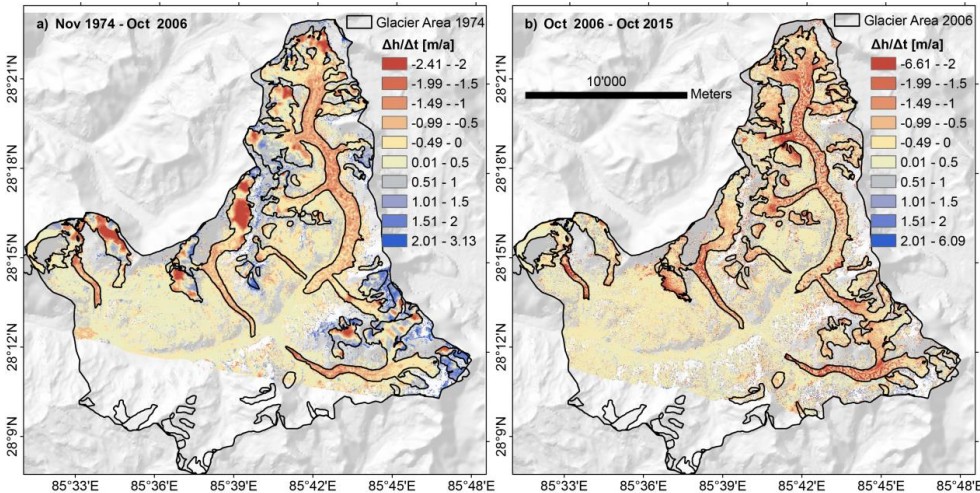

Figure 6. Elevation change rates (Δh/Δt) derived from a) Hexagon Nov 1974 and Cartosat-1 Oct 2006 DEMs and (b) Cartosat-1 Oct 2006 and SPOT7 Oct 2015 DEMs.





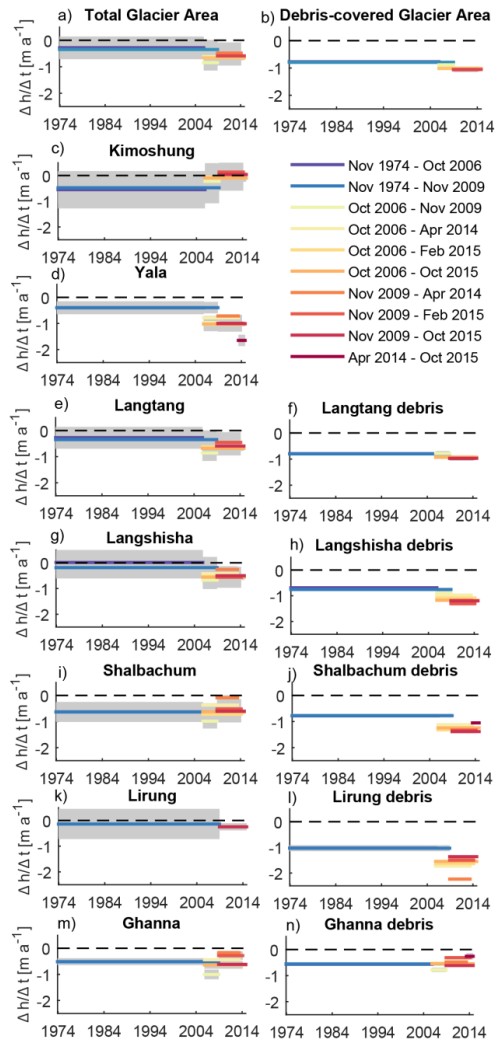

Figure 7. Mean elevation change rates ($\Delta h/\Delta t$) per period and glacier (left) or debris-covered area (right). Outliers according to Table S1 are not shown. For better readability, periods starting in 1974 and ending 2010 or later are not plotted. Periods involving the post-earthquake May 2015 SPOT7 DEM (Section 5.3) are not considered here. Uncertainty bounds (grey areas) correspond to uncertainty values derived for each glacier and debris-covered area individually (Figure 5a).





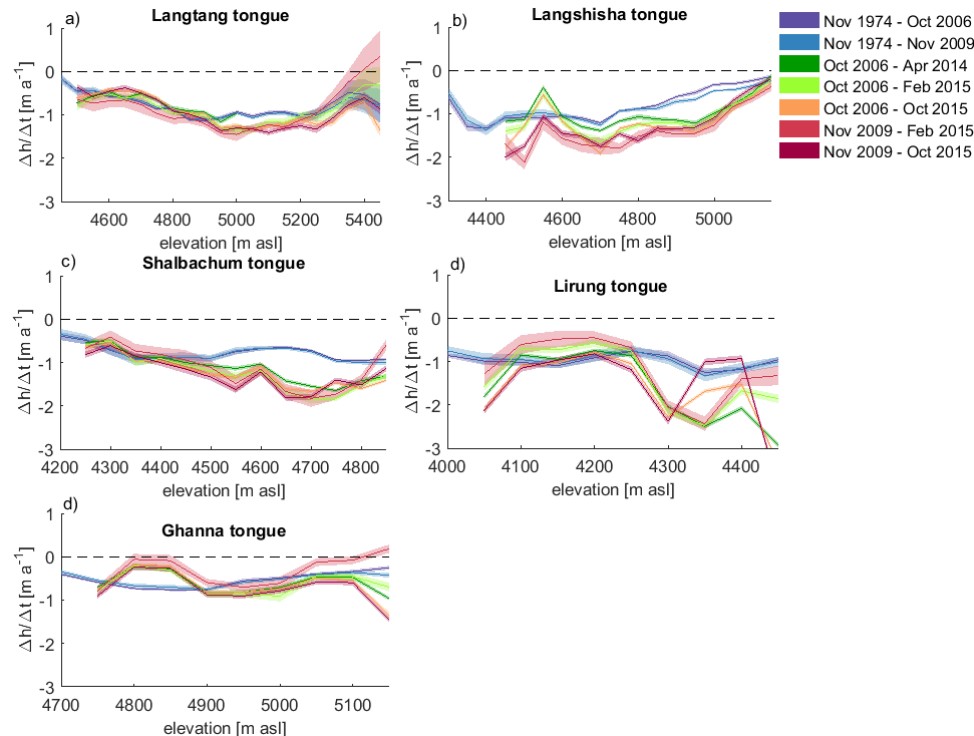

Figure 8. Altitudinal distribution of mean annual elevation change (Δh/Δt) over 50 m
elevation bands of debris-covered tongues (debris-covered area of each glacier excluding
tributary branches). For better readability, only selected periods not affected by outliers
(Table S1) are shown. Uncertainty bounds correspond to uncertainty in function of elevation
derived for each Δh/Δt map individually (Figure 4).





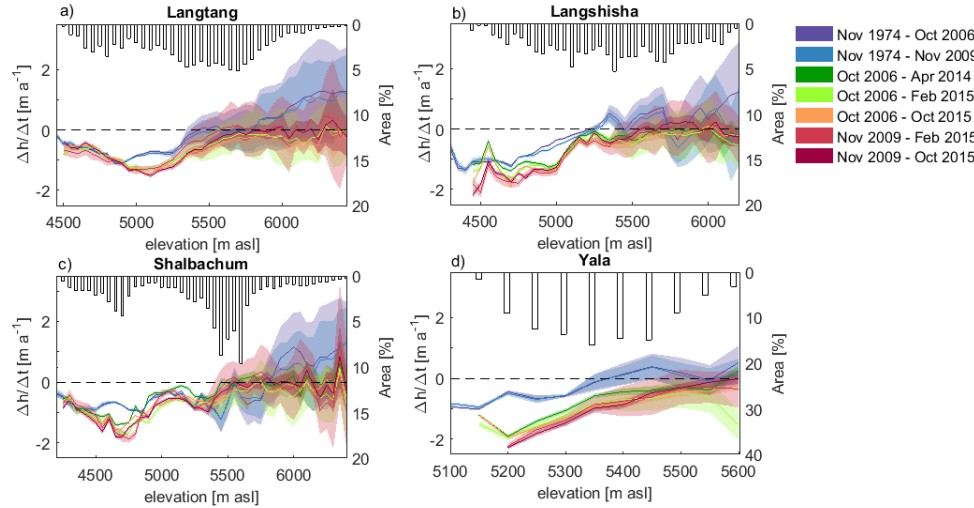

Figure 9. Altitudinal distribution of mean annual elevation change (Δh/Δt) and altitudinal distribution of glacier area (%) over 50 m elevation bands of selected glaciers. For better readability, only selected periods not affected by outliers (Table S1) are shown. Uncertainty bounds correspond to uncertainty in function of elevation derived for each Δh/Δt map individually (Figure 4).





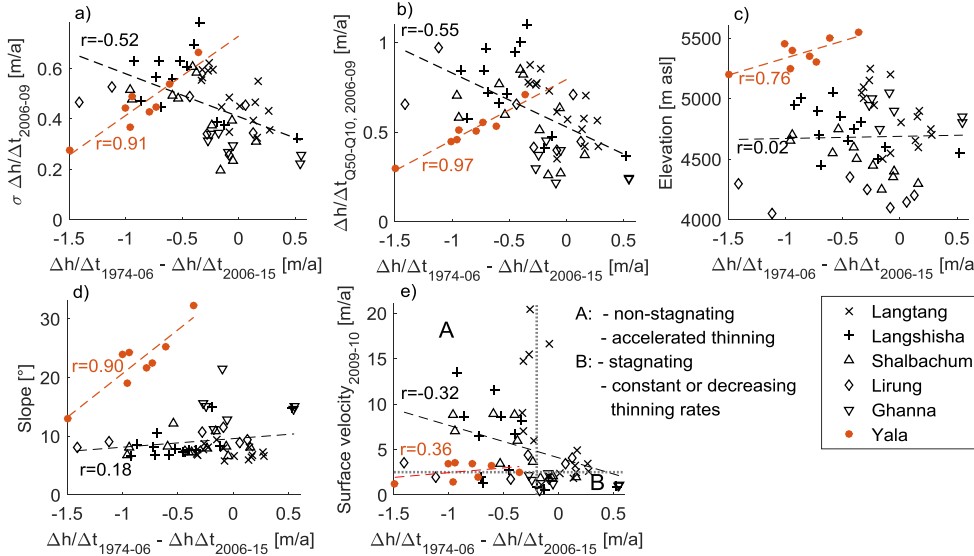

Figure 10: Changes in $\Delta h/\Delta t$ values 1974-2006 and 2006-2015 (a) vs. standard deviation of $\Delta h/\Delta t_{2006-09}$ pixels, (b) vs. difference between the 10th percentile (Q10) and the median (Q50) of $\Delta h/\Delta t_{2006-09}$ pixels, (c) vs. elevation, (d) vs. median slope, and (e) vs. median surface velocities (e). Each black marker corresponds to a debris-covered 50 m elevation band and red markers to debris-free 50 m elevation bands. Dashed lines represent linear trendlines and $r$ is the correlation coefficient.





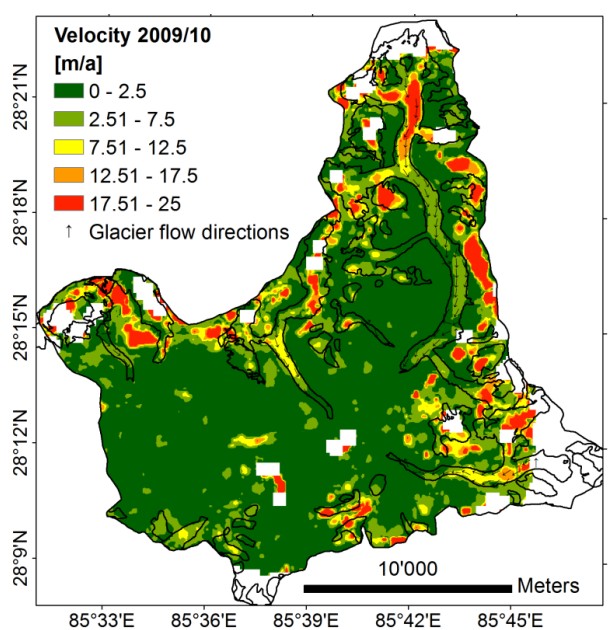

2  Figure 11: Surface velocities 2009-2010 cropped to catchment boundaries. Values have units

3  of *meters per year* and are derived by cross-correlation feature tracking. Black arrows indicate

4  derived glacier flow directions.





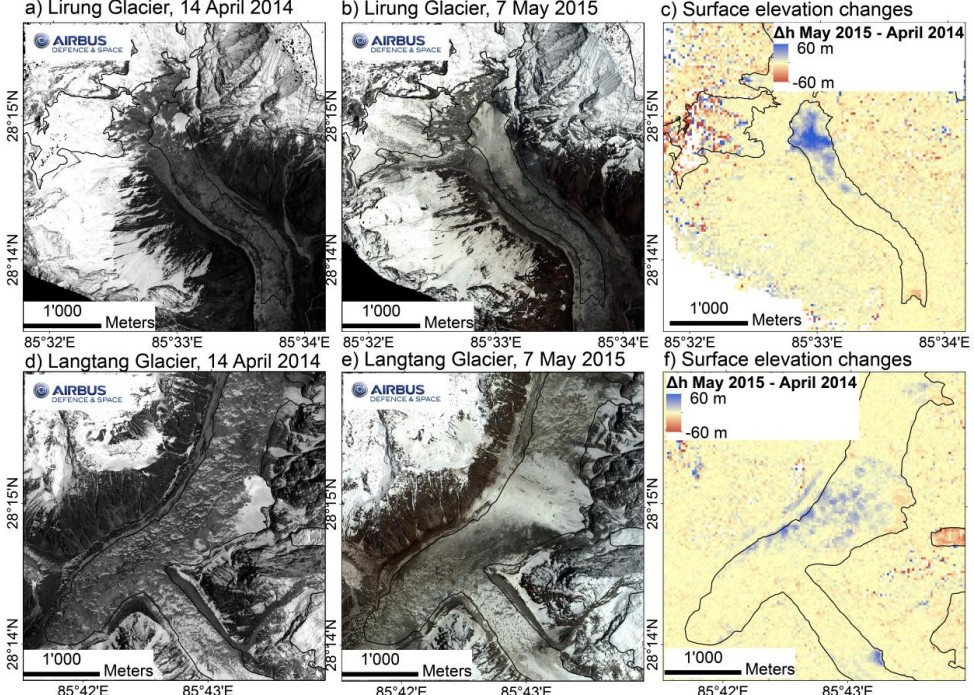

Figure 12. Avalanche affected sections of Lirung and Langtang glacier, pre- and after the earthquake on 25 April 2015, and corresponding surface elevation changes (Δh). Imagery ©Airbus DS 2014/2015.