# Peer review of "Heterogeneous glacier thinning patterns over the last"

_The Cryosphere, 2016_

## Referee Comment (RC1) · E. Magnusson (Referee) · 10 Mar 2016

General comments: In the paper Heterogeneous glacier thinning patterns over the last 40 years in Langtang Himal an interesting set of geodetic data from various sources is presented and used to infer the geodetic mass balance of the Langtang catchment in the Nepal part of the Himalayas focusing mostly on two periods (1974-2006 and 2006-2015). The authors undergo complex automatic classification of their data archive to sort out what they consider as reliable data for geodetic mass balance calculation as well as applying partly new approach to estimate the uncertainty. Unfortunately this work is not completed and is still some way from being scientifically sound. The main reasons for this are:

1) The logic behind the complex outlier removal is often difficult to understand, and

various steps in it are poorly justified. Despite this complex automatic outlier removal it seems to fail at many locations when looked at the difference maps of the 1974-2006 (Figure 6a). If the authors belief that the accumulation areas of glaciers thinned or thickened by 60-100 m at many locations as the this figure indicates they need to come up with some logical and justified explanation why (surges, enormous avalanches?) and they also need to explain the absence of this pattern of extreme thickening and thinning in the accumulation area of the glaciers for the period 2006-2015 (Figure 6b).

2) The explanation on how the uncertainty is calculated is not very clear and the procedure seems vague from statistical point of view. It is therefore hard to obtain any sense for its actual meaning. The authors do not even attempt to guess what confidence level it may represent. If more simple approaches such as using the e.g. standard deviation of off glacier DEM difference as proxy for the volume change uncertainty, it is at least known that such proxy is likely to result in very conservative uncertainty estimate compared to more advanced methods as shown by several studies.

3) The most critical weakness of this work is however that the authors seems neglect almost completely the uncertainty they actually obtain when discussing their results. A large proportion of the paper is spent on discussion on the temporal and spatial variation of the geodetic mass balance, while in most cases the variation they are discussing are not at all or barely significant if one believes the uncertainties obtained for the discussed values.

My main advices for the authors are the following:

a) Revise how you do your outlier removal, ideally make it more simple and if not make it such that the logic behind is understandable. It is also OK to use common sense when doing the outlier removal, instead of counting entirely on automatic outlier removal (this is presumably the difference between this work and the study of Pellicciotti et al. (2015) where part of the 1974 DEM of the accumulation area of Langshisha glacier was considered as erroneous data and therefore rejected).

b) Redo your uncertainty analysis. I would use approaches suggested by others unless you can better justify your approach and at least give the reader any evidence that the assumption you make when carrying out your uncertainty analysis is likely to result in an overestimate of your uncertainty rather than underestimate. You also need to be able to clarify what you mean by your uncertainty in terms of confidence level do give your uncertainty any meaning.

c) When the above has been done, carefully revise what your data actually tells you with any confidence. This could lead to a good concise paper if carried out in the above suggested manner.

Specific comments:

The list of the specific comments on the paper content here below should not be considered as complete, particularly regarding language, spelling, references etc., since in my opinion this manuscript and the work it describes needs almost a complete revision. The specific comments are mostly of two kind. Firstly, where I find reasoning of the methodology hard to understand or poorly justified. Secondly, where the authors are concluding much more from the data than they actually can, given the derived uncertainties (this is not a complete list, the remaining text free of such comments should also be critically revised, with this kept in mind).

Page 1, line 12: This first line does not tell the reader anything since glaciers are losing mass at very variable rate (even glaciers short distance apart).

Page 1, lines 18-19: The uncertainties here have large overlap. Assuming that the uncertainties where e.g. 95% confidence level (let alone lower confidence), you cannot state with great confidence that you show that the volume loss rate is higher now (even though it is more likely that it is, rather than the opposite).

Page 2, lines 8-10. Strange sentence, since you talk about examples of regional differences but only mention the upper limit values.

Page 2, line 15. Is "scientific debate" a good phrase to describe this, isn't the common goal of everyone studying this just to obtain answer to the same scientific questions?

Page 3, lines 7-17. Here the authors seem to give observations and models the same weight. When you have models on one hand and on the other hand conclusive observations, which don't fit the models, the reason for this is usually the incompleteness of the models, which in this case is probably the melting mechanism of the debris covered glacier.

Page 3, lines 15-17. I don't understand this sentence. What melt is caused by the glacier emergence velocity? Are you maybe referring to emergence of debris to the surface but not the classical glaciological term emergence velocity?

Page 5, lines 11-17. The author don't discuss at all the effects of seasonal changes on their geodetic results despite the fact that the DEMs (including the ones with most emphasizes, November 1974, October 2006 and February 2015) are from different time of the year. Can the seasonal effect be neglected? If so, based on what?

Page 6, lines 31-32. What about glacier motion, does your velocity data give any upper limit on what the motion of the GCPs could be within the time frame (if so state it)?

Page 8, line 1. Systematic errors in the glacier change map?

Page 8, line 4. Did Ragettli et al., (2015) do independent estimate on this or did they get the value from Sugiyama et al., 2013. If the latter Sugiyama et al., 2013 should be referenced for this. This ELA estimate, which presumably is just some average value for this catchment, is used in this paper to estimate accumulation area ratio (AAR) for each glacier. It is then repeatedly referred to in the paper like some actual observation of the AAR for the glaciers. It is not and given the unrealistically high variability of AAR in table 1 (15-86%) it is probably not even a good estimate for individual glaciers.

Page 9, line 1. I am not really following you here, when you mention the term automated flow accumulation process. Are you delineating ice divides between neighbouring ice

catchments? Is the big difference for Langshisha glacier between Pellicciotti et al., (2015) and this study caused by some part of Langshisha glacier as defined in the former study, being considered as separated ice catchment in this study? If so state this clearly. I would also recommend that you revise Figure 1 to better reveal the coverage of each glacier with improved background image behind it. By doing so you can (hopefully) convince the reader that your delineation of the glaciers is the more appropriate one.

Page 9, lines 2-5. This is a huge difference and is bound to have great effect on the result. You compere these two studies later on for this glacier, without even mentioning this important difference.

Page 9, line 12. Standard deviation of deltah/deltat at given point calculated for the up 28 difference maps or is this calculated over a given window?

Page 9, line 16. Well here is the answer to the question above. Personally I don't find this a good way writing, when something is only partly explained in a sentence and the same sentence and the following sentence does not indicate that further explanations will be given, but then later on the missing puzzle suddenly pups up. When I read such text, I am always asking myself "did I miss something?"

Page 10, lines 12-15. Here a justification why this should be errors but not actual elevation changes are completely missing. The span of elevation change rate over an entire glacier can easily be greater than the DEM errors but this depends on the time span, DEM quality, glacier type, etc.

Page 11, line 4. Outlier correction uncertainty? Do you maybe rather mean sensitivity to outlier removal?

Page 11, lines 5-17. This is very confusing text. I don't really understand what you are doing including why the thinning rate 2006-2015 is appropriate proxy for the outlier removal (of all data sets or just the 2006-2015 difference map?).

Page 11, line 18. DEM adjustment uncertainty? Is the term uncertainty appropriate here? I do not see that the parameter explained in this section is really used in your uncertainty assessment.

Page 11, line 29. I have problem obtaining the same results as the authors from this equation. If n=8 making Ndeltat=28 and k=3, I get

C_2= (8 over 3)/(2*28)=(8!/(3!*5!))/(2*28)=56/56=1, not 6 as authors say one should get.

Page 12, line 7-14. It took me quite a bit of time to actually understand what you are doing. I think I do now. Again I can't see what is logical about using the thinning rate from October 2006 to October 2015 as a threshold value. Can you explain that?

Page 12. Do I understand you right that the last outlier detection you do is the catchment scale outlier detection? Wouldn't be more appropriate to do that before you the do glacier scale outlier detection?

Page 12, lines 23-24. Here we are left with the question "how?" until half a page later. Again, this is not a good way of writing, it makes the paper hard to read.

Section 3.4.3. Here you come up with three outlier criterion. Why this complexity? It is not really justified in the paper.

Sections 3.4.2-3. It seems to me that the glacier catchment scale outlier removals are not likely to function appropriately when the time interval between DEMs is so variable and you do the outlier detection on deltah/deltat. deltat is ranging from < 1 year up to 32 year. This means e.g. for the last criteria that the DEM error for the 1974 DEM causing the 1974-2006 delath/deltat to be considered as an outlier would need to be 32 times larger than the error in 2009 DEM causing the 2009-2010 deltah/deltat to be considered an outlier. DEMs over short interval off course need to be very accurate to have informative value for volume change estimates, hence this is logical from that perspective. If my understanding of the outlier removal procedure is correct it does

however result in very weak outlier criterion for the 1974-2006 interval. If the authors rely entirely on this automatic outlier removal, it may result in erroneous result for this period, which to me, seems to be the case when looking at Figure 6 a. This is very unfortunate given that the main focus of your results and discussion is on the difference between the periods 1974-2006 and 2006-2015.

Page 14, line 5. I find the problem with your bias or trend correction approach manifest in this equation (I guess you are not the only one doing this). If this study had been only on one of these glaciers the data used for trend or bias correction would (presumably) only have been from the neighbouring area of this glacier resulting in MED~=0. But since you do the trend correction for the catchment as a whole (which I think is fine if you are studying the catchment but not individual glacier), MED~=0 is often not true for individual glacier, hence you will get different value for a given glacier than if you had focused the study only on that glacier. You are trying to compensate for this by adding this effect here into the uncertainty, but you are still left with the fact that the probabilistic mean of the actual average elevation change is likely not well represented by the centre of the given error bars. This becomes particularly awkward since your discussion of the results almost neglect the derived uncertainty limits and focuses on the centre of the error bars.

Page 14, lines 8-10. Are you saying that you use n=1? If so state it clearly, you could add to the sentence (i.e. n=1). Your usage of i.e. is not appropriate here (if I understand the sentence correctly). The fact that you use n=1 implies only that all pixels within the elevation band are fully dependent on one another (which truly is a conservative estimate). It does not however implies that there is no dependence between elevation bands. Since no attempts has really made to quantify the effect of the spatial correlation of your data (see e.g. Rolstad et al., 2009 or Magnússon et al., 2016, for further info) we don't really know if your assumption of no error compensation across elevation band is likely to lead to a conservative estimate of the uncertainty.

Section 3.6. It seems to me that your surface velocity could do with some more masking

of errors and outliers e.g. with correlation threshold. The masking that you are carrying leaves almost the entire velocity field intact as revealed by Figure 11 even though it is clear that much of it is just errors. The level of errors seen outside the glaciers is such that it is not clear if the signals on the glaciers are real or just errors as well. The figure itself is very hard view.

Page 15, line 9. Outlier and uncertainty assessment? Confusing. Wasn't this already done?

Section 4. This is all rather confusing. You calculate a lot of quality proxies used for outlier detection, mostly to convince yourself that the data that you derive your results from is of good quality. This is all good if one also reviews critically the outcome, which seem to be lacking in this study. A lot of these proxies are referred to as uncertainties apparently without being used to estimate uncertainty of the presented geodetic results. It is also not clear if all the DEM available during the period 2006-2015, apart from the initial and the final DEM, were really used to narrow down the uncertainty of volume change during this period. If not it seems to me that this paper would be much clearer if the focus of this paper were only on three DEMs, the ones from 1974, October 2006 and February 2015.

Page 18, line 18. Well if you think this is due to remaining systematic error, did you consider that your outlier removal is maybe not functioning so well?

Page 18, line 23-24. This is very true. Unfortunately you seem to forget it repeatedly in your discussion. Given that your uncertainties will be the same after revision of this work, much of the discussion on the results can be omitted because it is meaningless due to the large uncertainties.

Page 19, line 6. This is a good example of what I am talking about regarding the author neglecting the uncertainty in their discussion of the results. You cannot state here that the thinning rate increased by more than 100 %. If we know that John owns between 0 and 4 cars and Mike owns between 2 and 6 cars can you state that Mike owns at

least twice as many cars as John? No and the probability of such statement being true is only 14/25=0.56 (given even probability distributions for the car ownership in both cases).

Section 5.2.1. This comparison between debris covered and debris free glacier looking at "Explanatory variables" is rather primitive. For one thing it is rather inappropriate to refer to some of them as variables. I would rather refer to outcome of processes, which in some cases probably show correlation since they are dependent on the same physical variables. It is also strange that only Yala is included as candidate for the debris free glacier. The behaviour of Yala is then compared with 5 other debris covered glaciers. Even though the difference between Yala and each of the other 5 glacier is sometimes visually clear it is misleading to calculate the r-value for all the 5 debris covered glacier at ones and compare with a value calculated for a single glacier. When using data from several glaciers, various variables which effect the glaciers in different manner is bound to reduce the studied correlation compared to having data from just a single glacier.

Page 22, line 1. You mean April 2015.

Section 5.3. It is probably of interest for some to know the volume of these enormous avalanches. It however seems clear that avalanche falling on debris covered glacier (particularly the low insulated part of the glacier) is only going have minor and short last effect on the mass balance since it is going to melt much faster than the debris covered ice beneath it.

Page 24, line 25. What is numerical evidence?

Page 25, line 14. Is there no uncertainty in the area change?

Page 25, lines 16-17. I don't understand what you specifically mean by correlation between areal changes and surface elevation height.

Page 25, lines 23-26. Again not promising for your outlier removal, even though it is

better to admit it does not work to well. It would however be even better to justify why you think this is an error, e.g. by pointing out that local lowering of 60-100 m over 32 years (as indicated by figure 6a) on such small glacier at such high altitude is very unlikely to say the least.

Page 25, line 28-29. Even though it is likely that hypsometry plays a crucial role here this statement is far too bold given that it is based on very limited and apparently erroneous data (according to Figure 6a).

Page 25, line 31. See my previous comment regarding the AAR.

Page 26, line 26. Here and at other places in this paper, some temperature data (if available) would support your discussion.

Page 27, lines 16-18. You have far too little data with far too great uncertainty to make such statement.

Page 27, lines 20-23. You can say that Kimoshung glacier has higher hypsometry than Yala. The staggering difference between AAR values (which here are treated as some kind of truth but not as estimates based on the assumption of fixed ELA=5400 m a.s.l. for the whole catchment) is however misleading.

Page 28, lines 16-17. I am confused, is this in accordance with previous statement in this section (page 27, lines 16-18).

Page 28, lines 27-28. There is completely insignificant difference between these values. There is no point in trying to explain the "difference" between them.

Page 29, lines 6-10. I am very puzzled here. You need to justify here why this data is now suddenly considered as usable data, when the one processing the data rejected it in recently published paper. Why has he/she as the third author of this paper changed his/her mind?

Page 29, line 13. How can you state this? Does including apparently erroneous data

make the uncertainty estimate more realistic?

Page 29, lines 15-16. What other data did they use? They were hardly using GPS in 1982.

Page 30, lines 21-23. Sorry, I don't think many will agree on this statement.

Page 32, lines 5-9. This text does not fit into conclusion. If the authors think this text should be in the paper, it would be more appropriate to include it in the introduction.

Page 40, Table 1. See previous comments regarding the AAR.

Page 42, Table 5. It is not clear how the uncertainty of the average elevation change over the entire Langtang glacier catchment is calculated. Given its value it seems close to being basically (-)/(A*2), which basically corresponds to assuming that errors between glacier are completely dependent. Such assumption gives really conservative estimate, even too conservative causing the results to be downgraded. I also recommend that you stick to the same order of glaciers in the table as given in Table 7 with the glacier id.

Page 43, Table 7. Why no uncertainties? Are they within the digit of the given value, or did you simple not think about it? You are discussing these area changes in the paper without giving the reader any confirmation that these changes are significant.

Page 43, Figure 1. See my previous comments regarding this figure.

Page 44, Figure 2. The data on the debris covered glaciers is the most convincing part of this manuscript.

Page 45, Figure 3. It seems to me that using all the 6 proxies result in the same outlier removal as when you just use med2 and sigma2.

Page 46, Figure 4. 50% confidence level? What would the error bars be for a reasonably strict confidence level like 95%?

Page 47, Figure 5. Do you mean: a) A whiskers plot showing the distribution of uncertainties for the (up to?) 28 deltah/deltat maps. What do the red crosses indicate?

Page 47, Figure 6. See various previous comments on this figure. Should also be enlarged for better readability.

Page 48, Figure 7. The order of panels for the glaciers should be kept the same as the numbering of the glaciers in Table 7.

Page 50, Figure 9. Something went wrong with the altitudinal distribution for Yala.

Page 51, Figure 10. See previous comment regarding this figure.

Page 52, Figure 11. Very hard to read. The arrows are e.g. very hard to detect. Results do not appear very reliable (see previous comment).

References

Magnússon, E., Muñoz-Cobo Belart, J., Pálsson, F., Ágústsson, H., and Crochet, P.: Geodetic mass balance record with rigorous uncertainty estimates deduced from aerial photographs and lidar data – Case study from Drangajökull ice cap, NW Iceland, The Cryosphere, 10, 159-177, doi:10.5194/tc-10-159-2016, 2016.

Rolstad, C., Haug, T., and Denby, B.: Spatially integrated geodetic glacier mass balance and its uncertainty based on geostatistical analysis; application to the western Svartisen ice cap, Norway, J. Glaciol., 55, 666–680, 2009.

---

## Referee Comment (RC2) · Anonymous Referee #2 · 11 Mar 2016

General comments:

This paper presents glacier surface elevation change in Langtan Himal from 1974 to 2015 based on DEMs generated from satellite images. The authors analyzed temporal and spatial patterns of glacier thinning over the studied seven glaciers. Focuses of the discussion are spatial heterogeneity in the thinning rate, comparison of debris-covered and debris-free glaciers, changes in the thinning rate after 2006. The data are also used to quantify the impact of the earthquake in 2015.

Despite the increasing importance and interests on the Himalayan glaciers, long-term data on glacier changes are few in the region. Considering intensive research activities in the Langtang region in the past and recent periods, the presented data set is valuable. Nevertheless, uncertainty is rather large particularly in higher elevation areas.

This is very common in photogrammetric elevation analysis because snow covered surface loses surface features required for this method. Judging from the unrealistic thickening and thinning patterns in Figure 6a, it is questionable whether the DEM analysis is applicable in the accumulation areas. Moreover, estimated uncertainties are based on very complex outlier rejection criteria, which sometimes appear to be subjective and unconvincing. These problems result in limited reliability in the conclusions. Overall impression on the manuscript is that conclusions are too conclusive as compared to what are shown by the data.

I encourage the authors to thoroughly revise the manuscript (1) by using only reliable data, (2) with well focused objectives, (3) to draw only convincing conclusions. For example, omitting data from the accumulation reduces total uncertainties in Figure 7, which leads to more reliable discussion on recent increase in the thinning rate. Among others, elevation change over the debris-covered regions and impact of the earthquake are promising subjects.

Major concerns:

1. Reliability of the DEM in the accumulation area Figure 6a shows unusually large thickening and thinning patterns in the accumulation areas. The regions of the suspicious elevation change agree with the frequently snow covered regions shown in Figure 1. Most likely, photogrammetric analysis is hampered by featureless snow surfaces. Because such data from the accumulation areas are used for the mean thinning rate over each glacier, conclusions on the recent thinning acceleration and comparison between debris-covered and debris-free glaciers are unreliable.

2. Data and Method section The authors spend more than 1/3 of the manuscript for Data and Method section. This section is suffered from too much detailed explanations on how to reject outliers and estimate uncertainty. All details are given, but hard to understand the reasoning of each process. First, I suggest the author to move these details to the supplement, and describe in the main text only essence of the techniques

in an understandable way. Second, the structure of the section should be reconsidered. It can be something like, 3.1. Satellite data, 3.2. DEM (generation, differencing, processing, uncertainty), 3.3. delineation, 3.4. velocity.

3. Influence of the earthquake It is interesting and important to evaluate the impact of the earthquake on the glacier surface elevation. However, the elevation change due to the earthquake in 2015 is essentially different from those occurred from 1974 to 2014. Accordingly, elevation change from 1974 to 2015 (e.g. Table 5 and Figure 6b) is not suitable to discuss recent glacier changes in general. Therefore, I suggest the author to separate the elevation change after the earthquake from the rest of the study period.

4. Text I understand that the author tried to be careful and accurate in the text. However, the manuscript is lengthy, redundant and diffuse at many places. This hinders reader's understanding of the methodology, important results and conclusions. Please consider to shorten and simplifies sentences throughout the manuscript.

Specific comments:

page 1, line 15: we present volume and mass changes of . . . (omit "glacier")

page 1, line 22: "mass balance trends" sounds to me "surface mass balance trends". What about "mass loss trends" or "thinning trends"?

page 1, line 22: "highly non-linear" to what? elevation? time?

page 3, line 4: What do you mean by "downslope condition"?

page 3, line 8: . . . present-day "surface" lowering rates. . .

page 3, line 16: What is "melt due to glacier emergence velocity"?

page 4, line 26: . . ., Kimoshung Glaciers. . .

page 4, line 31-32: Please consider to shorten this kind of sentences. It should be OK to write ". . . are exceeded most part of the debris-covered area (Ragettle et al., 2015).

Relatively thin debris layer appears only near the equilibrium line."

page 5, line 6: a.s.l.

page 7, line 9: ALOS PRISM

page 8, line 5: What is "correlation score"?

page 8, line 9: Either of "older" or "earlier acquisition date" is fine.

page 9, line 12-13: I understand that these parameters are useful to measure spatial non-uniformity in the melt rate. However, I do not understand why you use both of them. Particularly, the second one needs a reason why you take 50% and 10%. Moreover, why not using the information on cliffs and lakes delineated from the satellite image (Figure 2b)?

page 10, line 16: I wonder why "higher accuracy" can be the reason to apply the higher threshold.

page 11, line 11: Why do you use the thinning rate from 2006 to 2015 as a threshold?

page 11, line 18-page 12, line 18: It is hard to understand the concept and the procedure to obtain U_adj. If this is a commonly used parameter, please provide a good reference. I recommend the author to describe this kind of details in supplement.

page 13, line 12-13: I wonder how these thresholds were chosen and why they "effectively minimize the uncertainty".

page 14, line 5: Using three characters as a symbol is not common. By the way, do you need to define this symbol "unc"?

page 14, line 14: Do you use the same density in the accumulation area?

page 15, line 12: Should be "92 maps were removed because they FULFIL outlier criteria"?

page 15, line 22: Define the acronym "RPC".

page 17, line 5: Please be consistent with the unit, m/a or m a-1.

page 17, line 29-30: "but a majority of values suggest that . . .." » This is not very sure from the data. It appears to me that the thinning rate is decreasing recently

page 18, line 1: What do you mean by "ensemble of values"?

page 18, line 30: What about simplifies the sentence to "The most negative elevation change for 1974-2006 was observed at Shalbachum . . ...".

page 19: line 2-3: It makes more sense to compare 1974-2006 and 2006-2014 to eliminate the influence of the earthquake.

page 19, line 1: m a-1 » You need a space between m and a-1.

page 19, line 9: "The most important differences in mean dh/dt values. . ." » "The greatest increase in thinning rate . . ."?

page 19, line 15-21: I find this paragraph is not necessary here. Because Figure 8 clearly shows the thinning patters, you do not need to give questionable comment on Figure 6.

page 19, line 32-page 20, line 3: This sentence is very hard to read. Please consider to rewrite it.

page 21, line 30: "dh/dt_1974-06-dh/dt_2006-15<-0.2 m/a" » Is this correct? Isn't the left side positive if the thinning is accelerated?

page 22, 5.3. Impacts of the April 2014 earthquake: This is an interesting analysis. I suggest the author to use the DEM after the earthquake only for this purpose. In other words, elevation change from 1974 to 2014 should be used for the rest part of the discussion.

page 22, line 2-8: This should be explained in the introduction section.

page 22, line8-21: This should be explained in the method section.

page 23, line 5: "compensated by about 50%" » What density do you assume for the avalanche debris deposition?

page 23, line 12: "Elevation changes in the debris-covered area are primarily independent of elevation (Figures 8 and 10c) as previously identified in Langtang catchment (Pellicciotti et al., 2015) and elsewhere . . .."

page 23, line 16: "downward-" » downglacier?

page 24, line 1-2: Not clear where and how water pressure is elevated.

page 24, line 5-6: Do you mean that thinning accelerated where ice motion is active because cliffs and lakes develops? It contradicts to my experience to observe cliffs and lakes formation on debris-covered stagnant ice.

page 24, line 20: "glacier uplift" » do you mean "ice thickening due to compressive flow regime"?

page 23, line 27-page 25, line 11: The goal of this section is not clear. It appears that this section discusses the mechanism of surface elevation change on debris-covered ice. However, the thinning rate is highly variable in space and time, and there is no general trend in the observed glaciers. What kind of results does the author try to explain here? Many processes related to surface elevation change of debris-covered glaciers are described, but none of them are connected to reliable interpretation of the data. Describe first an observational fact that you want to discuss, and interpret the observation in a logical manner.

page 25, line 16-17: What do you mean by "correlate with"? Which data show this?

page 25, line 13-page 26, line 9: The first part of this section 7.1.1. explains that thinning accelerated at Yala Glacier, whereas it appears to be at a similar level at Kimoshung Glacier. This kind of explanation should be completed in Result section. Interpretation on the difference begins at

page 25, line 28, but not convincing because there is no qualitative discussion. For example, hypsometry is not shown for Kimoshung Glacier, and no information about the 0 degree C isotherm altitude.

page 26, line 11-page 27, line 11: This section has the same problem as section 7.1.1. The first paragraph describes several different observations. These details should be explained in Result section, and here the focus of the discussion should be stated briefly. In the second paragraph, speculative conclusions are given without detailed/quantitative comparison with the modeling work.

page 27, line 18-20: Not clear what are compared in Figure 6b.

page 27, line 24-25: Not clear why you compare the elevation of Yala terminus and that of maximum thinning on Langtang.

page 27, line 26-28: It is not clear which part of the elevation range is compared here. If you discuss elevation change of debris-covered and debris-free glaciers at same elevation range, why not preparing a plot for this purpose?

page 28, line 3-15: The point of the discussion is unclear.

page 28, line 16-17: Not clear what you mean. Do you mean that your result support the studies by Kaab, Nuimura and Gardelle?

page 30, line 17-page 32, line 14: Only a few data appear in Conclusion section, which results in very qualitative descriptions. This represents the weakness of the paper. Please draw your conclusions which are supported by data.

page 52, Figure 11: This velocity map is not much used for the study. Judging from the vectors on the plot, it is not sure how much this analysis is reliable.

---

## Editor Comment (EC1) · E. Berthier (Editor) · 4 Apr 2016

Dear authors,

The on-line discussion of your paper 'tc-2016-25' is now closed. Your TCD manuscript has been thoroughly review by two external referees and I want first to acknowledge them for the time they spent on evaluating comprehensively your study.

Both referees raised some important issues on the methodology (outlier exclusion) and the significance of your results. In particular, they converge on the fact that most of the differences discussed are not significant considering the error bars. This is well summarized by the statement of reviewer#1 that "the authors seems neglect almost completely the uncertainty they actually obtain when discussing their results" and reviewer #2 "Overall impression on the manuscript is that conclusions are too conclusive

as compared to what are shown by the data".

In this context, I ask you to answer point by point to the general comments of both referees and explain how you would proceed if you were asked to provide a revised manuscript. Then, I will decide whether we will move forward with the review process of this study.

Best regards,

Etienne Berthier

---

## Author Comment (AC1) · 7 May 2016

**REVISION STATEMENT**

**"Heterogeneous glacier thinning patterns over the last 40 years in Langtang Himal"**

**by S. Ragettli, T. Bolch and F. Pelicciotti**

**GENERAL REVISIONS**

We would like to thank very much the two reviewers for their thorough and detailed comments. We have addressed all the reviewers concerns in our detailed point by point answers below. Both referees raised important methodological issues, mostly regarding outlier exclusion and uncertainty quantification. We agree that a revision of our methods was necessary in order to draw convincing conclusions. We have thus substantially modified some of our procedures and we have produced new figures (provided at the end of this document).

The major issues in the revision were: 1) to revise the outlier removal procedure to make it simpler and based on logical criteria (reviewer 1 and 2), 2) to increase the reliability of the data by addressing the unusually large thickening and thinning patterns in the accumulation areas (reviewers 1 and 2), 3) to redo the uncertainty analysis based on approaches that are more established in the literature (reviewer 1), 4) to improve the manuscript by reducing its length, make it more focused and by drawing only convincing conclusions based on reliable data (reviewers 1 and 2), 5) to separate the elevation changes after the earthquake from the rest of the study period (reviewer 2).

As a result of changes in response to the reviewers' comments, these are the main changes in our study:

1. We now use a simpler and more straightforward procedure for the selection of data for our ensemble approach.
2. We have revised our approach for outlier removal at the grid scale. The approach does not fail anymore to identify erroneous patterns in the accumulation areas of glaciers.
3. We use a new approach for uncertainty estimation based on Gardelle et al. (2013).
4a. We use a new dataset of supraglacial cliff and lake inventories to explain the observed thinning patterns instead of the proxies previously used in the submitted manuscript; these new data sets allow to directly relate spatial thickness change patterns to observations of glacier surface characteristics.
4b. We increased the readability and sharpened the focus of the manuscript: the 'Methods' section will be substantially shortened due to the use of simpler and more established procedures which require less explanations. The separate section on outliers and uncertainty will be removed.
4c. Generally, figures and text will better emphasize the added value of using an ensemble approach, which we are convinced is a novelty of this paper but has not been recognized (or only partly) by the reviewers. The availability of multiple independent DEM differencing results for overlapping periods allows identifying a sound signal and narrowing down the uncertainty of recent volume changes.
5. By the revised Figure 8 we show that over longer periods ($\Delta t > 4$ years) the effect of the post-earthquake avalanches six months after the earthquake (in April 2015) is negligible compared to the uncertainties in calculated elevation changes. The October 2015 DEM is thus still used to assess long term elevation changes, whereas the May 2015 DEM is not.

We are convinced that the changes in methodology and structure of the paper have benefited the papers' quality. They have also strengthened the novelty and relevance of the results. We are now able

to unambiguously identify at which glaciers overall thinning has accelerated in recent periods, or where thinning has remained approximately constant. The new dataset on cliff and lake area that reviewer 2 has suggested to include provides valuable insights regarding the mechanisms that lead to spatially heterogeneous thinning patterns on debris-covered glaciers. The more reliable results for the accumulation areas allow now for more convincing conclusions regarding the differences in glacier response to climate. None of our main results, figures or conclusions however changed significantly, thus supporting the methodological choices made originally. Overall, we strongly believe that our study now indeed represents one of the most rigorous documentations to date of glacier response to climate change over the last 40 years in the Himalaya. We therefore hope that the editor allows us to provide a revised manuscript and decides to proceed with the review process.

Since some of the comments were common to both reviewers, we describe below the major changes made and refer to those in our detailed responses to the reviewers.

We also include at the end of this document all the revised figures that will be included in the revised manuscript. In this way, the editor and reviewers can appreciate the changes done in response to the reviewers' comments. The figures show the new results as obtained following the reviewers' suggestions. We refer to them in our responses to reviewers.

**NEW METHODS**

**1. Data selection** (previously called 'Outlier detection at the catchment and glacier scale', Sections 3.4.2 and 3.4.3):  Instead of selecting the data based on four different data quality proxies (% data available after outlier correction at the grid scale, sensitivity to outlier correction, triangulation residuals, and mean off-glacier elevation differences) we will simply select all $\Delta h/\Delta t$ maps from the period 2006-2015 that cover periods of four years or longer. Shorter periods will be discarded and not discussed in the paper, except for assessing the short-term effect of post-earthquake avalanches. This change responds to the reviewers comments about simplifying our methods, which we agree with. There are two main reasons to discard short periods and focus only on multi-annual periods within 2006-2015 instead:

- Uncertainties substantially decrease for longer periods (see Figure 3 below, which will be a new paper figure).
- Data from overlapping periods within 2006-2015 provide a range of plausible values for this period, especially as the elevation changes between the different selected periods show very similar characteristics (Figures 8 and 9). This data ensemble allows narrowing down the uncertainty: the uncertainty in a sample mean is lower than the uncertainty in individual estimates (according to standard principles of error propagation). We thus obtain an ensemble of results for the relatively short period 2006-2015 that can be used to assess changes in thinning rates with respect to the longer period 1974-2006 in which much larger absolute elevation changes occurred.

The ALOS 2010 scene will be excluded from the ensemble since we can show that the uncertainties of the corresponding $\Delta h/\Delta t$ maps are 20-80% higher than the uncertainties calculated for other maps (we are referring here to the new uncertainties calculated with the approach based on Gardelle et al. 2013, see point 3. below). The post-earthquake scene from May 2015 will also be excluded from the 2006-2015 ensemble since elevation changes are essentially different due to the post-earthquake avalanches. The October 2015 DEM will be still considered for the ensemble, since a larger part of the avalanche deposits melted already and, hence, the avalanche effect on multi-annual glacier volume changes is

minor in comparison to the ensemble uncertainty (see our detailed answer to the third major comment by reviewer 2).

**2. Outlier correction** (revision of methods presented in 'Outlier detection at the grid scale', Section 3.4.1): We revised our outlier correction by narrowing the range of acceptable values for the *accumulation* areas and now use a 1σ threshold to identify outliers. Pixels are thus defined as outliers when the absolute elevation differences differ by more than one standard deviation (considering all elevation differences within glacier area located above the ELA). Below the ELA we now use a 3σ level (instead of a 2σ level for debris-free terrain as in the original manuscript), following Gardelle et al. (2013). We are aware that ELA estimates are uncertain (see our response to the comment on Page 8, line 4, by reviewer 1), and thus we will assess the sensitivity of our results to an ELA uncertainty of ±100 m.

The application of more restrictive criteria for plausible elevation change values in the accumulation areas required also a revised procedure for gap filling, because gaps tend to be quite large when using 1-sigma thresholds. We now first calculate the mean elevation change rates per 100-m elevation band of each glacier and then calculate the median of the ensemble (see revised Figure 9). This value is then used to replace outliers from a given elevation band in the accumulation area. For the ablation areas we still use inverse distance weighting (IDW) for gap filling, since gaps are very small and the variability in plausible values is high. This procedure led to more realistic values especially in the accumulation areas of the period 1974-2006. Previous geodetic studies have used glaciological expert knowledge  for outlier removal and gap filling in the accumulation areas (e.g. Pieczonka et al., 2013; Pieczonka and Bolch, 2015), considering that elevation changes in the accumulation areas are minor over periods of several years (e.g. Schwitter and Raymond, 1993; Huss et al., 2010). Since we now only consider time intervals between DEMs that are longer than 4 years we think it is justified to use empirical values from the same glacier to fill data gaps in the accumulation areas, even if data from very different periods are used to calculate ensemble-median values.

The revised outlier correction now detects obviously erroneous data in the accumulation areas of the 1974-2006 map (see revised Figure 6a).

**3. Uncertainty quantification** (Section 3.5): Our new uncertainty estimates are based on the standard error calculated per elevation band as in Gardelle et al. (2013). Accordingly, we now take into account the number of independent pixels per elevation band. The distance of spatial autocorrelation for each Δh/Δt map is calculated considering the range of the semivariogramm of all off-glacier elevation differences (e.g Magnússon et al., 2016). We identified distances between 350 and 600 m (average of the ensemble: 493 m). Weighted mean uncertainty values per glacier are then calculated as in the original manuscript by taking into account the altitudinal distribution of uncertainty and glacier hypsometry.

Our previous approach took into account both the mean elevation differences (MED) and the standard error (SE). The large uncertainties obtained for the 1974-2006 map were related to the erroneous elevation change patterns that were due to errors in the Hexagon 1974 DEM. However, both reviewers suggested discarding all unrealistic elevation changes in the accumulation areas. To account for the MED is therefore not necessary anymore, since deviations from zero are prevented by the more restrictive outlier definitions. Accordingly, the uncertainty estimates for the 1974-2006 map are now much lower (revised Figure 7). This facilitates the interpretation of results shown by the revised Figure 7 and allows for stronger conclusions regarding the differences between 1974-2006 and 2006-2015.

Ensemble-median and ensemble-uncertainty values will substitute the values provided in Table 5 for the period 2006-2015.  'Ensemble uncertainty' is defined as the standard deviation in observations

available for 2006- 2015 ($\Delta h/\Delta t$ maps with $\Delta t \geq 4$ years, see explanations above) multiplied by 1.96. Standard deviation is commonly interpreted as 68% confidence level assuming normal error distribution. By multiplication with 1.96 we obtain 95% confidence levels.

**4. Explanatory variables** (Sections 3.3 and 5.2.1): We now use six quality checked maps of cliffs and lakes from each available satellite image for the period 2006-2015 (2006, 2009, 2010, 2014, May and October 2015). These inventories are used to calculate cliff and lake area per elevation band, and replace the statistical proxies ($\sigma \Delta h/\Delta t$, $\Delta h/\Delta t$ Q50-Q10). Since both reviewers criticized that only limited conclusions are possible from the original Figure 10, we have replaced this figure by elevation profiles showing ensemble-median thinning rate changes ($\Delta \Delta h/\Delta t$), surface velocities and lake/cliff area (in % area per elevation band).

The new dataset allows for interesting and convincing conclusions regarding the mechanisms that lead to spatially heterogeneous thinning patterns on debris-covered glaciers. Correlations between the presence of cliffs and accelerations in local thinning are evident from the revised Figure 10. The correlation coefficient between mean cliff area per tongue and changes in mean thinning rates per tongue ($\Delta \Delta h/\Delta t$) is -0.77, which strongly suggests that thinning accelerations are related to the presence of cliffs.

Details regarding the cliff/lake delineation procedure, quality check, the calculations of cliff/lake area per elevation band and a detailed discussion of the new Figure 10 will be provided in a revised manuscript.

**RESPONSE TO REVIEWER 1**

**General comments: In the paper Heterogeneous glacier thinning patterns over the last 40 years in Langtang Himal an interesting set of geodetic data from various sources is presented and used to infer the geodetic mass balance of the Langtang catchment in the Nepal part of the Himalayas focusing mostly on two periods (1974-2006 and 2006-2015). The authors undergo complex automatic classification of their data archive to sort out what they consider as reliable data for geodetic mass balance calculation as well as applying partly new approach to estimate the uncertainty. Unfortunately this work is not completed and is still some way from being scientifically sound.**

We would like to thank E. Magnusson for his very useful review. His suggestions for improvement of our outlier removal and uncertainty estimation procedures helped us to considerably increase the robustness of our conclusions. We believe that our results are now scientifically sound, in the sense that now only reliable data are used and that we can show that identified variations in glacier thinning are significant. The revision of the methods led to a more robust statistical assessment of our main results.

However, we also would like to note that the changes in outlier removal and uncertainty estimation procedures did not lead to different conclusions as in the original manuscript. As in the original manuscript, our results depict a heterogeneous response of glaciers to climate, with a strong spatio-temporal variability of thinning trends at debris-covered tongues and clear evidence about the crucial role of glacier hypsometry for mean thinning trends. We also would like to note that the reviewer ignored some key novel aspects of our work, such as using an ensemble of independent observations to narrow down the uncertainty of our results. We were puzzled by his proposition in one of his detailed comments below on Section 4 to use less data (only three DEMs). To us it is obvious that several independent measurements (differential DEMs for the period 2006-2015) lead to higher confidence in detected signals. To make this clearer we will provide ensemble-median and ensemble-uncertainty values, as described at the beginning of this revision statement under 'New Methods'. Overall, we are however thankful for all the reviewers' detailed comments since they helped us to understand where the methods needed to be improved and where the advantages of a given approach needed to be clarified.

**1) The logic behind the complex outlier removal is often difficult to understand, and various steps in it are poorly justified. Despite this complex automatic outlier removal it seems to fail at many locations when looked at the difference maps of the 1974-2006 (Figure 6a). If the authors belief that the accumulation areas of glaciers thinned or thickened by 60-100 m at many locations as the this figure indicates they need to come up with some logical and justified explanation why (surges, enormous avalanches?) and they also need to explain the absence of this pattern of extreme thickening and thinning in the accumulation area of the glaciers for the period 2006-2015 (Figure 6b).**

We agree with the reviewer that the outlier removal procedure used in the paper is complex and the presentation of the method takes substantial manuscript space. We also agree that the threshold criteria are sometimes difficult to justify, but this is because perfectly objective criteria are not available.

Regarding the 1974-2006 elevation difference map it is true that outliers in the accumulation areas remained and therefore the outlier removal procedure failed for these areas. We do not believe that thinning or thickening values of 60-100 m are plausible in the accumulation areas. This was stated clearly in the manuscript (P. 17, lines 10-12: "*The Nov 1974 - Oct 2006 Δh/Δt map (Figure 6a) reveals an irregular and unrealistic distribution of Δh/Δt values at high altitudes, which can be likely*

*associated to errors in the Hexagon 1974 DEM*", P. 25, lines 25-26: "*Presumably, unrealistically high thinning rates at high altitudes due to errors in the Hexagon 1974 DEM led to this result*"). It is therefore not necessary to explain the absence of these patterns in other maps, since we clearly say that those unrealistic values are due to errors in the Hexagon map and thus the absence of errors in the other maps is due to better data quality.

However, we have carefully revised our outlier removal procedure to address the reviewers' concern and following his suggestions. The revised outlier detection algorithm now identifies unrealistic patterns in the accumulation areas. Missing data in the accumulation areas are replaced by plausible values. All new methods are described above under 'New Methods' and 'Outlier correction'.

The procedure for the selection of an ensemble of maps from the 28 available elevation change maps has been substantially simplified (see 'New Methods' and 'Data selection' above), even though we retain the ensemble as we think this approach provides a valuable estimate of uncertainty and sounder signal. The main focus of the paper will be, as suggested by this reviewer, on the comparison of the periods 1974-2006 and 2006-2015, and for the second period we consider a number of overlapping periods that allow narrowing down the uncertainty of volume change during this period.

**2) The explanation on how the uncertainty is calculated is not very clear and the procedure seems vague from statistical point of view. It is therefore hard to obtain any sense for its actual meaning. The authors do not even attempt to guess what confidence level it may represent. If more simple approaches such as using the e.g. standard deviation of off glacier DEM difference as proxy for the volume change uncertainty, it is at least known that such proxy is likely to result in very conservative uncertainty estimate compared to more advanced methods as shown by several studies.**

Our uncertainty calculations were based on the approach used in Bolch et al. (2011), a published and established approach (used e.g. in Thompson et al., 2016, JG). We summed quadratically the mean off-glacier elevation differences (MED) and the standard error (SE) (eq. 5). We noticed that the uncertainties increase with altitude (which is common for geodetic elevation changes in the Himalaya, (e.g. Nuimura et al., 2011). This is partly due to the fact that higher elevations tend to have steeper slopes and it is well know that the accuracy of DEMs derived from stereo data decreases with increasing slope. Therefore, we first calculated the uncertainty for each elevation band independently and then calculated a weighted average per glacier by taking into account glacier hypsometry. This resulted in conservative uncertainty estimates (i.e. large uncertainties) since both the mean error and the standard deviation were taken into account and we assumed no error compensation across elevation bands. However, when revising our approach we identified the following issues:

- The spatial autocorrelation of the error was insufficiently accounted for by considering *n* in eq. 4 equal to the number of pixels per elevation band (as explained on P. 14, lines 6-13) and not the number of independent pixels per elevation band (such as in Gardelle et al. 2013).
- Summing quadratically MED and SE led to very high uncertainty estimates at high altitudes, especially where the Hexagon 1974 DEM was used (because of errors in this DEM over snow-covered surfaces). This led to uncertainty ranges which suggested that even positive glacier mass balances are possible for debris-covered glaciers such as Langtang or Langshisha, although at the glacier tongues we unambiguously identified strong surface lowering (Figure 8 and 9 of the manuscript)

Following the reviewer's remark, we therefore used a different approach for uncertainty estimation. The new uncertainty estimates are based on the standard error calculated per elevation band as in

Gardelle et al. (2013). Detailed explanations regarding the new method are provided above under 'New Methods' and 'Uncertainty quantification'.

To use just a crude proxy such as the standard deviation of the off-glacier elevation differences, however, seems not appropriate. This would imply assuming that the DEM errors at all locations are totally correlated, which we know is not the case. The standard error (thus the standard deviation of the sample-mean's estimate of the error) can be interpreted as 68% confidence level assuming normal distribution. Since we are assuming no error compensation across elevation bands the confidence level in our uncertainty estimates per glacier is higher than 68%. This will be stated in the revised manuscript.

**3) The most critical weakness of this work is however that the authors seems neglect almost completely the uncertainty they actually obtain when discussing their results. A large proportion of the paper is spent on discussion on the temporal and spatial variation of the geodetic mass balance, while in most cases the variation they are discussing are not at all or barely significant if one believes the uncertainties obtained for the discussed values.**

We kindly disagree with the reviewer here. There are two main reasons for this. First, uncertainties and outliers are discussed in detail in the manuscript. Indeed, we devote a whole section to this (Section 4). In the Results section (sections 5) the uncertainties are always provided when presenting mass balance values or mean thinning rates. If necessary, uncertainties are addressed explicitly when discussing results (e.g. P. 18, lines 3-5; P. 18, lines 23-25; P. 20, lines 5-12; P. 21, lines 21-23; P. 25, lines 23-27).

Second, the reviewer does not take into account that we use an ensemble of independent measurements, which allows constraining the uncertainty of individual Δh/Δt maps. This was stated in the discussion section, although we agree that it needs to be stated more clearly (e.g. at page 18, lines 1-3, *"The ensemble of values helps to distinguish between trends that should be classified as uncertain… from trends that are consistent within the ensemble)"*. To us it was clear that the ensemble of values, available for overlapping periods, is an asset of this study. Now we understand that the advantages of the ensemble approach might not have been clear and need to be stressed more. We do this in the revised Figure 7 and we will emphasize this point in the text of the revised manuscript.

We also point here to the fact that our results regarding the spatial and temporal variations in elevation change rates in the ablation areas are affected by very low uncertainties. Consequently, a large part of the Results and Discussion sections are devoted to discuss the spatio-temporal patterns in the ablation areas. We are surprised that the reviewer does not mention this here and instead suggests that most our results are affected by high uncertainty, although he agrees that "the data on the debris covered glaciers is the most convincing part of this manuscript." (see his comment below about Page 44, Figure 2.).

**My main advices for the authors are the following:**

**a) Revise how you do your outlier removal, ideally make it more simple and if not make it such that the logic behind is understandable. It is also OK to use common sense when doing the outlier removal, instead of counting entirely on automatic outlier removal (this is presumably the difference between this work and the study of Pelicciotti et al. (2015) where part of the 1974 DEM of the accumulation area of Langshisha glacier was considered as erroneous data and therefore rejected).**

We have simplified the outlier removal, especially regarding the selection of maps for the ensemble which was based on four different data quality proxies. In the revised paper, we will simply select only $\Delta h/\Delta t$ maps from the period 2006-2015 that cover periods of four years or longer. Detailed explanations are provided above under 'New Methods' and 'Data selection'.

Regarding the grid scale outlier correction it is necessary to define clear criteria to prevent arbitrary or subjective choices. Note that also in Pellicciotti et al. (2015) outlier detection was based on an automatic algorithm, but in the case of Langshisha Glacier the threshold of acceptable elevation changes was lower (see our response below to the reviewers' comment on Page 29, lines 6-10). Accordingly, we revised our outlier correction by narrowing the range of acceptable values for the accumulation areas. Detailed explanations are provided above under 'New Methods' and 'Outlier correction'.

**b) Redo your uncertainty analysis. I would use approaches suggested by others unless you can better justify your approach and at least give the reader any evidence that the assumption you make when carrying out your uncertainty analysis is likely to result in an overestimate of your uncertainty rather than underestimate. You also need to be able to clarify what you mean by your uncertainty in terms of confidence level do give your uncertainty any meaning.**

We have followed the reviewer's advice and now use an approach that is more established in the literature (see detailed explanations above under 'New Methods' and 'Outlier correction'). We will state in the revised manuscript that the estimated confidence level of our uncertainty values is higher than 68% (see our answer to the reviewers' second main point above).

Finally, we will also provide the ensemble uncertainties (procedure described as well above under 'New Methods' and 'Outlier correction'). The variability in the ensemble of values extracted for overlapping periods is a better indicator for the actual uncertainty in the values identified for the period 2006-2015.

**c) When the above has been done, carefully revise what your data actually tells you with any confidence. This could lead to a good concise paper if carried out in the above suggested manner.**

We have done all of the above in terms of methodology, and have revised the paper accordingly. With the improvements in our procedures for outlier correction and uncertainty analysis it is possible to clearly identify changes in mean thinning rates over time (see revised Figure 7). The $\Delta h/\Delta t$ glacier profiles (Figures 8 and 9) allow identifying unambiguously where in the ablation areas thinning has accelerated.

While it is true that the paper is more concise now as a result of the simplifications suggested by the reviewer, our main results however have not changed.

In addition, with a new dataset of cliff and lake areas (see our answers to the second reviewer) it is possible to directly relate spatial patterns of change to glacier surface characteristics (see the revised Figure 10).

Indeed, we think the suggestions by the two reviewers have helped us to present more concise and interesting results.

**Specific comments:**

**The list of the specific comments on the paper content here below should not be considered as complete, particularly regarding language, spelling, references etc., since in my opinion this manuscript and the work it describes needs almost a complete revision. The specific comments are mostly of two kind. Firstly, where I find reasoning of the methodology hard to understand or poorly justified. Secondly, where the authors are concluding much more from the data than they actually can, given the derived uncertainties (this is not a complete list, the remaining text free of such comments should also be critically revised, with this kept in mind).**

We thank E. Magnusson for his detailed comments. As stated above, we have revised the methodology. Regarding the uncertainties, the advantage of using an ensemble DEMs to constrain uncertainty will be better emphasized in the text. We do not agree that we concluded more than allowed from the data, for the reasons summarized in the general response.

The remaining text free of comments will also be revised. We will make sure the sentences are clear and that the methods are well explained. In our detailed answers below we provide indications regarding which sections will be removed to streamline the text and to increase the readability of the paper.

**Page 1, line 12: This first line does not tell the reader anything since glaciers are losing mass at very variable rate (even glaciers short distance apart).**

The reviewer is right that the mass loss rates of individual glaciers are variable. However, we are referring to regional trends here. It is true that most Himalayan glaciers are losing mass at rates similar to glaciers elsewhere (Bolch et al. 2012).

**Page 1, lines 18-19: The uncertainties here have large overlap. Assuming that the uncertainties where e.g. 95% confidence level (let alone lower confidence), you cannot state with great confidence that you show that the volume loss rate is higher now (even though it is more likely that it is, rather than the opposite).**

We agree that the sentence needed clarification. In the revised manuscript we will emphasize the ensemble of independent values available for the period 2006-2015, which allows constraining uncertainty. The new uncertainty estimates based on the standard error (Gardelle et al., 2013) yield lower uncertainties for the period 1974-2006. It is therefore possible to state now with great confidence that volume loss rates are higher. We will replace the sentence with the following text:

*"The ensemble of DEMs available for the period 2006-2015 allows detecting a clear signal of and constraining uncertainty about recent glacier volume loss and identifying a strong acceleration in thinning rates in comparison to the period 1974-2006. Considering six overlapping periods within 2006-2015 we calculate a mean thinning rate of -0.46 m a$^{-1}$ and an ensemble uncertainty of ± 0.18 m a$^{-1}$, while for the period 1974-2006 we identify a thinning rate of -0.25 ± 0.08 m a$^{-1}$."*

Note that the uncertainty bounds provided above are still overlapping at the ends. In the revised manuscript we will thus quantify the confidence level in our statement that thinning rates have accelerated. Expressing ensemble uncertainties with a confidence level of 90% instead of 95% (standard deviation multiplied by 1.65 instead of 1.96, assuming normal distributions) results in practically non-overlapping error bounds (ensemble uncertainty ± 0.15 m a$^{-1}$). Thus, the probability that 1974-2006 elevation changes are below -0.31 m a$^{-1}$ and 2006-2015 above this value is less than 5%, and the confidence level in accelerated thinning rates is 95%.

**Page 2, lines 8-10. Strange sentence, since you talk about examples of regional differences but only mention the upper limit values.**

The reviewer is right that the sentence was incomplete. We will change the sentence as follows: "Prominent examples of current-day regional differences in glacier evolution across the Hindu Kush–Karakoram–Himalaya (HKH) are the reported positive glacier mass balances in the Pamir and Karakoram ), *while glaciers are thinning and receding in the rest of the HKH (e.g. Bolch et al., 2012; Kääb et al., 2012; Gardelle et al., 2013)"*

**Page 2, line 15. Is "scientific debate" a good phrase to describe this, isn't the common goal of everyone studying this just to obtain answer to the same scientific questions?**

We agree with the reviewer. We will replace "scientific debate" by "research" and change the sentence as follows: "However, also within the same climatic region the rate of glacier changes can be highly heterogeneous (Scherler et al., 2011b). As such, a main focus of current *research* concerns *the effect of supraglacial debris-cover on glacier response to climate*."

**Page 3, lines 7-17. Here the authors seem to give observations and models the same weight. When you have models on one hand and on the other hand conclusive observations, which don't fit the models, the reason for this is usually the incompleteness of the models, which in this case is probably the melting mechanism of the debris covered glacier.**

We kindly disagree with the reviewer. In our opinion the results of the two cited detailed modeling studies (Juen et al., 2014; Ragettli et al., 2015) are also relevant when discussing the effect of debris cover on melt. Note that these two modeling studies are based on a large number of field data that were used to inform, develop and validate the model. The two modeling studies include point scale glacier mass balance observations while geodetic studies usually do not. Moreover, the glacier thinning rates derived by geodetic studies are not equivalent to melt rates, because glacier uplift affects the derived thinning rates (see our answer to the reviewers' next comment below), while models can provide actual melt rates. Both modeling studies and geodetic studies have therefore limitations when assessing the role of supraglacial debris on glacier response. We will however slightly change the sentence on model results:

"Several detailed modelling studies on the other hand have *provided evidence for a* melt reducing effect of debris at the glacier scale (e.g. Juen et al., 2014; Ragettli et al., 2015), and have concluded that supraglacial debris prolongs the response of the glacier to warming (*Banerjee and Shankar, 2013*; Rowan et al., 2015)

**Page 3, lines 15-17. I don't understand this sentence. What melt is caused by the glacier emergence velocity? Are you maybe referring to emergence of debris to the surface but not the classical glaciological term emergence velocity?**

The sentence was not clear and we apologize for this. What we meant by the "discrepancy between thinning and melt due to glacier emergence velocity" was that melt and thinning is not the same thing, since glacier emergence has to be accounted for when comparing thinning rates to melt rates (see e.g. Immerzeel et al., 2014). We will state this more clearly in the revised manuscript.

**Page 5, lines 11-17. The author don't discuss at all the effects of seasonal changes on their geodetic results despite the fact that the DEMs (including the ones with most emphasizes, November 1974, October 2006 and February 2015) are from different time of the year. Can the seasonal effect be neglected? If so, based on what?**

According to detailed simulations by Ragettli et al. (2015) for the Upper Langtang catchment and the hydrological year 2012/2013, icemelt during post-monsoon and winter only represents about 2% of annual icemelt from debris-free glacier area and about 3% of icemelt from debris-covered glacier area. The model that had been used for these simulations was informed by a large number of field data to guarantee internal consistency of simulated processes (data from glacier ablation stakes, temperature sensor network, automatic weather stations, glacier surface elevation change derived from UAV observations, glacier runoff data, debris thickness observations) and was thoroughly calibrated and validated. Moreover, also precipitation (and thus snow accumulation in the accumulation areas) is highest during the monsoon season. Post-monsoon and winter precipitation represents less than 20% of annual precipitation (Immerzeel et al., 2014b). Elevation changes during the winter half-year are thus minor in comparison to the changes during pre-monsoon and monsoon (March to September). To convert elevation changes into units of *meters per year* we therefore divide by the number of ablation seasons. This is stated on page 8, line 17. All our DEMs are either from late winter/early pre-monsoon (February – April) or from post-monsoon (October-November). Effects of seasonal changes on the geodetic results can therefore be neglected, especially since we mainly discuss time intervals between DEMs of 4 years or longer. We will state this more clearly in the revised manuscript.

**Page 6, lines 31-32. What about glacier motion, does your velocity data give any upper limit on what the motion of the GCPs could be within the time frame (if so state it)?**

According to our velocity data, glacier motion during a period of 9-18 days leads to a horizontal shift of 10-20 cm. This is less than the grid size of the Pléiades image (0.5 m) and is therefore negligible.

**Page 8, line 1. Systematic errors in the glacier change map?**

We will change the sentence as follows: "Systematic errors *in the elevation change map* due to tectonic uplift which could be relevant after the April 2015 Nepal earthquake are also corrected with the co-registration."

**Page 8, line 4. Did Ragettli et al., (2015) do independent estimate on this or did they get the value from Sugiyama et al., 2013. If the latter Sugiyama et al., 2013 should be referenced for this. This ELA estimate, which presumably is just some average value for this catchment, is used in this paper to estimate accumulation area ratio (AAR) for each glacier. It is then repeatedly referred to in the paper like some actual observation of the AAR for the glaciers. It is not and given the unrealistically high variability of AAR in table 1 (15-86%) it is probably not even a good estimate for individual glaciers.**

ELA estimates: these are two independent observations. The ELA estimate of Sugiyama et al. (2013) is based on thinning profiles of Yala Glacier determined from surface elevation measurements. The ELA estimate of Ragettli et al. (2015) is based on observations from glacier ablation stakes. We agree there is uncertainty in our ELA estimate but it is the best assessment possible for the Langtang catchment. In the revised manuscript we will assess the effect of ±100 m ELA uncertainty.

AAR estimates: we agree that the AARs in our paper should not be regarded as derived from observations but as an estimate. We will make this very clear in the revised manuscript. However, the variability of AARs is not unrealistic, given the large heterogeneity of glaciers in our study catchment. Similar ranges of values can be found in literature (e.g. Khan et al., 2015, find AARs ranging from 7% to 80% in the Upper Indus Catchment based on end-of-summer snow line elevation observations). Extreme values, such as the AAR estimate of 86% for Kimoshung Glacier, are discussed in the paper (P. 16, line 2) and can be explained by topographic characteristics.

**Page 9, line 1. I am not really following you here, when you mention the term automated flow accumulation process. Are you delineating ice divides between neighbouring ice catchments? Is the big difference for Langshisha glacier between Pellicciotti et al.,(2015) and this study caused by some part of Langshisha glacier as defined in the former study, being considered as separated ice catchment in this study? If so state this clearly. I would also recommend that you revise Figure 1 to better reveal the coverage of each glacier with improved background image behind it. By doing so you can (hopefully) convince the reader that your delineation of the glaciers is the more appropriate one.**

Yes, some parts of Langshisha glacier as defined in the former study by Pellicciotti et al. (2015) belong to a different catchment. This is very clear if a high resolution DEM is used to delineate the upper boundaries of glacier but is not evident from optical images, since the ice divides are often entirely snow covered. We will state this more clearly in the revised manuscript.

In the revised manuscript we will use the Cartosat-1 2006 ortho-image as a background image (see revised Figure 1), from which the glacier delineation should be clear. The shading on north-aspect slopes slightly facilitates the visual identification of ice divides.

**Page 9, lines 2-5. This is a huge difference and is bound to have great effect on the result. You compere these two studies later on for this glacier, without even mentioning this important difference.**

We agree the differences in area are large. However, the area differences do not explain the different results obtained by the two studies. With larger accumulation areas our value for 1974-2006 would be even more different from the value identified by Pellicciotti et al. 2015, since it can be assumed that our value would become more positive (now $0.09 \pm 0.09$ m w.e. a$^{-1}$), and not more negative as in Pelliccotti et al. 2015 (-$0.79 \pm 0.18$ m w.e. a$^{-1}$). Note that the reasons for the differences in the determined mass balances for Langshisha Glacier are discussed in the manuscript (P. 29, line 6). The differences in glacier area between the two studies can therefore not explain the differences in mass balance, but the discrepancy has to be explained by differences in the outlier correction procedures and by the possible underestimation of the SRTM radar penetration depth by Pellicciotti et al. 2015. We will state this more clearly in the revised manuscript.

**Page 9, line 12. Standard deviation of deltah/deltat at given point calculated for the up 28 difference maps or is this calculated over a given window?**

The standard deviations of $\Delta h/\Delta t$ ($\sigma \Delta h/\Delta t$) values were calculated for each difference map and each 50 m elevation band of each debris-covered glacier tongue. In the revised manuscript $\sigma \Delta h/\Delta t$ will not be used anymore but we will directly use the information of the cliff and lake inventories to identify cliff/lake areas.

**Page 9, line 16. Well here is the answer to the question above. Personally I don't find this a good way writing, when something is only partly explained in a sentence and the same sentence and the following sentence does not indicate that further explanations will be given, but then later on the missing puzzle suddenly pups up. When I read such text, I am always asking myself "did I miss something?"**

We agree that the two sentences (one starting on line 13 and one on line 16) should have been presented in reverse order and we will do this in the revised manuscript.

**Page 10, lines 12-15. Here a justification why this should be errors but not actual elevation changes are completely missing. The span of elevation change rate over an entire glacier can**

**easily be greater than the DEM errors but this depends on the time span, DEM quality, glacier type, etc.**

We agree that a justification was missing. We will correct this in the revised manuscript. $3\sigma$ levels will be selected for outlier definitions outside the accumulation areas following e.g. Gardelle et al. (2013). $3\sigma$ error levels are less strict that the $2\sigma$ levels that were used in the original manuscript and therefore the risk of misclassifying actual elevation changes as errors decreases. On the other hand, more strict $1\sigma$ error levels will be applied in the accumulation areas since here outliers are more likely to occur and since in the accumulation areas only narrow ranges of values are plausible over periods of several years (see our explanations above).

**Page 11, line 4. Outlier correction uncertainty? Do you maybe rather mean sensitivity to outlier removal?**

In the revised manuscript the term will be renamed 'sensitivity to outlier correction'. We think the two terms are mostly equivalent, since sensitivity to outlier removal leads to uncertainty in the geodetic estimates, given that perfectly objective and unambiguous threshold criteria for outlier detection do not exist.

**Page 11, lines 5-17. This is very confusing text. I don't really understand what you are doing including why the thinning rate 2006-2015 is appropriate proxy for the outlier removal (of all data sets or just the 2006-2015 difference map?).**

Here we assess if the mean $\Delta h/\Delta t$ values are sensitive to outlier definitions. Since 'Outlier correction uncertainties' (now renamed to 'sensitivity to outlier correction') correlate with the uncertainty estimates (in the revised manuscript determined following Gardelle et al. 2013) this subsection will be removed from the main paper to shorten the paper and to make it more readable. 'Outlier correction uncertainties' (i.e. uncertainties associated with the correction of outliers) will not be used anymore for the detection of $\Delta h/\Delta t$ map outliers.

The thinning rate 2006-2015 was the value used as threshold to identify outliers, in order to keep the level of noise below the level of signal (where the thinning rate 2006-2015 is the signal and the outlier correction uncertainty is the noise). However, we agree that we could as well have chosen a different thinning rate from the ensemble as threshold. In this respect the 2006-2015 thinning rate was not a good choice, because our decision lacked objectivity.

**Page 11, line 18. DEM adjustment uncertainty? Is the term uncertainty appropriate here? I do not see that the parameter explained in this section is really used in your uncertainty assessment.**

We agree that the term 'DEM adjustment uncertainty' was not a good choice since in the literature it is known as 'triangulation residual' (e.g. Paul et al., 2015). In the revised manuscript the results will be presented as 'triangulation residuals' and moved to the Supplement, also to shorten the paper. 'Triangulation residuals' will not be used anymore as outlier criteria in the revised manuscript since the results correlate with the uncertainty estimates (now determined following Gardelle et al. 2013). We agree that the determined triangulation residuals were not abundantly discussed in the original manuscript (we only refer to Figure S2 on page 16, line 1).

**Page 11, line 29. I have problem obtaining the same results as the authors from this equation. If n=8 making Ndeltat=28 and k=3, I get C_2= (8 over 3)/(2*28)=(8!/(3!*5!))/(2*28)=56/56=1, not 6 as authors say one should get.**

The equation as shown in the paper was wrong but our calculations were correct. The denominator should return the number of permutations for a given k-element subset that can be selected from a number of n objects. The correct expression is 'n!/(n-k)!)' instead of 'n over k'.

The equation will be removed from the main manuscript. It is sufficient to refer to Nuth and Kääb (2011) when introducing the concept of triangulation residuals (and these will only be presented in the Supplement).

**Page 12, line 7-14. It took me quite a bit of time to actually understand what you are doing. I think I do now. Again I can't see what is logical about using the thinning rate from October 2006 to October 2015 as a threshold value. Can you explain that?**

The thinning rate 2006-2015 was used as a threshold to guarantee that the signal to noise ratio is higher than 1:1, which indicates more signal than noise (same explanation as above regarding the comment on P. 11, lines 5-17). However, the outlier criterion discussed here will not be used anymore in the revised manuscript and the selection of Δh/Δt maps for the ensemble will be based on simple and straightforward criteria, following the reviewer's comments.

**Page 12. Do I understand you right that the last outlier detection you do is the catchment scale outlier detection? Wouldn't be more appropriate to do that before you the do glacier scale outlier detection?**

The order of steps is not significant, since the results of the glacier scale outlier detection do not depend on the catchment scale outlier correction. Note that the outlier criteria discussed here will not be used anymore in the revised manuscript.

**Page 12, lines 23-24. Here we are left with the question "how?" until half a page later. Again, this is not a good way of writing, it makes the paper hard to read.**

We apologize for the writing style here. The outlier criteria discussed here will not be used anymore in the revised manuscript and therefore the sentences "half a page later" will be removed.

**Section 3.4.3. Here you come up with three outlier criterion. Why this complexity? It is not really justified in the paper.**

The three criteria look at mean off-glacier elevation differences (MED) and the role of slope and snow cover for MED. These criteria will not be used anymore for outlier detection in the revised manuscript. Figure 3 of the original manuscript will be moved to the Supplement.

**Sections 3.4.2-3. It seems to me that the glacier catchment scale outlier removals are not likely to function appropriately when the time interval between DEMs is so variable and you do the outlier detection on deltah/deltat. deltat is ranging from < 1 year up to 32 year. This means e.g. for the last criteria that the DEM error for the 1974 DEM causing the 1974-2006 delath/deltat to be considered as an outlier would need to be 32 times larger than the error in 2009 DEM causing the 2009-2010 deltah/deltat to be considered an outlier. DEMs over short interval off course need to be very accurate to have informative value for volume change estimates, hence this is logical from that perspective. If my understanding of the outlier removal procedure is correct it does however result in very weak outlier criterion for the 1974-2006 interval. If the authors rely entirely on this automatic outlier removal, it may result in erroneous result for this period, which to me, seems to be the case when looking at Figure 6 a. This is very unfortunate given that the main focus of your results and discussion is on the difference between the periods 1974-2006 and 2006-2015.**

At the catchment scale we used the distributions of the mean off-glacier elevation differences (MED) to identify outliers. We agree that the DEM errors are more likely to be classified as outliers if the intervals between DEMs are short. This is certainly one of the reasons why the DEM differencing maps involving the Hexagon 1974 DEM were not identified as outliers. However, we think it is justified to assess here the signal to noise ratio (thus using units of m/a) and not to compare absolute values (due to DEM errors, units in m) for outlier detection, since throughout the manuscript we are using units of m/a to discuss elevation changes.

To prevent erroneous results for the period 1974-2006 we now apply stricter outlier definitions for the accumulation areas, since only narrow ranges of Δh/Δt values are realistic in the accumulation areas. Note that also before the revisions of our methods; it would not have been justified to reject the entire 1974-2006 Δh/Δt map, since the quality at debris-covered areas is very good. Figure 6a confirms that the off-glacier elevation differences at lower elevations (where the image contrast is high and the terrain is less steep) are very small.

**Page 14, line 5. I find the problem with your bias or trend correction approach manifest in this equation (I guess you are not the only one doing this). If this study had been only on one of these glaciers the data used for trend or bias correction would (presumably) only have been from the neighbouring area of this glacier resulting in MED~=0. But since you do the trend correction for the catchment as a whole (which I think is fine if you are studying the catchment but not individual glacier), MED~=0 is often not true for individual glacier, hence you will get different value for a given glacier than if you had focused the study only on that glacier. You are trying to compensate for this by adding this effect here into the uncertainty, but you are still left with the fact that the probabilistic mean of the actual average elevation change is likely not well represented by the centre of the given error bars. This becomes particularly awkward since your discussion of the results almost neglect the derived uncertainty limits and focuses on the centre of the error bars.**

We agree with the reviewer that the center of the error bars does not well represent the probabilistic mean of the actual elevation change if MED~=0, and we agree that our discussions should have better reflected this. However, MED can only be different from zero at the very high elevations, where the presence of snow does not allow using off-glacier terrain for bias correction. In the revised paper we will use an entirely different approach to deal with Δh/Δt errors in the accumulation areas of glaciers and consider only a narrow range of plausible values close to zero (see the description of the new outlier correction approach above). As a consequence of this new approach the center of the error bars now agrees with the probabilistic mean of the actual elevation change, and it is also not necessary anymore to consider MED for the uncertainty calculations (hence we use only the standard error following Gardelle et al. 2013).

**Page 14, lines 8-10. Are you saying that you use n=1? If so state it clearly, you could add to the sentence (i.e. n=1). Your usage of i.e. is not appropriate here (if I understand the sentence correctly). The fact that you use n=1 implies only that all pixels within the elevation band are fully dependent on one another (which truly is a conservative estimate). It does not however implies that there is no dependence between elevation bands. Since no attempts has really made to quantify the effect of the spatial correlation of your data (see e.g. Rolstad et al., 2009 or Magnússon et al., 2016, for further info) we don't really know if your assumption of no error compensation across elevation band is likely to lead to a conservative estimate of the uncertainty.**

We now use the standard error following Gardelle et al. 2013 for uncertainty estimations, where n is equal to the number of independent measurements per altitude band, which means that the spatial correlation of the data is taken into account. However, we would like to clarify that in the original manuscript n was not equal to 1, but n was the number of pixels per elevation band (Page 14, line 3). We apologize if our usage of i.e. was not clear ("*i.e. assuming no error compensation across elevation bands*"). It really meant that we weighted the uncertainties identified per elevation band according to elevation distributions to calculate weighted averages per glacier. This implies that we are assuming no error compensation across elevation bands (and thus 100% dependency between elevation bands). This is indeed a conservative estimate.

**Section 3.6. It seems to me that your surface velocity could do with some more masking of errors and outliers e.g. with correlation threshold. The masking that you are carrying leaves almost the entire velocity field intact as revealed by Figure 11 even though it is clear that much of it is just errors. The level of errors seen outside the glaciers is such that it is not clear if the signals on the glaciers are real or just errors as well. The figure itself is very hard view.**

We agree that Figure 11 needed to be improved, so that signal from glaciers can be better distinguished from errors outside the glacier area. Note that on the relatively flat debris covered areas errors are much less likely to occur. Errors occur where the terrain is steep or where image contrasts are low.

In the revised Figure 11 we masked out areas with slopes that are not representative for glacier area. We used a threshold of 45°, which corresponds to the 95th percentile of the slope of all glacier grid cells. Off-glacier velocity data are shown in transparent color so that signal from glaciers can be better distinguished.

The velocity profiles of debris-covered tongues (and error bars) are now also shown in the revised Figure 10. The error bars represent the standard deviation in pixel values per elevation band and do not suggest that additional outlier correction is necessary

**Page 15, line 9. Outlier and uncertainty assessment? Confusing. Wasn't this already done?**

Here we presented results and not methods. As such, this section could have been part of the result section, but we decided to discuss uncertainties and outliers in a separate section given their importance. In the revised manuscript this section will be removed to shorten the paper, since most of its content will become redundant after the simplifications in the data selection procedure.

**Section 4. This is all rather confusing. You calculate a lot of quality proxies used for outlier detection, mostly to convince yourself that the data that you derive your results from is of good quality. This is all good if one also reviews critically the outcome, which seem to be lacking in this study. A lot of these proxies are referred to as uncertainties apparently without being used to estimate uncertainty of the presented geodetic results. It is also not clear if all the DEM available during the period 2006-2015, apart from the initial and the final DEM, were really used to narrow down the uncertainty of volume change during this period. If not it seems to me that this paper would be much clearer if the focus of this paper were only on three DEMs, the ones from 1974, October 2006 and February 2015.**

We are not convincing ourselves of a good data quality. This section simply presents an honest assessment of uncertainties and outliers. The quality proxies that were chosen have all been already applied in previous studies (although mostly for quality assessments and not for outlier removal). We agree however that outlier removal algorithm obviously failed in the accumulation areas. In this

respect we should have reviewed the outcome more critically. In the revised manuscript we correct this by improving the outlier removal at the grid scale. Those instances of failure are no longer there.

It is true that the proxies were not used to estimate uncertainty of the presented geodetic results but only for quality assessments and for outlier removal. This made the paper lengthy and difficult to read. In the revised manuscript the quality proxies DEM adjustment uncertainty (or 'triangulation residual'), outlier correction uncertainty (or 'sensitivity to outlier correction') and mean elevation differences (MED) will only be presented in the Supplement and only for quality assessment. Our main criteria to select Δh/Δt maps for the ensemble will be the time interval between DEMs, since we can show that the uncertainties (now calculated following Gardelle et al. 2013) decrease with period length (see new Figure 3). In this respect the uncertainty estimates will be used directly for outlier detection. Section 4 will be entirely removed to shorten the paper.

We are sorry if it is not clear that we used all DEMs available for 2006-2015 to narrow down the uncertainty. We thought it was (e.g. P. 18, lines 1-3; P. 18, lines 10-12; P. 18, lines 25-26, P. 24, lines 30-33, P. 30, lines 9-13). We did use all the DEMs and are convinced that there are clear advantages in doing so, since several independent measurements (differential DEMs for the period 2006-2015) lead to higher confidence in detected signals, even if the uncertainty of each measurement is high. For this reason, we do not see the point of using only three DEMs (the alternative suggestion of the reviewer). However, we will make an effort to more clearly explain the advantages of the ensemble approach in the revised paper.

**Page 18, line 18. Well if you think this is due to remaining systematic error, did you consider that your outlier removal is maybe not functioning so well?**

The outlier removal functions excellently for ablation areas, considering for instance Figure 8, which clearly shows which thinning patterns are consistent across different dataset. If we considered all 28 differential DEMs for Figure 8 it would not have been possible to clearly identify patterns which are consistent across datasets, because the ability to identify these patterns requires a level of accuracy which is not granted per se. However, in the accumulation areas we agree that the outlier removal did not help to clarify thinning/thickening changes. This is why we decided to choose a different approach for grid scale outlier correction here (see above). The outlier removal the glacier and catchment scale has also been completely revised (see above).

**Page 18, line 23-24. This is very true. Unfortunately you seem to forget it repeatedly in your discussion. Given that your uncertainties will be the same after revision of this work, much of the discussion on the results can be omitted because it is meaningless due to the large uncertainties.**

After revision of the methods the uncertainty estimates are different, especially for the periods 1974-2006 and 1974-2009. Regarding the particular example of Kimoshung Glacier, to which the reviewer is pointing here, the patterns are consistent and the uncertainty could be constrained by only accepting realistic Δh/Δt values in the accumulation areas. We therefore retain our conclusions. We are puzzled by the dismissive tone of the reviewer.

**Page 19, line 6. This is a good example of what I am talking about regarding the author neglecting the uncertainty in their discussion of the results. You cannot state here that the thinning rate increased by more than 100 %. If we know that John owns between 0 and 4 cars and Mike owns between 2 and 6 cars can you state that Mike owns at least twice as many cars as John? No and the probability of such statement being true is only 14/25=0.56 (given even probability distributions for the car ownership in both cases).**

The reviewer's schoolmasterly example is not appropriate here. We are not comparing only two estimates, but thinning rates of two periods (1974-2006, and 2006-2015), where for the second period we use several independent datasets. This changes the situation as a whole. To take up the reviewers' example, John belongs to an automobile club and we are estimating the average number of cars owned by the members of the club. If each member owns between 2 and 6 cars, then the standard deviation of the sample mean is

$$\mathrm{SD}_{\bar{x}} = \frac{\sigma}{\sqrt{n}}$$

(https://en.wikipedia.org/wiki/Standard_error).

whereas σ is the standard deviation of each single estimate and $n$ the sample size (number of club members). Therefore the standard deviation converges towards zero for a large ensemble. Already with only four ensemble members the standard deviation of the sample mean decreases by 50%. In other words, the error might be too large when comparing only two periods, but when comparing two groups of values for the two periods the differences between the values become significant. We will show this clearly in the revised manuscript (see also the revised Figure 7).

In a revised manuscript we will emphasize better the value of the ensemble approach. Rather than stating "on average, thinning rates increased by more than 100% between the two periods" we will emphasize the ensemble results and state "the mean thinning rates of all differential DEMs in the 2006-2015 ensemble increased by more than 100% with respect to the results for 1974-2006".

We would also like to strongly rebut the reviewer's statement that we are neglecting the uncertainties in the discussion of our results. In the particular sentence to which the reviewer is pointing here the uncertainties are provided in brackets. We think this is an honest way of presenting the results. Uncertainties are discussed abundantly throughout the manuscript (with Section 4 we dedicated a whole section to the discussion of outliers and uncertainties). We will however revise the manuscript in order to discuss uncertainties together with the results (and remove Section 4).

**Section 5.2.1. This comparison between debris covered and debris free glacier looking at "Explanatory variables" is rather primitive. For one thing it is rather inappropriate to refer to some of them as variables. I would rather refer to outcome of processes, which in some cases probably show correlation since they are dependent on the same physical variables. It is also strange that only Yala is included as candidate for the debris free glacier. The behaviour of Yala is then compared with 5 other debris covered glaciers. Even though the difference between Yala and each of the other 5 glacier is sometimes visually clear it is misleading to calculate the r-value for all the 5 debris covered glacier at ones and compare with a value calculated for a single glacier. When using data from several glaciers, various variables which effect the glaciers in different manner is bound to reduce the studied correlation compared to having data from just a single glacier.**

This section will be thoroughly revised, also based on comments by reviewer 2. We now use a new dataset of cliff and lake areas, which substitutes the cliff/lake proxies (detailed explanations above under 'New Methods' and 'Explanatory variables'). The old Figure 10 will be removed and replaced with a new one. With the scatterplots and r values (old Figure 10) we attempted to explain the variance in thinning rate changes of all debris-covered glaciers at the same time. However, single variables cannot possibly explain all the spatial variation in thinning rates, and therefore the r values were generally low (max. 0.55 for debris-covered glaciers). We agree that the sample size affects the correlation coefficients, and therefore the r values calculated separately for debris-covered terrain and Yala Glacier were not comparable.

In the revised section 5.2.1 we will discuss the spatial variability in thinning rates at each debris-covered glacier separately (see revised Figure 10). Regarding the differences between debris-free and debris-covered glaciers it is sufficient to state that at debris-free glaciers thinning rate changes are elevation dependent while this is not the case at debris-covered glaciers (and refer to Figure 9). Note that we think it is justified to consider Yala Glacier as the main reference for debris-free glacier characteristics. Kimoshung Glacier has an unusually high accumulation area ratio (~80%), is very dynamic and is affected by higher $\Delta h/\Delta t$ uncertainties (due to the low image contrast in the large accumulation area). Yala Glacier has an AAR of approximately 40% (Table 1) which is identical to the mean AAR indicated by Kääb et al. (2012) for the entire HKH region and is therefore suitable as a reference. In the revised paper this will be mentioned

It is not clear to us which of the variables we cannot refer as "variables". In Section 5.2.1 we considered elevation, slope, surface velocity and the variability of local thinning rates (as cliff/lake proxies). The reviewer is right that e.g. the variability of thinning rates ($\sigma$ dt/dh) is an outcome of processes. All the others are variables. Still, in our opinion it was correct to refer to them as explanatory variables ($\sigma$ dt/dh explains the role of heterogeneous surface properties for local changes in thinning rates).

**Page 22, line 1. You mean April 2015.**

Yes, thank you for noticing. We will correct this.

**Section 5.3. It is probably of interest for some to know the volume of these enormous avalanches. It however seems clear that avalanche falling on debris covered glacier (particularly the low insulated part of the glacier) is only going have minor and short last effect on the mass balance since it is going to melt much faster than the debris covered ice beneath it.**

It is not clear per se that the avalanche cones disappear quickly. It is documented and described in our manuscript that a new debris layer appears on top of the avalanche material (as snow/ice melts out from the debris). Avalanche accumulation is one of the most important processes for debris-covered glacier formation (Scherler et al., 2011a). Likely the tongue of Lirung Glacier would not exist without accumulation through avalanches (Ragettli et al. 2015). We therefore do not agree that the long term effect on mass balance of the enormous post-earthquake avalanches is clear. In the revised manuscript we will provide more background information about debris-covered glacier formation.

**Page 24, line 25. What is numerical evidence?**

We will remove "numerical" and just leave the "evidence".

**Page 25, line 14. Is there no uncertainty in the area change?**

Uncertainties are negligible for the purpose of this study. See our response to the reviewer's comment regarding Table 7 below.

**Page 25, lines 16-17. I don't understand what you specifically mean by correlation between areal changes and surface elevation height.**

What we meant here is that one can expect glacier mass balances near steady state if the glacier area remains nearly constant, as it is the case for Kimoshung Glacier. We will revise the manuscript text to be clearer.

**Page 25, lines 23-26. Again not promising for your outlier removal, even though it is better to admit it does not work to well. It would however be even better to justify why you think this is an error, e.g. by pointing out that local lowering of 60-100 m over 32 years (as indicated by figure 6a) on such small glacier at such high altitude is very unlikely to say the least.**

We agree with the reviewer that the values were unrealistic. We correct this with a new outlier removal approach for accumulation areas (see response in our general statement above).

**Page 25, line 28-29. Even though it is likely that hypsometry plays a crucial role here this statement is far too bold given that it is based on very limited and apparently erroneous data (according to Figure 6a).**

After replacing the erroneous data we now get results which are trustworthy. The data now unambiguously reveal the differences between Kimoshung and Yala Glacier (see new Figure 7). Those differences can only be explained by the very different altitudinal distributions of the two glaciers (see **Error! Reference source not found.** below in our response to reviewer 2 regarding the same point). The results now clearly suggest that the hypsometry plays a crucial role.

**Page 25, line 31. See my previous comment regarding the AAR.**

We think the AARs in our study are realistic estimates. Especially at Kimoshung Glacier an uncertainty in the ELA by ±100m would not lead to very different AARs because the glacier is very steep near the ELA (AARs given an ELA uncertainty of ±100 m: 80%-87.5%). The uncertainty about the ELA does not change the fact that the AAR of Kimoshung Glacier is high, and this is what really counts for the discussion here.

**Page 26, line 26. Here and at other places in this paper, some temperature data (if available) would support your discussion.**

It is not the purpose of this study to document warming trends in the Nepalese Himalaya. It is difficult to relate spatial changes in elevation over different glaciers to remote point observations of temperature trends and the meaning of this would be very limited. In fact, no study of geodetic mass balance has used this coarse information to explain some of the observed thinning patterns. We therefore refrain from doing this here. We can provide more references however.

**Page 27, lines 16-18. You have far too little data with far too great uncertainty to make such statement.**

We agree with the reviewer that the wording can be improved here. We stated that our observations do not support the findings of previous studies and that is true. We find this an important result especially because it is the first time that this is proven with detailed, multi-ensemble data for a specific catchment using high resolution DEMs (in contrast to large scale regional studies). We agree that we cannot extrapolate this finding to larger glacier samples.

**Page 27, lines 20-23. You can say that Kimoshung glacier has higher hypsometry than Yala. The staggering difference between AAR values (which here are treated as some kind of truth but not as estimates based on the assumption of fixed ELA=5400 m a.s.l. for the whole catchment) is however misleading.**

We do not agree with the reviewer, as explained above. Previous studies have found similar ranges of AARs in the HKH (e.g. Khan et al. 2015) and the AAR differences between individual glaciers (e.g. Yala and Kimoshung Glacier) can be explained by topographic differences.

**Page 28, lines 16-17. I am confused, is this in accordance with previous statement in this section (page 27, lines 16-18).**

We apologize if the wording was not clear here. Our observations tell us that the thinning rates at debris-covered tongues are lower than at debris-free glacier area AT THE SAME ELEVATION (this relates to the discussion on page 27, lines 16-18).However, when it comes to glacier mass balances or glacier-wide average thinning rates, no significant differences between debris-free and debris-covered glaciers can be identified. We will explain this better in the revised manuscript.

**Page 28, lines 27-28. There is completely insignificant difference between these values. There is no point in trying to explain the "difference" between them.**

We agree that the differences are not significant, but we think it is justified to point to a possible overestimation of thinning rates identified by Pellicciotti et al. (2015) due to an underestimation of the SRTM radar penetration depth. We will revise the text of the manuscript to make this clearer.

**Page 29, lines 6-10. I am very puzzled here. You need to justify here why this data is now suddenly considered as usable data, when the one processing the data rejected it in recently published paper. Why has he/she as the third author of this paper changed his/her mind?**

The two studies (Pellicciotti et al. 2015 and the present study) use slightly different outlier correction approaches, but both studies tried to avoid arbitrary or subjective criteria (both are based on 2σ thresholds to define outliers at the grid scale, but in Pellicciotti et al., 2015, those thresholds are calculated only once for all altitude ranges together). In the present study we used a slightly less restrictive outlier definition (thresholds calculated separately for accumulation and ablation areas), with the consequence that at some places (e.g at Langshisha Glacier) erroneous data remained in the dataset, while at other places (e.g. at the tongues of debris-free glaciers) the application of a less strict criteria lowered the risk of classifying correct data as outliers. This choice is justified by the fact that the quality of the new DEMs (2006-2015) is generally much higher than the quality of the Hexagon 1974 and the SRTM DEMs used in Pellicciotti et al. (2015). The third author therefore did not change her mind. The choice of the method depended on data quality. We also notice that there is progress in scientific research, and that improvements of revision of methods are common in papers by the same authors when data of better quality are available or new approaches are deemed more suitable. We do not see anything wrong here.

However, the 1974 DEM is the same for both studies, and therefore the less restrictive outlier definition failed to identify erroneous data in the 1974 difference maps. We have corrected this with the revised outlier correction procedure (see 'New Methods' above).

**Page 29, line 13. How can you state this? Does including apparently erroneous data make the uncertainty estimate more realistic?**

Yes, we think that the uncertainty estimates should reflect the quality of the data. On the other hand it is also justifiable to exclude erroneous data from the start and use uncertainty estimates that reflect the quality of the data without considering those outliers. We think this is what the reviewer suggests and this is what we will do in the revised manuscript. In general, please see our response on the new outlier removal and uncertainty estimates.

**Page 29, lines 15-16. What other data did they use? They were hardly using GPS in 1982.**

The reviewer is right, they did not use GPS in 1982. To cite Sugiyama et al. (2013): "The surface elevation in 1982 was surveyed by ground photogrammetry (Yokoyama, 1984) and later digitized into

a 10m resolution digital elevation model (DEM) (Fujita and Nuimura, 2011)." We will state this correctly in the revised manuscript.

**Page 30, lines 21-23. Sorry, I don't think many will agree on this statement.**

We think that many will agree on this statement – especially after the revision of this paper. An ensemble of DEM differencing maps that is based on 8 high resolution DEMs is unique for the HKH region. The dataset provides detailed insights into glacier response to climate change. The spatio-temporal resolution of our analysis is higher than of any previous geodetic study before. We think it is therefore justified to call our work "one of the most rigorous documentations to date of glacier response to climate change over the last 40 years in the Himalaya".

**Page 32, lines 5-9. This text does not fit into conclusion. If the authors think this text should be in the paper, it would be more appropriate to include it in the introduction.**

We agree that the text does not fit here. We will move the text to the introduction or remove those lines, as suggested by the reviewer.

**Page 40, Table 1. See previous comments regarding the AAR.**

The reviewer's comments regarding AAR are addressed above.

**Page 42, Table 5. It is not clear how the uncertainty of the average elevation change over the entire Langtang glacier catchment is calculated. Given its value it seems close to being basically (-)/(A\*2), which basically corresponds to assuming that errors between glacier are completely dependent. Such assumption gives really conservative estimate, even too conservative causing the results to be downgraded. I also recommend that you stick to the same order of glaciers in the table as given in Table 7 with the glacier id.**

We calculate uncertainties of the entire glacierized terrain identically as for any other given area: we consider the off-glacier uncertainty per elevation band (eq. 5) and the altitudinal distribution to calculate weighted averages (page 14, lines 10-12).

It is not clear to us what the reviewer means by (-)/(A\*2). It is true that errors between glaciers are dependent, since always the same off-glacier data are used. Only the altitudinal distributions are different (and therefore the uncertainty estimates). We will state this more clearly in the revised manuscript.

We will change the order of glaciers in Table 5 as suggested.

**Page 43, Table 7. Why no uncertainties? Are they within the digit of the given value, or did you simple not think about it? You are discussing these area changes in the paper without giving the reader any confirmation that these changes are significant.**

Certainly there are uncertainties in the area change values. However, we are confident that by using 1.5 to 4 m resolution optical satellite images we can determine area changes at the glacier tongues with a level of detail that justifies presenting values rounded to 10'000 m$^2$ (two digits after the decimal point if units are expressed in km$^2$ as in Table 7). For example, 2.5 m horizontal uncertainty (pixel size Cartosat-1 2006 ortho-image) at the tongue of Kimoshung Glacier leads to a 2006 area uncertainty of only 2000 m$^2$. Uncertainties in area changes are therefore within the digit of a given value and are thus not provided in Table 7. Note that we only re-delineate the tongues to assess area changes. The

accumulation areas (where the glacier outlines are most uncertain) are only delineated once (P. 8, lines 24-26).

**Page 43, Figure 1. See my previous comments regarding this figure.**

The reviewer's comments regarding this figure are addressed above.

**Page 44, Figure 2. The data on the debris covered glaciers is the most convincing part of this manuscript.**

We thank the reviewer for this assessment and we agree with his evaluation. Indeed, the uncertainties are low over debris because of good image contrast and shallow slopes. We agree that here the data are least ambiguous, and allow to unambiguously infer some very interesting results on thinning patterns.

**Page 45, Figure 3. It seems to me that using all the 6 proxies result in the same outlier removal as when you just use med2 and sigma2.**

The reviewer is right. As explained above, none of these criteria will be used anymore for outlier removal in the revised manuscript. This figure will be moved to the Supplement and will only show the stable terrain uncertainties of the $\Delta h/\Delta t$ maps in the final selected ensemble (excluding short periods and the ALOS PRISM 2010 DEM).

**Page 46, Figure 4. 50% confidence level? What would the error bars be for a reasonably strict confidence level like 95%?**

In the revised manuscript we will provide the error bars with a confidence level of 95% but only considering the $\Delta h/\Delta t$ maps of the final selected ensemble. The revised Figure 4 is provided at the end of this revision statement.

**Page 47, Figure 5. Do you mean: a) A whiskers plot showing the distribution of uncertainties for the (up to?) 28 deltah/deltat maps. What do the red crosses indicate?**

The boxplots show the distribution of uncertainties for non-rejected $\Delta h/\Delta t$ maps. We will specify this in the caption text of the revised manuscript. The red crosses indicated outliers according to the standard Matlab boxplot function. In the revised Figure 5 we removed the red crosses and whiskers extend to the most extreme data points. The revised Figure 5 is provided at the end of this revision statement.

**Page 47, Figure 6. See various previous comments on this figure. Should also be enlarged for better readability.**

The reviewer's comments on this figure are addressed above. We will decide later (in the proof stage) if this figure will be sufficiently large if printed as a two-column paper figure.

**Page 48, Figure 7. The order of panels for the glaciers should be kept the same as the numbering of the glaciers in Table 7.**

Ok, we have reordered the panels as suggested (see revised Figure 7 below).

**Page 50, Figure 9. Something went wrong with the altitudinal distribution for Yala.**

The altitudinal distribution for Yala Glacier is shown correctly. The altitudinal distributions of all glaciers are shown for 50 m elevation bands. The reviewer might not have noticed that the x-axis

ranges are different for each sub-figure. We now point to this in the revised caption text (see revised Figure 9 below).

**Page 51, Figure 10. See previous comment regarding this figure.**

This figure will be removed from the paper and replaced by a new figure (see Figure 10 below) that better shows the relationship between glacier thinning and glacier characteristics and supraglacial features (see previous comment on this).

**Page 52, Figure 11. Very hard to read. The arrows are e.g. very hard to detect. Results do not appear very reliable (see previous comment).**

The reviewer's comments on the reliability of the velocity data are addressed above. We have improved the quality of this figure (larger arrows, focus on debris-covered areas, see revised Figure 11).

**IN RESPONSE TO REVIEWER 2**

**General comments:**

**This paper presents glacier surface elevation change in Langtan Himal from 1974 to 2015 based on DEMs generated from satellite images. The authors analyzed temporal and spatial patterns of glacier thinning over the studied seven glaciers. Focuses of the discussion are spatial heterogeneity in the thinning rate, comparison of debris-covered and debris-free glaciers, changes in the thinning rate after 2006. The data are also used to quantify the impact of the earthquake in 2015.**

**Despite the increasing importance and interests on the Himalayan glaciers, long-term data on glacier changes are few in the region. Considering intensive research activities in the Langtang region in the past and recent periods, the presented data set is valuable. Nevertheless, uncertainty is rather large particularly in higher elevation areas.**

**This is very common in photogrammetric elevation analysis because snow covered surface loses surface features required for this method. Judging from the unrealistic thickening and thinning patterns in Figure 6a, it is questionable whether the DEM analysis is applicable in the accumulation areas. Moreover, estimated uncertainties are based on very complex outlier rejection criteria, which sometimes appear to be subjective and unconvincing. These problems result in limited reliability in the conclusions. Overall impression on the manuscript is that conclusions are too conclusive as compared to what are shown by the data.**

**I encourage the authors to thoroughly revise the manuscript (1) by using only reliable data, (2) with well focused objectives, (3) to draw only convincing conclusions. For example, omitting data from the accumulation reduces total uncertainties in Figure 7, which leads to more reliable discussion on recent increase in the thinning rate. Among others, elevation change over the debris-covered regions and impact of the earthquake are promising subjects.**

We thank the reviewer for his/her very useful comments and appreciating that the data set is valuable. We agree with his/her comment on the accumulation area values and have changed for this our outlier removal procedure (following also suggestions from Reviewer 1). As a result, erroneous data from the accumulation areas are now corrected. In this way, all conclusions presented can be solidly justified.

We also agree that the paper should have well focused objectives. We have simplified the procedure for selecting maps for the 2006-2015 ensemble (see explanations above under 'New Methods' and 'Data Selection') and our uncertainty estimates, which will allow to focus better on the three paper objectives as defined in the Introduction (P.4, lines 12-16).

We agree with the reviewer that the results on the debris covered sections and impact of the earthquake are relevant and interesting results of our work, and thank him/her for noticing this.

We kindly disagree that overall "the conclusions are too conclusive as compared to what are shown by the data"; since also with our original methods and data we were able to present unambiguous evidence for heterogeneous thinning patterns (e.g. spatially variable thinning trends at debris-covered tongues). We think this result alone deserves publication in TC, but we agree that the manuscript required revision. After following both reviewers' suggestions for improvement, we are now able to draw more convincing conclusions regarding all addressed main points.

To demonstrate that with the revised methods we are able to draw relevant conclusions that are supported by data we shortly summarize the main quantitative outputs of our study regarding the three main goals:

1.) Assess if overall thinning of glaciers in the region has accelerated in recent years.

Thinning rates have increased, from $-0.25 \pm 0.08$ m a$^{-1}$ (1974-2006) to $-0.46 \pm 0.18$ m a$^{-1}$ (2006-2015, ensemble mean). The uncertainty bounds are overlapping at the ends. However, the probability that thinning rates have not increased is less than 5% (see explanations provided in our response to the comment by reviewer 1 on page 1, lines 18-19).

2.) Determine if spatial thinning patterns have changed over time.

We can now conclude that spatial thinning patterns have changed over time, since thinning accelerations at the debris-covered tongues are highly non-uniform in space. Local changes in thinning rates (comparing the periods 1974-2006 and 2006-2015, Figure 8) range from -80% (at Ghanna Glacier, 4800 -4850 m a.s.l.) to +150% (at Shalbachum Glacier, 4650-4700 m a.s.l.). The uncertainty in identified thinning accelerations is only about ±10%.

3.) Assess if there are major differences between the response of debris-covered and debris-free glaciers in the sample.

Here we partly agree that we were too conclusive in the original manuscript regarding this point. In the revised manuscript we will state clearly that our sample of only two debris-free glaciers (Kimoshung and Yala Glacier) is too small for a robust comparison with debris-covered glaciers, especially since we obtained contrasting results for the two debris-free glaciers.

However, considering that the elevation distribution of Yala Glacier is common for the HKH (an AAR of 40% is common in the HKH, see Kääb et al. 2012), this glacier can be used as a reference. There are indeed ***major differences*** between debris-covered glaciers and Yala Glacier: Within the same altitudinal range, thinning rates of debris-covered glaciers do not exceed 65% - 75% of the thinning rates at Yala Glacier (P. 27, line 26). Considering the changes in mean thinning rates, we identified a strong thinning acceleration at Yala Glacier from $-0.32 \pm 0.06$ m a$^{-1}$ (1974-2006) to $-0.88 \pm 0.23$ m a$^{-1}$ (2006-2015, ensemble median). Thinning accelerations at debris-covered glaciers do not exceed 58% of the thinning accelerations identified for Yala Glacier (largest change in mean $\Delta h/\Delta t$ at Langshisha Glacier, from $-0.11 \pm 0.09$ m a$^{-1}$ to $-0.44 \pm 0.19$ m a$^{-1}$). We will include a balanced discussion of this point, which however seems important to be included as our work is one of the first to assess differences between debris covered and debris-free glaciers at this level of details and high spatial resolution.

**Major concerns:**

**1. Reliability of the DEM in the accumulation area Figure 6a shows unusually large thickening and thinning patterns in the accumulation areas. The regions of the suspicious elevation change agree with the frequently snow covered regions shown in Figure 1. Most likely, photogrammetric analysis is hampered by featureless snow surfaces. Because such data from the accumulation areas are used for the mean thinning rate over each glacier, conclusions on the recent thinning acceleration and comparison between debris-covered and debris-free glaciers are unreliable.**

We have carefully revised our outlier removal procedure to address the reviewers' concern. It is true that outliers in the accumulation areas remained in the 1974-2006 map and therefore the outlier

removal procedure failed for these areas. Our new approach for outlier correction is described at the beginning of this document under 'General Revisions'. The revised outlier detection algorithm now identifies unrealistic patterns in the accumulation areas and removes them. Missing data in the accumulation areas are replaced by plausible values. The revised Figure 6a is provided at the end of this document.

Although erroneous data over featureless snow surfaces in the Hexagon 1974 DEM are evident, we are convinced that with the revised outlier correction and gap filling procedure now allows for convincing conclusions regarding recent thinning accelerations. As the reviewer states above it is common in photogrammetric elevation analysis that uncertainties are high over featureless snow surfaces. Many previous studies addressed the same problem. Errors in the accumulation areas do not require rejecting the whole dataset, since it known that over long time periods only narrow ranges of $\Delta h/\Delta t$ values close to zero are realistic in the accumulation areas (e.g. Schwitter and Raymond, 1993; Huss et al., 2010). Previous geodetic studies have thus assumed no elevation changes in the accumulation areas (Pieczonka et al.,2013) or have used empirical values to define acceptable $\Delta h/\Delta t$ ranges in the accumulation areas (Pieczonka and Bolch, 2015). In our study we benefit of a large dataset of several independent $\Delta h/\Delta t$ maps. We now use the available information to narrow down the uncertainty and replace missing data in the accumulation area with data from the same glaciers (see detailed explanations under 'New Methods' regarding our new approach for outlier detection and gap filling in the accumulation areas).

We now also use a more established method for uncertainty quantification (Gardelle et al. 2013). The new approach results in substantially lower uncertainty estimates for the 1974-2006 scene (see revised Figure 7). Our previous approach clearly overestimated the uncertainties (see our response to the second main comment by reviewer 1). We are therefore convinced that the revision of the outlier correction and uncertainty estimation procedures allows now for substantially more convincing conclusions regarding recent thinning accelerations and differences between debris-covered and debris-free glaciers.

**2. Data and Method section. The authors spend more than 1/3 of the manuscript for Data and Method section. This section is suffered from too much detailed explanations on how to reject outliers and estimate uncertainty. All details are given, but hard to understand the reasoning of each process. First, I suggest the author to move these details to the supplement, and describe in the main text only essence of the techniques in an understandable way. Second, the structure of the section should be reconsidered. It can be something like, 3.1. Satellite data, 3.2. DEM (generation, differencing, processing, uncertainty), 3.3. delineation, 3.4. velocity.**

We agree with the reviewer that the outlier removal procedure is complex and the presentation of the method takes substantial manuscript space. The selection of maps for the ensemble was based on four different data quality proxies (% data available after outlier correction at the grid scale, sensitivity to outlier correction, triangulation residuals, and mean off-glacier elevation differences). Although all these proxies have already been applied in previous studies for quality assessments, perfectly objective criteria are not available to decide whether a map should be included in the ensemble or not. We assume this is the reason why the reviewer states that it is "hard to understand the reasoning of each process".

We have therefore decided to considerably simplify the procedure for the selection of maps. We now simply select all $\Delta h/\Delta t$ maps from the period 2006-2015 that cover periods of four years or longer (see detailed explanations under 'New Methods' and 'Data selection' at the beginning of this document). Short periods will be discarded from the beginning, since uncertainties increase with shorter time

intervals (due to lower signal to noise ratios). Following the advice of the reviewer, details about the quality proxies will be transferred to the Supplement, since these proxies are now not used anymore as criteria for outlier detection. We will restructure the Methods section (Section 3) as suggested by the reviewer.

**3. Influence of the earthquake It is interesting and important to evaluate the impact of the earthquake on the glacier surface elevation. However, the elevation change due to the earthquake in 2015 is essentially different from those occurred from 1974 to 2014. Accordingly, elevation change from 1974 to 2015 (e.g. Table 5 and Figure 6b) is not suitable to discuss recent glacier changes in general. Therefore, I suggest the author to separate the elevation change after the earthquake from the rest of the study period.**

We agree that elevation changes after the earthquake are locally very different from those during the period before the earthquake. For this reason, the May 2015 SPOT7 DEM has not been used to assess long-term glacier changes. However, we also concluded from our analysis that "Over periods of several years, the effect of the post-earthquake avalanches on the altitudinal thinning profiles such as presented in Figure 8 is only minor." (P. 23, lines 6-8). The long-term effect of the avalanches is lower than the ensemble uncertainty. The deposited avalanche volume of 0.52 m (Table 6) divided by $\Delta t = 4$ years equals 0.14 m a$^{-1}$, which is less than the ensemble uncertainty in average thinning of debris-covered ice ($\pm$ 0.18 m a$^{-1}$). Excluding the Oct 2015 DEM would lead to higher and not lower ensemble uncertainties. We will state all this clearly in the revised manuscript.

Almost 90% of the avalanche debris remaining in October 2015 accumulated on Lirung and Langtang glacier tongues, but also regarding these two glaciers the avalanche effect is minor in comparison to the ensemble uncertainty. To illustrate this, the short-term effects of the post-earthquake avalanches (April 2014 – Oct 2015 elevation changes) are now shown in the revised Figure 8 and can be compared to the elevation changes 2006 - Oct 2015 and 2009 - Oct 2015. It becomes clear that in contrary to Apr 2014 – Oct 2015, the periods 2006 - Oct 2015 and 2009 - Oct 2015 are not outliers in comparison to other recent periods.

For the reasons stated above we therefore still plan to use the October 2015 SPOT7 scene to discuss glacier changes in general (e.g. Figures 7-9). However, we replaced the differential DEM on Figure 6b by a map showing only pre-earthquake elevation changes (see revised Figure 6b). Note that Table 5 already only reported pre-earthquake elevation changes (2006 – February 2015, which was before the earthquake in April 2015).

**4. Text. I understand that the author tried to be careful and accurate in the text. However, the manuscript is lengthy, redundant and diffuse at many places. This hinders reader's understanding of the methodology, important results and conclusions. Please consider to shorten and simplifies sentences throughout the manuscript.**

We agree that the text can be improved by shortening and simplifying sentences throughout the manuscript. We will revise the text as requested by the reviewer.

**Specific comments:**

**page 1, line 15: we present volume and mass changes of . . . (omit "glacier")**

Ok.

**page 1, line 22: "mass balance trends" sounds to me "surface mass balance trends". What about "mass loss trends" or "thinning trends"?**

We indeed refer to "surface mass balance trends".

**page 1, line 22: "highly non-linear" to what? elevation? time?**

We will add "spatially non-linear thinning profiles".

**page 3, line 4: What do you mean by "downslope condition"?**

We will revise the sentence as follows: "Debris-covered glaciers are also often avalanche nourished (Scherler et al., 2011), which means that *locally the surface mass balance at the tongues* may be influenced more quickly by changes in high-altitude precipitation".

**page 3, line 8: . . . present-day "surface" lowering rates. . .**

Ok. We will change the sentence as suggested.

**page 3, line 16: What is "melt due to glacier emergence velocity"?**

The sentence was not clear and we apologize for this. What we meant by the "discrepancy between thinning and melt due to glacier emergence velocity" was that melt and thinning is not the same thing, since glacier emergence has to be accounted for when comparing thinning rates to melt rates. We will state this more clearly in the revised manuscript.

**page 4, line 26: . . ., Kimoshung Glaciers. . .**

We will do this.

**page 4, line 31-32: Please consider to shorten this kind of sentences. It should be OK to write ". . . are exceeded most part of the debris-covered area (Ragettle et al., 2015). Relatively thin debris layer appears only near the equilibrium line."**

Ok. We will revise the sentences as suggested.

**page 5, line 6: a.s.l.**

Ok. We will consistently use a.s.l. instead of asl

**page 7, line 9: ALOS PRISM**

Ok.

**page 8, line 5: What is "correlation score"?**

During the automatic DEM extraction, image correlation is used to extract matching pixels in two overlapping images. The correlation score indicates if pixels have been matched successfully. We will refer to Pieczonka et al. (2011) here.

**page 8, line 9: Either of "older" or "earlier acquisition date" is fine.**

Ok. We will simply state 'the older DEM'.

**page 9, line 12-13: I understand that these parameters are useful to measure spatialnon-uniformity in the melt rate. However, I do not understand why you use both of them. Particularly, the second one needs a reason why you take 50% and 10%. Moreover, why not using the information on cliffs and lakes delineated from the satellite image (Figure 2b)?**

This was a good comment and we have revised our approach accordingly. Quality-checked cliff and lake inventories are now available for all years for which we have DEMs. Those inventories will be used in the revised manuscript to directly relate cliff/lake area to local thinning rates. The two cliff proxies will not be used anymore.

**page 10, line 16: I wonder why "higher accuracy" can be the reason to apply the higher threshold.**

We show that the standard deviations ($\sigma$) are lower over flat and non-snow covered terrain (Figure 3 of the original manuscript). In the accumulation areas, on the other hand, the presence of many outliers leads to higher standard deviations in the elevation differences. Accordingly, with a $3\sigma$-threshold we identify outliers at debris-covered terrain, whereas over featureless and steep terrain a lower threshold is necessary for efficient outlier detection.

**page 11, line 11: Why do you use the thinning rate from 2006 to 2015 as a threshold?**

The thinning rate 2006-2015 was used as a threshold to guarantee that the signal to noise ratio is higher than 1:1, which indicates more signal than noise (whereas the thinning rate 2006-2015 is the signal and the DEM adjustment uncertainty is the noise). However, the outlier criterion discussed here will not be used anymore in the revised manuscript and the selection of $\Delta h/\Delta t$ maps for the ensemble will be based on simple and straightforward criteria (see 'General Revisions' above).

**page 11, line 18-page 12, line 18: It is hard to understand the concept and the procedure to obtain U_cadj. If this is a commonly used parameter, please provide a good reference. I recommend the author to describe this kind of details in supplement.**

To compute triangulation residuals is a quite common approach for differential DEM quality assessments (e.g. Paul et al. 2015). However, we agree that the presentation of the method is lengthy and the results are not much discussed in the paper. Since we are not using this quality proxy anymore for outlier detection we will move this part to the Supplement as suggested by the reviewer.

**page 13, line 12-13: I wonder how these thresholds were chosen and why they "effectively minimize the uncertainty".**

These thresholds minimized uncertainty because they allowed detecting outliers regarding mean elevation differences (MED). However, these thresholds and MED will not be used anymore for outlier detection in the revised manuscript.

**page 14, line 5: Using three characters as a symbol is not common. By the way, do you need to define this symbol "unc"?**

We will now use $E_{\Delta h}$ following Gardelle et al. (2013)

**page 14, line 14: Do you use the same density in the accumulation area?**

Yes. This is a common assumption in geodetic mass balance studies (e.g. Bolch et al., 2011, Gardelle et al. 2013). We will state this more clearly in the revised manuscript.

**page 15, line 12: Should be "92 maps were removed because they FULFIL outlier criteria"?**

Ok. This text is now redundant since we do not perform an outlier correction at the glacier scale anymore.

**page 15, line 22: Define the acronym "RPC".**

The acronym indicates Rational Polynomial Coefficients and the reviewer is correct that it needs to be spelled out. We will do this in the revised manuscript.

**page 17, line 5: Please be consistent with the unit, m/a or m a-1.**

Agree. We will use m a$^{-1}$

**page 17, line 29-30: "but a majority of values suggest that . . .." » This is not very sure from the data. It appears to me that the thinning rate is decreasing recently**

After applying the new outlier correction and uncertainty estimation procedures the data is now less ambiguous about mean thinning rates at Shalbachum Glacier (where indeed the thinning rates seem to have decreased, see new Figure 7). At Ghanna Glacier the data are still unclear. We will state that the ensemble uncertainty is too high to draw any conclusions regarding thinning trends at Ghanna Glacier.

**page 18, line 1: What do you mean by "ensemble of values"?**

This is an important point and we will explain this better in the revised manuscript. By 'ensemble of values' we mean the ensemble of observations available for overlapping periods. In the revised manuscript we will use the data ensemble available for the period 2006-2015 more systematically to identify a sound signal and narrow down uncertainty. See our general response at the beginning of this document.

**page 18, line 30: What about simplifies the sentence to "The most negative elevation change for 1974-2006 was observed at Shalbachum . . ..".**

We will revise the sentence as suggested.

**page 19: line 2-3: It makes more sense to compare 1974-2006 and 2006-2014 to eliminate the influence of the earthquake.**

Here we reference to the values reported in Table 5 where the pre-earthquake February 2015 DEM was considered. There was a mistake in the manuscript text (it should be "Comparing the two periods 1974-2006 and 2006-*Feb* 2015"). However, we think the October 2015 DEM is valuable to discuss multi-annual ($\Delta h > 4$ years) elevation changes and should not be excluded from the ensemble. Accordingly, in the revised manuscript we will mainly discuss ensemble mean results for the period 2006-Oct 2015. See our comment to the reviewers' main comment 3 above.

**page 19, line 1: m a-1 » You need a space between m and a-1.**

Agree.

**page 19, line 9: "The most important differences in mean $\Delta h/\Delta t$ values. . ." » "The greatest increase in thinning rate . . ."?**

We will revise the sentence as suggested.

**page 19, line 15-21: I find this paragraph is not necessary here. Because Figure 8 clearly shows the thinning patters, you do not need to give questionable comment on Figure 6.**

Ok. We agree and will remove the paragraph as suggested.

**page 19, line 32-page 20, line 3: This sentence is very hard to read. Please consider to rewrite it.**

We agree that the sentence was too long. We will rewrite the sentence as follows:

"The fragmentation of the tongue leads to mean thinning rates close to zero at elevation bands where a substantial part of the glacier area disappeared during a given period. This seems to have been the case at Langshisha Glacier near 4500 m a.s.l. shortly after 2006."

**page 21, line 30: "$\Delta h/\Delta t\_1974\text{-}06 - \Delta h/\Delta t\_2006\text{-}15 < -0.2$ m/a" » Is this correct? Isn't the left side positive if the thinning is accelerated?**

The reviewer is right here. It should be "$\Delta h/\Delta t\_2006\text{-}15 - \Delta h/\Delta t\_1974\text{-}06$". We prefer to use negative values for thinning accelerations. In the caption text of new Figure 10 we now state "Negative $\Delta(\Delta h/\Delta t)$ values represent thinning accelerations" (see below).

**page 22, 5.3. Impacts of the April 2014 earthquake: This is an interesting analysis. I suggest the author to use the DEM after the earthquake only for this purpose. In other words, elevation change from 1974 to 2014 should be used for the rest part of the discussion.**

We agree that the May 2015 DEM should be used only for this purpose. However, the long-term impacts of the post-earthquake avalanches ($\Delta h > 4$ years) are already negligible in October 2015 and we will thus use the October 2015 DEM also for the long-term comparison. See our statements above.

**page 22, line 2-8: This should be explained in the introduction section.**

We will comment on the post-earthquake avalanches in the introduction section as suggested by the reviewer.

**page 22, line8-21: This should be explained in the method section.**

Ok. We will follow the reviewers' advice.

**page 23, line 5: "compensated by about 50%" » What density do you assume for the avalanche debris deposition?**

Here we only discussed volume changes and no mass changes. We therefore did not make any assumption about density. However, we will add to the paper an estimation of mass change due to the post-earthquake avalanches. According to Scally and Gardner (1989) avalanche deposit density increases until the end of the ablation season to about 720 kg/m$^3$. Considering this value and a density of ice of 900 kg/m$^3$, the mass deposits compensate by about 40% for glacier mass loss during an average year.

**page 23, line 12: "Elevation changes in the debris-covered area are primarily independent of elevation (Figures 8 and 10c) as previously identified in Langtang catchment (Pellicciotti et al., 2015) and elsewhere . . ..."**

Thank you. We will revise the sentence as suggested.

**page 23, line 16: "downward-" » downglacier?**

Ok. We will replace 'downward' by 'downglacier'.

**page 24, line 1-2: Not clear where and how water pressure is elevated.**

The delivery of surface-generated meltwater to the en- and subglacial environment is the driver of raising water pressure. Likely, lake formation itself can be attributed to enhanced englacial water pressure. It is therefore sufficient to state: "Such stresses are usually not large enough to initiate open surface crevasses, but in combination with elevated water pressure due to local water inputs lead to hydrologically driven fracture propagation (hydrofracturing) and englacial conduit formation".

**page 24, line 5-6: Do you mean that thinning accelerated where ice motion is active because cliffs and lakes develops? It contradicts to my experience to observe cliffs and lakes formation on debris-covered stagnant ice.**

Yes, with the revised Figure 10 we can show clearly that especially cliffs appear more frequently where the glacier is not stagnant. This is not contradictory to previous studies. Several studies have shown that ice cliffs on depris-covered glaciers in the Himalaya appear most frequently in the transition zone between the active and inactive glacier parts (Sakai et al., 2002; Bolch et al., 2008; Thompson et al., 2016). The appearance of supraglacial lakes, on the other hand, is strongly related to the surface gradient. Large supraglacial lakes can only form where the slope is less than 2° (Reynolds, 2000), and the largest supraglacial lakes in the Himalaya form near the terminus of glaciers where a terminal moraine prevents free drainage of meltwater (Benn et al., 2012). The large debris-covered glaciers in the Upper Langtang catchment, however, have not reached this regime yet.

**page 24, line 20: "glacier uplift" » do you mean "ice thickening due to compressive flow regime"?**

Yes. Uplift of ice occurs by convergence of the ice flux. However, we will use 'ice thickening due to compressive flow regime' in the revised manuscript.

**page 23, line 27-page 25, line 11: The goal of this section is not clear. It appears that this section discusses the mechanism of surface elevation change on debris-covered ice. However, the thinning rate is highly variable in space and time, and there is no general trend in the observed glaciers. What kind of results does the author try to explain here? Many processes related to surface elevation change of debris-covered glaciers are described, but none of them are connected to reliable interpretation of the data. Describe first an observational fact that you want to discuss, and interpret the observation in a logical manner.**

This is a good comment, and we will substantially revise this section on the basis of the reviewers' comment. This entire section was based on the results presented in Figure 10. With the scatterplots and correlation coefficients we attempted to explain the variance in thinning rate changes of all debris-covered glaciers at the same time. However, single variables cannot explain all the spatial variation in thinning rates, and therefore the r values were generally low (max. 0.55 for debris-covered glaciers). We therefore revised Figure 10 and now show the most important variables (cliff area, lake area, surface velocity and changes in thinning) separately for each glacier. The old Figure 10 will be removed.

We now use a *new dataset of cliff and lake areas*, which substitutes the cliff/lake proxies. The new Figure 10 allows for a more reliable interpretation of the data. The main 'observational facts' that we will discuss are i) the relation between thinning acceleration and active glacier dynamics (given by the glacier velocity) at all debris-covered glaciers except Lirung, and ii) the relation between supraglacial cliff area and thinning acceleration. The latter is not always clear (e.g. at Langtang Glacier), but where thinning accelerations do not correspond to high cliff area percentages, this can be explained by thinner debris or avalanche accumulation (more details regarding will be provided in a revised manuscript). Overall, the correlation between mean thinning rate changes per debris-covered tongue

and percent cliff area per tongue is very high (r=-0.77). Regarding the spatial patterns on debris-free glaciers it is sufficient to state that thinning accelerations are elevation dependent and refer to Figure 9.

**page 25, line 16-17: What do you mean by "correlate with"? Which data show this?**

What we meant here is that one can expect glacier mass balances near steady state if the glacier area remains nearly constant, as it is the case for Kimoshung Glacier. We will revise the manuscript text to be clearer.

**page 25, line 13-page 26, line 9: The first part of this section 7.1.1. explains that thinning accelerated at Yala Glacier, whereas it appears to be at a similar level at Kimoshung Glacier. This kind of explanation should be completed in Result section. Interpretation on the difference begins at page 25, line 28, but not convincing because there is no qualitative discussion. For example, hypsometry is not shown for Kimoshung Glacier, and no information about the 0 degree C isotherm altitude.**

The first part of Section 7.1.1. will be completely removed from the discussion section. Areal changes will be presented in a separate results section, where it will also be stated that the observed areal changes correlate strongly with mean elevation changes (now also for the period 1974-2006, due to the better outlier correction in the accumulation area of Kimoshung Glacier).

Regarding the second part, we guess the reviewer wanted to say that a quantitative discussion was missing (since the discussion was essentially qualitative). In the revised manuscript we will thus state that variations in the estimated equilibrium line altitude (5400 m a.s.l.) of ± 100 m lead to an AAR variation of 13%-70% at Yala Glacier, but at Kimoshung Glacier only to a variation of 80%-88%. This explains why Yala Glacier is much more sensitive to warming than Kimoshung Glacier. Consequently, thinning at Yala Glacier accelerated from -0.32 ± 0.06 m a$^{-1}$ (1974-2006) to -0.88 ± 0.23 m a$^{-1}$ (ensemble mean 2006-2015), while it accelerated only insignificantly at Kimoshung Glacier from +0.06 ± 0.13 m a$^{-1}$ to -0.02 ± 0.17 m a$^{-1}$ (see revised Figure 7). Hypsometry of Kimoshung Glacier is not provided in the paper. We think that the AAR sensitivities speak for themselves. However, if requested, we can add Figure R1 below to the Supplement.

[Figure]

**Figure R1. Elevation distribution of Kimoshung and Yala Glacier.**

**page 26, line 11-page 27, line 11: This section has the same problem as section 7.1.1. The first paragraph describes several different observations. These details should be explained in Result section, and here the focus of the discussion should be stated briefly. In the second paragraph, speculative conclusions are given without detailed/quantitative comparison with the modeling work.**

In the revised manuscript we will present areal changes in the result section, as suggested by the reviewer. Our conclusions regarding differences between debris-covered glaciers will be more robust, since the new Figure 10 depicts these differences more clearly. In the revised text we will also provide quantitative details (e.g. regarding the differences in average surface velocity or cliff and lake area) which will also improve the comparison with the modeling work.

**page 27, line 18-20: Not clear what are compared in Figure 6b.**

We referred to Figure 6b because the figure shows also the thinning rates of Kimoshung Glacier. However, we will remove the comparison of thinning rates of Kimoshung Glacier and debris-covered area, and focus on the comparison with Yala Glacier. Yala Glacier has an AAR of 40% (Table 1), which is identical to the mean AAR indicated by Kääb et al. (2012) for the entire HKH region and is suitable as a reference. Kimoshung Glacier, on the other hand, has a very steep tongue that reaches to low elevations (terminus at 4385 m, according to Table 1). The average slope of this tongue is 33%, whereas 95% of the debris-covered areas have a slope of less than 18%. A comparison of thinning rates is therefore not meaningful.

**page 27, line 24-25: Not clear why you compare the elevation of Yala terminus and that of maximum thinning on Langtang.**

We apologize if this was not clear. The main point here is to compare lowering rates of debris-free and debris-covered glacier area at the same altitudinal range. Since debris-covered glaciers reach much lower elevations, a comparison is only possible from 5150 m a.s.l. upwards. The elevation of maximum thinning on Langtang Glacier coincides with Yala terminus elevation, but this is just a coincidence and should not be our main point here. We will revise the text to be clear about this. See also our response to the next comment below.

**page 27, line 26-28: It is not clear which part of the elevation range is compared here. If you discuss elevation change of debris-covered and debris-free glaciers at same elevation range, why not preparing a plot for this purpose?**

Here we compare the thinning rates near the terminus of Yala Glacier (5150-5200 m a.s.l.) to the thinning rates at Langtang Glacier at the same elevation. We will explain this better in the revised manuscript. If the reviewer asks for it we can also add Figure R2 below to the Supplement (comparing thinning profiles of Yala Glacier to thinning profiles of Langtang Glacier tongue). However, the same information is provided by Figure 8. We therefore do not think an additional figure is required here.

[Figure]

**Figure R2. Elevation change profiles of Langtang Glacier tongue (debris-covered) and Yala Glacier (debris-free). The figure shows the ensemble-median results for the period 2006-2015, and error bars represent the ensemble-uncertainty.**

**page 28, line 3-15: The point of the discussion is unclear.**

We apologize for the lack of clarity. We will improve the paragraph by reordering the sentences, although we think the point of the discussion is clear. The first important point here is that "*average thinning of all glaciers has increased in the recent period*" (lines 5-6). Then we state that the magnitude of this thinning increase is very different from glacier to glacier ("*there are examples ... of glaciers where thinning seems to increase rapidly or where thinning remains approximately constant*", lines 8-9). Then we state that "*A main difference between debris-free and debris-covered glaciers in our sample cannot be identified*". This is why we conclude that "*both the debris-free and the debris-covered glaciers show a heterogeneous response to climate*" (lines 4-5). Finally, we suggest that the altitude distribution of glaciers likely plays a more important role for average thinning rates than debris-cover alone.

**page 28, line 16-17: Not clear what you mean. Do you mean that your result support the studies by Kaab, Nuimura and Gardelle?**

No, our results do **not** support the observations by Kääb, Nuimura and Gardelle, since our observations do not show that the lowering rates of debris-covered glacier area are similar to those of debris-free areas at the same elevation. We have stated this clearly on Page 27, lines 15-17. On the other hand, we do not observe a significant difference in the overall mass balance trends of debris-free and debris-covered glaciers (see our previous comment above). Here we want to say that these two observations are not contradicting. A comparison of thinning rates at the same elevation is not representative of average thinning rates due to the differences in altitude distribution. In the revised manuscript we will rewrite this paragraph to be clearer.

**page 30, line 17-page 32, line 14: Only a few data appear in Conclusion section, which results in very qualitative descriptions. This represents the weakness of the paper. Please draw your conclusions which are supported by data.**

The reviewer is right that some more quantitative measures should be provided in the conclusion section. A summary of quantitative outputs with respect to the three main goals is provided at the beginning of the revision statements to this reviewer. The summary shows that we are able to draw relevant conclusions supported by data.

**page 52, Figure 11: This velocity map is not much used for the study. Judging from the vectors on the plot, it is not sure how much this analysis is reliable.**

Velocity is a key variable to discuss thinning patterns on debris-covered glaciers (see revised Figure 10). We think the data are reliable, especially since the vectors consistently point in down-glacier direction. It is not clear to us how the reviewer came to a different conclusion regarding the vectors. The revised Figure 11 shows larger vectors, so we hope this is now clearer.

[revised manuscript text omitted]

**REVISED MANUSCRIPT FIGURES**

Above each figure we shortly summarize the changes to the figures. Caption texts (below each figure) have also been updated. Note that the final order of figures might be different in a revised manuscript.

Figure 1: New background image (Cartosat-1 Oct 2006 ortho-image)

[Figure]

**Figure 1. Map of the upper Langtang catchment. The numbers on the map correspond to the glaciers listed in Table 1. Monsoon snow-cover frequency is based on Landsat 1999 to 2013 land cover classifications (Miles et al., 2016b). 1974 glacier area (dotted lines) is shown for the seven studied glaciers only.**

Figure 2: New quality-checked cliff and lake inventories used to characterize 50-m elevation bands in Figure 2b).

[Figure]

**Figure 2. (a) Δh/Δt map of Langtang Glacier tongue, Oct 2006 – Nov 2009 (outlier corrected and missing data filled with inverse distance weighting), (b) Cliff area (% of 30-m pixels containing cliffs per 50m elevation band; median of 6 available cliff maps from 2006-2015). The cliffs and lakes indicated in (b) were delineated manually on the basis of the Oct 2006 orthorectified satellite image.**

Figure 3: This is a new figure. Figure 3 of the original manuscript will be shown in the Supplement of the paper. Uncertainties shown here are calculated based on Gardelle et al. (2013).

[Figure]

**Figure 3. Uncertainties in elevation change rates (Δh/Δt) in function of the time interval between DEMs (Δt). Median results of all available 28 Δh/Δt maps shown and the error bars extend to the most extreme data points. Uncertainties represent weighted mean values of the 50-m elevation band off-glacier errors shown in Figure 4. Elevation distributions of all glacier area and debris-covered glacier area, respectively, are considered to calculate uncertainties for each surface type separately.**

Figure 4: Δh/Δt off-glacier errors per 50-m elevation are now calculated following Gardelle et al. (2013). The distribution of mean Δh/Δt errors per 50-m elevation band is now shown only for the ensemble of Δh/Δt maps which are effectively considered after initial quality check (short periods with Δt < 4 years and all Δh/Δt maps involving the ALOS PRISM 2010 DEM are excluded). 95% confidence intervals are shown instead of 50% confidence intervals.

[Figure]

**Figure 4. Off-glacier elevation change error (Δh/Δt) in function of elevation. The lines represent the median error in the ensemble (1974-2006 Δh/Δt map, all Δh/Δt maps with Δt ≥ 4 years within 2006-2015 excluding the ALOS PRISM 2010 DEM) per 50-m elevation band. Error bars represent the 95% confidence interval.**

Figure 5: Δh/Δt uncertainties are now calculated based on Gardelle et al. (2013). The distribution of mean Δh/Δt errors per 50-m elevation band is now shown only for the ensemble of Δh/Δt maps which are effectively considered after initial quality check (short periods with Δt < 4 years and all Δh/Δt maps involving the ALOS PRISM 2010 DEM are excluded).

[Figure]

**Figure 5. a) Uncertainty estimates of average elevation change rates (Δh/Δt) per individual glacier and per debris-covered tongue. b) Fraction of glacier pixels remaining after removing low stereo matching scores (Figure S1) and outliers. On each box, the central mark is the median of the ensemble (1974-2006 Δh/Δt map, all Δh/Δt maps with Δt ≥ 4 years within 2006-2015 excluding the ALOS PRISM 2010 DEM) and the edges of the box are the 25th and 75th percentiles. The whiskers extend to the most extreme data points.**

Figure 6: The map in b) now shows a differential DEM involving only pre-earthquake DEMs (Cartosat-1 Oct 2006 and WorldView Feb 2015).

[Figure]

**Figure 6. Elevation change rates (Δh/Δt) derived from a) Hexagon Nov 1974 and Cartosat-1 Oct 2006 DEMs and (b) Cartosat-1 Oct 2006 and WorldView Feb 2015 DEMs.**

Figure 7: Error bounds corresponding to individual periods are now calculated based on Gardelle et al. (2013). The figure now also shows the ensemble mean for all considered periods within 2006-2015 (excluding short periods with Δt < 4 years and all Δh/Δt maps involving the ALOS PRISM 2010 and SPOT7 May 2015 DEMs) and the ensemble uncertainty 2006-2015 (calculated as described under 'New Methods' in the revision statement). In a revised paper, this figure will be divided in two individual figures.

[Figure]

**Figure 7. Mean elevation change rates (Δh/Δt) per period and glacier (top) or debris-covered area (below). Periods involving the post-earthquake May 2015 SPOT7 DEM (Section 5.3) are not considered here. For better readability, only the maximum width of error bounds corresponding to individual periods 2006-2015 are shown.**

Figure 8. Error bounds are now calculated according to Gardelle et al. (2013). Pre/post-earthquake elevation changes are now shown in green color. April 2014 – Oct 2015 elevation changes are also shown to depict the short-term effect of the post-earthquake avalanches. Over longer periods (Δt > 4 years) the effect of the post-earthquake avalanches is negligible compared to uncertainties.

[Figure]

**Figure 8. Altitudinal distribution of mean annual elevation change (Δh/Δt) over 50 m elevation bands of debris-covered tongues (debris-covered area of each glacier excluding tributary branches). Uncertainty bounds correspond to uncertainty in function of elevation derived for each Δh/Δt map individually (Figure 4). Pre/post-earthquake elevation changes are shown in green color.**

Figure 9. Error bounds are now calculated according to Gardelle et al. (2013). Ensemble median values are now shown for all elevations above the ELA (5400 m a.s.l.). Ensemble median values as shown here are used to replace missing data in the accumulation areas of glaciers (as described under 'New Methods' in the revision statement).

[Figure]

**Figure 9. Altitudinal distribution of mean annual elevation change (Δh/Δt) and altitudinal distribution of glacier area (%) over 50 m elevation bands of selected glaciers. Uncertainty bounds correspond to uncertainty in function of elevation derived for each Δh/Δt map individually (Figure 4). Ensemble median values shown here are used to replace missing data in the accumulation areas of glaciers after outlier exclusion (Section 3.4.1). Note that the x-axis ranges are different for each sub-figure.**

Figure 10. This is an entirely new figure replacing Figure 10 of the original manuscript.

[Figure]

**Figure 10. Altitudinal distribution of cliff and lake area, glacier velocity and changes in thinning rates. Cliff and lake area is shown as % of 30-m pixels containing cliffs per 50m elevation band, whereas the values represent the median of 6 available cliff and lake maps from the period 2006-2015. Glacier velocities (m/a) represent the median per 50m elevation band of data shown in Figure 11 and error bars represent the standard deviation in pixel values per elevation band. Changes in thinning rates (Δ(Δh/Δt) [m/a]) are calculated comparing 1974-2006 and 2006-2015 (ensemble-median). Negative Δ(Δh/Δt) values represent thinning accelerations. Error bars represent the maximum variations in Δ(Δh/Δt) considering all individual periods within the 2006-2015 ensemble.**

Figure 11. Areas steeper than 45° (95th percentile of glacier slopes) have been masked out from the velocity map. Black arrows indicating glacier flow directions have been enlarged and are now scaled according to surface velocity. Off-glacier velocities are now shown only in transparent color. The hillshade in the background is based on the SRTM DEM.

[Figure]

**Figure 11. Surface velocities 2009-2010 cropped to catchment boundaries. Values have units of meters per year and are derived by cross-correlation feature tracking. Black arrows indicate derived glacier flow directions and are scaled according to surface velocity. Off-glacier velocities are shown in transparent color.**

---

## Editor Comment (EC2) · E. Berthier (Editor) · 18 May 2016

Dear authors,

Thanks a lot for your point-by-point responses to the referee's comments. Your letter suggests that you may be able to address most of their comments in a revised manuscript.

Thus, I would like to ask you to submit a deeply-revised version of the manuscript that will be evaluated by the referees. Make sure that all your conclusions are firmly supported by your data.

Best regards,

Etienne Berthier

---

## Author Response (AR1)

REVISION STATEMENT

**"Heterogeneous glacier thinning patterns over the last 40 years in Langtang Himal"**

**by S. Ragettli, T. Bolch and F. Pelliciotti**

**IN RESPONSE TO THE EDITOR**

We thank the editor for his comments and suggestions. All comments by the two reviewers have been carefully considered and the entire manuscript has been deeply revised. We implemented the corrections as outlined in our response to the reviewers in the interactive discussion. In our statements below we specify clearly what we have changed in the revised manuscript and where. Some of the detailed comments have been addressed differently than announced in the interactive discussion and we have updated the revision statement accordingly:

- Rev. 1, Page 8, line 4; Rev. 2, page 25, line 13-page 26, line 9: a new figure in the Supplement (Figure S5) compares the hypsometry of Yala and Kimoshung Glacier, which depicts why the two glaciers have such different AARs.
- Rev. 1, Page 9, lines 2-5: we have added a new figure to the Supplement (Figure S2) which compares the glacier outlines for Langshisha Glacier used by the present study and by Pellicciotti et al. (2015).
- Rev. 1, Page 11, line 18; Rev. 2, Data and Method section: to shorten the manuscript we have removed all content on triangulation residuals.
- Rev. 1, Page 43, Table 7: we have added uncertainty estimates to the revised Table 7 (glacier area changes).
- Rev. 2, 3. Influence of the earthquake: We now assess more carefully in Section 4.1 ("Impacts of the April 2015 earthquake") which of the post-earthquake DEMs can be considered to discuss recent glacier changes.
- Rev. 2, page 26, line 11-page 27, line 11: We merged the previous section 7.1.2. with the new Section 5.1. We have removed all statements that cannot be firmly supported by our data.

Together with this document we submit also a marked-up manuscript version showing the changes made. From this, it is evident that all sections have been deeply revised. We hope that the effort in the revision of the original material will be recognized and that the manuscript is now acceptable for publication in TC.

**GENERAL REVISIONS**

We would like to thank very much the two reviewers for their thorough and detailed comments. We have addressed all the reviewers concerns in our detailed point by point answers below. Both referees raised important methodological issues, mostly regarding outlier exclusion and uncertainty quantification. We agree that a revision of our methods was necessary. We have thus substantially modified some of our procedures and we have recalculated all thinning rates and mass balances. Nearly all figures have been modified as a result of these changes.

The major issues in the revision were: 1) to revise the outlier removal procedure to make it simpler and based on logical criteria (reviewer 1 and 2), 2) to increase the reliability of the data by addressing the unusually large thickening and thinning patterns in the accumulation areas (reviewers 1 and 2), 3) to redo the uncertainty analysis based on approaches that are established in the literature (reviewer 1), 4) to improve the manuscript by reducing its length, make it more focused and by drawing only

conclusions based on reliable data (reviewers 1 and 2), 5) to separate the elevation changes after the earthquake from the rest of the study period (reviewer 2).

As a result of changes in response to the reviewers' comments, these are the main changes in our study:

1. We now use a simpler and more straightforward procedure for the selection of data for our ensemble approach.
2. We have revised our approach for outlier removal at the grid scale. The approach does not fail anymore to identify erroneous patterns in the accumulation areas of glaciers.
3. We use a new approach for uncertainty estimation based on Gardelle et al. (2013).
4a. We use a new dataset of supraglacial cliff and lake inventories to explain the observed thinning patterns instead of the proxies previously used in the submitted manuscript; these new data sets allow to directly relate spatial thickness change patterns to observations of glacier surface characteristics.
4b. We increased the readability and sharpened the focus of the manuscript: the 'Methods' section has been shortened and the separate section on outliers and uncertainty has been removed.
4c. Generally, figures and text now better emphasize the added value of using an ensemble approach, which we are convinced is a novelty of this paper but has not been recognized (or only partly) by the reviewers. The availability of multiple independent DEM differencing results for overlapping periods allows identifying a sound signal and narrowing down the uncertainty of recent volume changes.
5. We show that over longer periods ($\Delta t > 4$ years) the effect of the post-earthquake avalanches six months after the earthquake (in April 2015) is negligible compared to the uncertainties in calculated elevation changes (Section 4.1, Figure 8). The October 2015 DEM is thus still used to assess long term elevation changes, whereas the May 2015 DEM is not.

We are convinced that the changes in methodology and structure of the paper have benefited the papers' quality. They have also strengthened the novelty and relevance of the results. We are now able to unambiguously identify for which glaciers overall thinning has accelerated in recent periods, or where thinning has remained approximately constant. The new dataset on cliff and lake area that reviewer 2 suggested to include provides valuable insights regarding the mechanisms that lead to spatially heterogeneous thinning patterns on debris-covered glaciers. The more reliable results for the accumulation areas allow now for more convincing conclusions regarding the differences in glacier response to climate. None of our main results or conclusions however changed significantly, thus supporting the methodological choices made originally. Overall, we strongly believe that our study now indeed represents one of the most rigorous documentations to date of glacier response to climate change over the last 40 years in the Himalaya.

Since some of the comments were common to both reviewers, we describe below the major changes made and refer to those in our detailed responses to the reviewers.

**NEW METHODS**

**1. Data selection** (Section 3.2.5, previously 'Outlier detection at the catchment and glacier scale', Sections 3.4.2 and 3.4.3): Instead of selecting the data based on four different data quality proxies (% data available after outlier correction at the grid scale, sensitivity to outlier correction, triangulation residuals, and mean off-glacier elevation differences) we now simply select all $\Delta h / \Delta t$ maps from the period 2006-2015 that cover periods of four years or longer. Shorter periods are discarded and not discussed in the paper, except for assessing the short-term effect of post-earthquake avalanches. This change responds to the reviewers comments about simplifying our methods, which we agree with.

There are two main reasons to discard short periods and focus only on multi-annual periods within 2006-2015 instead:

- Uncertainties substantially decrease for longer periods (see new Figure 5).
- Data from overlapping periods within 2006-2015 provide a range of plausible values for this period, especially as the elevation changes between the different selected periods show very similar characteristics (Figures 8 and 11). This data ensemble allows narrowing down the uncertainty: the uncertainty in a sample mean is lower than the uncertainty in individual estimates (according to standard principles of error propagation). We thus obtain an ensemble of results for the relatively short period 2006-2015 that can be used to assess changes in thinning rates with respect to the longer period 1974-2006 in which much larger absolute elevation changes occurred.

The ALOS 2010 scene is excluded from the ensemble since we can show that the uncertainties of the corresponding Δh/Δt maps are 20-80% higher than the uncertainties calculated for other maps (see revised Table 3; we are referring here to the new uncertainties calculated with the approach based on Gardelle et al. 2013, see point 3. below). The post-earthquake scene from May 2015 is also excluded from the 2006-2015 ensemble since elevation changes are essentially different due to the post-earthquake avalanches. The October 2015 DEM is still considered for the ensemble, since a larger part of the avalanche deposits melted already and, hence, the avalanche effect on multi-annual glacier volume changes is minor in comparison to the ensemble uncertainty (see our detailed answer to the third major comment by reviewer 2).

**2. Outlier correction** (Section 3.2.3, previously 'Outlier detection at the grid scale', Section 3.4.1): We revised our outlier correction by narrowing the range of acceptable values for the *accumulation* areas and now use a 1σ threshold to identify outliers. Pixels are thus defined as outliers when the absolute elevation differences differ by more than one standard deviation (considering all elevation differences within glacier area located above the ELA). Below the ELA we now use a 3σ level (instead of a 2σ level for debris-free terrain as in the original manuscript), following Gardelle et al. (2013). We are aware that ELA estimates are uncertain (see our response to the comment on Page 8, line 4, by reviewer 1), and thus we assess the sensitivity of our results to an ELA uncertainty of ±100 m (Section 4.2.1, Table 6).

The application of more restrictive criteria for plausible elevation change values in the accumulation areas required also a revised procedure for gap filling, because gaps tend to be quite large when using 1-sigma thresholds. We now first calculate the mean elevation change rates per 100-m elevation band of each glacier and then calculate the median of the ensemble (see revised Figure 11). This value is then used to replace outliers from a given elevation band in the accumulation area. For the ablation areas we still use inverse distance weighting (IDW) for gap filling, since gaps are very small and the variability in plausible values is high. This procedure led to more realistic values especially in the accumulation areas of the period 1974-2006. Previous geodetic studies have used glaciological expert knowledge for outlier removal and gap filling in the accumulation areas (e.g. Pieczonka et al., 2013; Pieczonka and Bolch, 2015), considering that elevation changes in the accumulation areas are minor over periods of several years (e.g. Schwitter and Raymond, 1993; Huss et al., 2010). Since we now only consider time intervals between DEMs that are longer than 4 years we think it is justified to use empirical values from the same glacier to fill data gaps in the accumulation areas, even if data from very different periods are used to calculate ensemble-median values.

The revised outlier correction now detects obviously erroneous data in the accumulation areas of the 1974-2006 map (see revised Figure 4a).

**3. Uncertainty quantification** (Section 3.2.4): Our new uncertainty estimates are based on the standard error calculated per elevation band as in Gardelle et al. (2013). Accordingly, we now take into account the number of independent pixels per elevation band. The distance of spatial autocorrelation for each $\Delta h/\Delta t$ map is calculated considering the range of the semivariogramm of all off-glacier elevation differences (e.g Magnússon et al., 2016). We identified distances between 260 and 730 m (average of all $\Delta h/\Delta t$ maps: 495 m). Weighted mean uncertainty values per glacier are then calculated as in the original manuscript by taking into account the altitudinal distribution of uncertainty and glacier hypsometry.

Our previous approach took into account both the mean elevation differences (MED) and the standard error (SE). The large uncertainties obtained for the 1974-2006 map were related to the erroneous elevation change patterns that were due to errors in the Hexagon 1974 DEM. However, both reviewers suggested discarding all unrealistic elevation changes in the accumulation areas. To account for the MED is therefore not necessary anymore, since deviations from zero are prevented by the more restrictive outlier definitions. Accordingly, the uncertainty estimates for the 1974-2006 map are now much lower (revised figures 9 and 10). This facilitates the interpretation of results shown by figures 9 and 10 and allows for stronger conclusions regarding the differences between 1974-2006 and 2006-2015.

Ensemble-mean and ensemble-uncertainty values are now provided in Table 5 for the period 2006-2015. 'Ensemble uncertainty' is defined as the standard deviation in ensemble values for 2006-2015 multiplied by 1.96 (p.11, lines 15-21). Standard deviation is commonly interpreted as 68% confidence level assuming normal error distribution. By multiplication with 1.96 we obtain 95% confidence levels.

**4. Supraglacial cliffs/lakes** (Section 3.3): We now use six quality checked maps of cliffs and lakes from each available satellite image for the period 2006-2015 (2006, 2009, 2010, 2014, May and October 2015). These inventories are used to calculate cliff and lake area per elevation band, and replace the statistical proxies ($\sigma \Delta h/\Delta t$, $\Delta h/\Delta t$ Q50-Q10). Since both reviewers criticized that only limited conclusions are possible from the original Figure 10, we have replaced this figure by elevation profiles showing ensemble-mean thinning rate changes ($\Delta \Delta h/\Delta t$), surface velocities and lake/cliff area (Figure 12).

The new dataset allows for interesting and convincing conclusions regarding the mechanisms that lead to spatially heterogeneous thinning patterns on debris-covered glaciers. Correlations between the presence of cliffs and accelerations in local thinning are evident from the revised Figure 12.

**RESPONSE TO REVIEWER 1**

**General comments: In the paper Heterogeneous glacier thinning patterns over the last 40 years in Langtang Himal an interesting set of geodetic data from various sources is presented and used to infer the geodetic mass balance of the Langtang catchment in the Nepal part of the Himalayas focusing mostly on two periods (1974-2006 and 2006-2015). The authors undergo complex automatic classification of their data archive to sort out what they consider as reliable data for geodetic mass balance calculation as well as applying partly new approach to estimate the uncertainty. Unfortunately this work is not completed and is still some way from being scientifically sound.**

We would like to thank E. Magnusson for his very useful review. His suggestions for improvement of our outlier removal and uncertainty estimation procedures helped us to considerably increase the robustness of our conclusions. We believe that our results are now scientifically sound, in the sense that now only reliable data are used and that we can show that identified variations in glacier thinning are significant. The revision of the methods led to a more robust statistical assessment of our main results.

However, we also would like to note that the changes in outlier removal and uncertainty estimation procedures did not lead to different conclusions compared to the original manuscript. As in the original manuscript, our results depict a heterogeneous response of glaciers to climate, with a strong spatio-temporal variability of thinning trends at debris-covered tongues and clear evidence about the crucial role of glacier hypsometry for mean thinning trends. We also would like to note that the reviewer ignored some key novel aspects of our work, such as using an ensemble of independent observations to narrow down the uncertainty of our results. We were puzzled by his proposition in one of his detailed comments below on Section 4 to use less data (only three DEMs). To us it is obvious that several independent measurements (differential DEMs for the period 2006-2015) lead to higher confidence in detected signals. To make this clearer we now provide ensemble-mean and ensemble-uncertainty values (Table 5), as described at the beginning of this revision statement under 'New Methods'. Overall, we are however thankful for all the reviewers' detailed comments since they helped us to understand where the methods needed to be improved and where the advantages of a given approach needed to be clarified.

**1) The logic behind the complex outlier removal is often difficult to understand, and various steps in it are poorly justified. Despite this complex automatic outlier removal it seems to fail at many locations when looked at the difference maps of the 1974-2006 (Figure 6a). If the authors belief that the accumulation areas of glaciers thinned or thickened by 60-100 m at many locations as the this figure indicates they need to come up with some logical and justified explanation why (surges, enormous avalanches?) and they also need to explain the absence of this pattern of extreme thickening and thinning in the accumulation area of the glaciers for the period 2006-2015 (Figure 6b).**

We agree with the reviewer that the outlier removal procedure used in the original paper was complex and the presentation of the method takes substantial manuscript space. We also agree that the threshold criteria are sometimes difficult to justify, but this is because perfectly objective criteria are not available.

Regarding the 1974-2006 elevation difference map it is true that outliers in the accumulation areas remained and therefore the outlier removal procedure failed for these areas. We do not believe that thinning or thickening values of 60-100 m are plausible in the accumulation areas. This was stated clearly in the manuscript (p. 17, lines 10-12: "*The Nov 1974 - Oct 2006 Δh/Δt map (Figure 6a) reveals*

*an irregular and unrealistic distribution of Δh/Δt values at high altitudes, which can be likely associated to errors in the Hexagon 1974 DEM*", p. 25, lines 25-26: "*Presumably, unrealistically high thinning rates at high altitudes due to errors in the Hexagon 1974 DEM led to this result*"). It is therefore not necessary to explain the absence of these patterns in other maps, since we clearly say that those unrealistic values are due to errors in the Hexagon map and thus the absence of errors in the other maps is due to better data quality.

However, we have carefully revised our outlier removal procedure to address the reviewers' concern and following his suggestions. The revised outlier detection algorithm now identifies unrealistic patterns in the accumulation areas. Missing data in the accumulation areas are replaced by plausible values. All new methods are described above under 'New Methods' and 'Outlier correction'.

The procedure for the selection of an ensemble of maps from the 28 available elevation change maps has been substantially simplified (see 'New Methods' and 'Data selection' above), even though we retain the ensemble as we think this approach provides a valuable estimate of uncertainty and sounder signal. The main focus of the paper now is, as suggested by this reviewer, on the comparison of the periods 1974-2006 and 2006-2015, and for the second period we consider a number of overlapping periods that allow narrowing down the uncertainty of volume change during this period.

**2) The explanation on how the uncertainty is calculated is not very clear and the procedure seems vague from statistical point of view. It is therefore hard to obtain any sense for its actual meaning. The authors do not even attempt to guess what confidence level it may represent. If more simple approaches such as using the e.g. standard deviation of off glacier DEM difference as proxy for the volume change uncertainty, it is at least known that such proxy is likely to result in very conservative uncertainty estimate compared to more advanced methods as shown by several studies.**

Our uncertainty calculations were based on the approach used in Bolch et al. (2011), a published and established approach (used e.g. also in Thompson et al., 2016, JG). We summed quadratically the mean off-glacier elevation differences (MED) and the standard error (SE) (eq. 5). We noticed that the uncertainties increase with altitude (which is common for geodetic elevation changes in the Himalaya, e.g. Nuimura et al., 2011). This is partly due to the fact that higher elevations tend to have steeper slopes and it is well know that the accuracy of DEMs derived from stereo data decreases with increasing slope. Therefore, we first calculated the uncertainty for each elevation band independently and then calculated a weighted average per glacier by taking into account glacier hypsometry. This resulted in conservative uncertainty estimates (i.e. large uncertainties) since both the mean error and the standard deviation were taken into account and we assumed no error compensation across elevation bands. However, when revising our approach we identified the following issues:

- The spatial autocorrelation of the error was insufficiently accounted for by considering $n$ in eq. 4 equal to the number of pixels per elevation band (as explained on p. 14, lines 6-13) and not the number of independent pixels per elevation band (such as in Gardelle et al. 2013).
- Summing quadratically MED and SE led to very high uncertainty estimates at high altitudes, especially where the Hexagon 1974 DEM was used (because of errors in this DEM over snow-covered surfaces). This led to uncertainty ranges which suggested that even positive glacier mass balances are possible for debris-covered glaciers such as Langtang or Langshisha, although at the glacier tongues we unambiguously identified strong surface lowering (figures 8 and 11 of the manuscript).

Following the reviewer's remark, we therefore used a different approach for uncertainty estimation. The new uncertainty estimates are based on the standard error calculated per elevation band as in Gardelle et al. (2013).

To use just a crude proxy such as the standard deviation of the off-glacier elevation differences, however, seems not appropriate. This would imply assuming that the DEM errors at all locations are totally correlated, which we know is not the case. The standard error (thus the standard deviation of the sample-mean's estimate of the error) can be interpreted as 68% confidence level assuming normal distribution. Since we are assuming no error compensation across elevation bands the confidence level in our uncertainty estimates per glacier is higher than 68%. This is now stated in the revised manuscript (p.11, lines 11-14).

**3) The most critical weakness of this work is however that the authors seems neglect almost completely the uncertainty they actually obtain when discussing their results. A large proportion of the paper is spent on discussion on the temporal and spatial variation of the geodetic mass balance, while in most cases the variation they are discussing are not at all or barely significant if one believes the uncertainties obtained for the discussed values.**

We kindly disagree with the reviewer here. There are two main reasons for this. First, uncertainties and outliers were discussed in detail in the manuscript. Indeed, we devoted a whole section to this (previous Section 4). In the Results section the uncertainties were always provided when presenting mass balance values or mean thinning rates. If necessary, uncertainties were addressed explicitly when discussing results (e.g. original manuscript p. 18, lines 3-5; p. 18, lines 23-25; p. 20, lines 5-12; p. 21, lines 21-23; p. 25, lines 23-27).

Second, the reviewer does not take into account that we use an ensemble of independent measurements, which allows constraining the uncertainty of individual Δh/Δt maps. This was stated in the original manuscript (e.g. at page 18, lines 1-3, "*The ensemble of values helps to distinguish between trends that should be classified as uncertain… from trends that are consistent within the ensemble*"). To us it was clear that the ensemble of values, available for overlapping periods, is an asset of this study. Now we understand that the advantages of the ensemble approach might not have been clear and needed to be stressed more. We do this in the revised figures 9 and 10 and we emphasize this point in the text of the revised manuscript (e.g. in the Introduction on p. 4, lines 1-6, or in Section 3.2.5).

We also point here to the fact that our results regarding the spatial and temporal variations in elevation change rates in the ablation areas are affected by very low uncertainties. Consequently, a large part of the Results and Discussion sections are devoted to discuss the spatio-temporal patterns in the ablation areas. We are surprised that the reviewer does not mention this here and instead suggests that most our results are affected by high uncertainty, although he agrees that "the data on the debris covered glaciers is the most convincing part of this manuscript." (see his comment below about Page 44, Figure 2).

**My main advices for the authors are the following:**

**a) Revise how you do your outlier removal, ideally make it more simple and if not make it such that the logic behind is understandable. It is also OK to use common sense when doing the outlier removal, instead of counting entirely on automatic outlier removal (this is presumably the difference between this work and the study of Pellicciotti et al. (2015) where part of the 1974 DEM of the accumulation area of Langshisha glacier was considered as erroneous data and therefore rejected).**

We have simplified the outlier removal, especially regarding the selection of maps for the ensemble which was based on four different data quality proxies. In the revised paper, we simply select only Δh/Δt maps from the period 2006-2015 that cover periods of four years or longer (Section 3.2.5).

Regarding the grid scale outlier correction it is necessary to define clear criteria to prevent arbitrary or subjective choices. Note that also in Pellicciotti et al. (2015) outlier detection was based on an automatic algorithm, but in the case of Langshisha Glacier the threshold of acceptable elevation changes was lower (see our response below to the reviewers' comment on Page 29, lines 6-10). Accordingly, we revised our outlier correction by narrowing the range of acceptable values for the accumulation areas. Detailed explanations are provided in Section 3.2.3.

**b) Redo your uncertainty analysis. I would use approaches suggested by others unless you can better justify your approach and at least give the reader any evidence that the assumption you make when carrying out your uncertainty analysis is likely to result in an overestimate of your uncertainty rather than underestimate. You also need to be able to clarify what you mean by your uncertainty in terms of confidence level do give your uncertainty any meaning.**

We have followed the reviewer's advice and now use an approach that is more established in the literature (see detailed explanations above under 'New Methods' and 'Outlier correction'). We now state in the revised manuscript that the estimated confidence level of our uncertainty values is higher than 68% (see our answer to the reviewers' second main point above).

Finally, we now also provide the ensemble uncertainties (procedure summarized above and described in Section 3.2.4.). The variability in the ensemble of values extracted for overlapping periods is a better indicator for the actual uncertainty in the values identified for the period 2006-2015.

**c) When the above has been done, carefully revise what your data actually tells you with any confidence. This could lead to a good concise paper if carried out in the above suggested manner.**

We have done all of the above in terms of methodology, and have revised the paper accordingly. With the improvements in our procedures for outlier correction and uncertainty analysis it is possible to clearly identify changes in mean thinning rates over time (see revised figures 9 and 10). The Δh/Δt glacier profiles (Figures 8 and 11) allow identifying unambiguously where in the ablation areas thinning has accelerated.

While it is true that the paper is more concise now as a result of the simplifications suggested by the reviewer, our main results however have not changed.

In addition, with a new dataset of cliff and lake areas (see our answers to the second reviewer) it is possible to directly relate spatial patterns of change to glacier surface characteristics (see the revised Figure 12).

Indeed, we think the suggestions by the two reviewers have helped us to present more concise and interesting results.

**Specific comments:**

**The list of the specific comments on the paper content here below should not be considered as complete, particularly regarding language, spelling, references etc., since in my opinion this manuscript and the work it describes needs almost a complete revision. The specific comments are mostly of two kind. Firstly, where I find reasoning of the methodology hard to understand or poorly justified. Secondly, where the authors are concluding much more from the data than they actually can, given the derived uncertainties (this is not a complete list, the remaining text free of such comments should also be critically revised, with this kept in mind).**

We thank E. Magnusson for his detailed comments. As stated above, we have revised the methodology. Regarding the uncertainties, the advantage of using an ensemble DEMs to constrain uncertainty is now better emphasized in the text. We do not agree that we concluded more than allowed from the data, for the reasons summarized in the general response.

The remaining text free of comments has also been revised. We tried to make shorter sentences and made sure that the methods are well explained. In our answers below we provide detailed indications how we streamlined the text, which we hope increased the readability of the paper.

**Page 1, line 12: This first line does not tell the reader anything since glaciers are losing mass at very variable rate (even glaciers short distance apart).**

The reviewer is right that the mass loss rates of individual glaciers are variable. However, we are referring to regional trends here. It is true that most Himalayan glaciers are losing mass at rates similar to glaciers elsewhere (Bolch et al. 2012). We modified the sentence slightly ("Himalayan glaciers are *on average* losing mass at rates similar to glaciers elsewhere").

**Page 1, lines 18-19: The uncertainties here have large overlap. Assuming that the uncertainties where e.g. 95% confidence level (let alone lower confidence), you cannot state with great confidence that you show that the volume loss rate is higher now (even though it is more likely that it is, rather than the opposite).**

We agree that the sentence needed clarification. In the revised manuscript we emphasize the ensemble of independent values available for the period 2006-2015, which allows constraining uncertainty. The new uncertainty estimates based on the standard error (Gardelle et al., 2013) yield lower uncertainties for the period 1974-2006. It is therefore possible to state now with great confidence that volume loss rates are higher. We have replaced the sentence with the following text:

*"The availability of multiple independent DEM differences allows identifying a robust signal and narrowing down the uncertainty about recent volume changes. The volume changes calculated over several multi-year periods between 2006 and 2015 consistently indicate that glacier thinning has accelerated with respect to the period 1974-2006. We calculate an ensemble-mean thinning rate of -0.45 ± 0.18 m a$^{-1}$ for 2006-2015, while for the period 1974-2006 we identify a thinning rate of 0.24 ± 0.08 m a$^{-1}$."*

Note that the uncertainty bounds provided above are still overlapping at the ends. In the revised manuscript we thus quantify the confidence level in our statement that thinning rates have accelerated (p. 16, lines 1-6). The estimated confidence level in accelerated thinning rates that is higher than 99%.

**Page 2, lines 8-10. Strange sentence, since you talk about examples of regional differences but only mention the upper limit values.**

The reviewer is right that the sentence was incomplete. We have changed the sentence as follows: "Prominent examples of current-day regional differences in glacier evolution across the Hindu Kush–Karakoram–Himalaya (HKH) are the reported positive glacier mass balances in the Pamir and Karakoram). *Glaciers in the rest of the HKH are thinning and receding (e.g. Bolch et al., 2012; Kääb et al., 2012; Gardelle et al., 2013)*" (p. 2, lines 13-16).

**Page 2, line 15. Is "scientific debate" a good phrase to describe this, isn't the common goal of everyone studying this just to obtain answer to the same scientific questions?**

We agree with the reviewer. We have replaced "scientific debate" by "research" and changed the sentence as follows: "However, also within the same climatic region the rate of glacier changes can be highly heterogeneous (Scherler et al., 2011b). A main focus of current *research is on the effect of supraglacial debris-cover on glacier response to climate.*" (p. 2, lines 20-22).

**Page 3, lines 7-17. Here the authors seem to give observations and models the same weight. When you have models on one hand and on the other hand conclusive observations, which don't fit the models, the reason for this is usually the incompleteness of the models, which in this case is probably the melting mechanism of the debris covered glacier.**

We kindly disagree with the reviewer. In our opinion the results of the two cited detailed modeling studies (Juen et al., 2014; Ragettli et al., 2015) are also relevant when discussing the effect of debris cover on melt. Note that these two modeling studies are based on a large number of field data that were used to inform, develop and validate the model. The two modeling studies include point scale glacier mass balance observations while geodetic studies usually do not. Moreover, the glacier thinning rates derived by geodetic studies are not equivalent to melt rates, because glacier uplift affects the derived thinning rates (see our answer to the reviewers' next comment below), while models can provide actual melt rates. Both modeling studies and geodetic studies have therefore limitations when assessing the role of supraglacial debris on glacier response. We have however slightly changed the sentence on model results:

"Several detailed modelling studies on the other hand have *provided evidence for a* melt reducing effect of debris at the glacier scale (e.g. Juen et al., 2014; Ragettli et al., 2015), and have concluded that supraglacial debris prolongs the response of the glacier to warming (*Banerjee and Shankar, 2013*; Rowan et al., 2015)" (p. 3, lines 2-6).

**Page 3, lines 15-17. I don't understand this sentence. What melt is caused by the glacier emergence velocity? Are you maybe referring to emergence of debris to the surface but not the classical glaciological term emergence velocity?**

The sentence was not clear and we apologize for this. What we meant by the "discrepancy between thinning and melt due to glacier emergence velocity" was that melt and thinning is not the same thing, since glacier emergence has to be accounted for when comparing thinning rates to melt rates (see e.g. Immerzeel et al., 2014). We have stated this more clearly in the revised manuscript (p. 3, lines 9-10).

**Page 5, lines 11-17. The author don't discuss at all the effects of seasonal changes on their geodetic results despite the fact that the DEMs (including the ones with most emphasizes, November 1974, October 2006 and February 2015) are from different time of the year. Can the seasonal effect be neglected? If so, based on what?**

According to detailed simulations by Ragettli et al. (2015) for the Upper Langtang catchment and the hydrological year 2012/2013, icemelt during post-monsoon and winter only represents about 2% of annual icemelt from debris-free glacier area and about 3% of icemelt from debris-covered glacier area.

The model that had been used for these simulations was informed by a large number of field data to guarantee internal consistency of simulated processes (data from glacier ablation stakes, temperature sensor network, automatic weather stations, glacier surface elevation change derived from UAV observations, glacier runoff data, debris thickness observations) and was thoroughly calibrated and validated. Moreover, also precipitation (and thus snow accumulation in the accumulation areas) is highest during the monsoon season. Post-monsoon and winter precipitation represents less than 20% of annual precipitation (Immerzeel et al., 2014b). Elevation changes during the winter half-year are thus minor in comparison to the changes during pre-monsoon and monsoon (March to September). To convert elevation changes into units of *meters per year* we therefore divide by the number of ablation seasons (p. 8, line 22). All our DEMs are either from late winter/early pre-monsoon (February – April) or from post-monsoon (October-November). Effects of seasonal changes on the geodetic results can therefore be neglected, especially since we mainly discuss time intervals between DEMs of 4 years or longer. We state this now clearly in the revised manuscript (p. 8, lines 23-26).

**Page 6, lines 31-32. What about glacier motion, does your velocity data give any upper limit on what the motion of the GCPs could be within the time frame (if so state it)?**

According to our velocity data, glacier motion during a period of 9-18 days leads to a horizontal shift of 10-20 cm. This is less than the grid size of the Pléiades image (0.5 m) and is therefore negligible.

**Page 8, line 1. Systematic errors in the glacier change map?**

We have changed the sentence as follows: "Systematic errors *in the elevation change maps* due to tectonic uplift which could be relevant after the April 2015 Nepal earthquake are also corrected with the co-registration."

**Page 8, line 4. Did Ragettli et al., (2015) do independent estimate on this or did they get the value from Sugiyama et al., 2013. If the latter Sugiyama et al., 2013 should be referenced for this. This ELA estimate, which presumably is just some average value for this catchment, is used in this paper to estimate accumulation area ratio (AAR) for each glacier. It is then repeatedly referred to in the paper like some actual observation of the AAR for the glaciers. It is not and given the unrealistically high variability of AAR in table 1 (15-86%) it is probably not even a good estimate for individual glaciers.**

ELA estimates: these are two independent observations. The ELA estimate of Sugiyama et al. (2013) is based on thinning profiles of Yala Glacier determined from surface elevation measurements. The ELA estimate of Ragettli et al. (2015) is based on observations from glacier ablation stakes. We agree there is uncertainty in our ELA estimate but it is the best assessment possible for the Langtang catchment. In the revised manuscript we have assessed the effect of ±100 m ELA uncertainty (Section 4.2.1 and Table 6).

AAR estimates: we agree that the AARs in our paper should not be regarded as derived from observations but as estimates. This is clear now in the revised manuscript (e.g. in Section 5.2 where we now explicitly state that our AARs are estimates). However, the variability of AARs is not unrealistic, given the large heterogeneity of glaciers in our study catchment. Similar ranges of values can be found in literature (e.g. Khan et al., 2015, find AARs ranging from 7% to 80% in the Upper Indus Catchment based on end-of-summer snow line elevation observations). Extreme values, such as the AAR estimate of 86% for Kimoshung Glacier, are discussed in the paper (Section 5.2) and can be explained by topographic characteristics. A new figure in the Supplement (Figure S5) compares the hypsometry of Yala and Kimoshung Glacier, which depicts why the two glaciers have such different AARs.

**Page 9, line 1. I am not really following you here, when you mention the term automated flow accumulation process. Are you delineating ice divides between neighbouring ice catchments? Is the big difference for Langshisha glacier between Pellicciotti et al.,(2015) and this study caused by some part of Langshisha glacier as defined in the former study, being considered as separated ice catchment in this study? If so state this clearly. I would also recommend that you revise Figure 1 to better reveal the coverage of each glacier with improved background image behind it. By doing so you can (hopefully) convince the reader that your delineation of the glaciers is the more appropriate one.**

Yes, some parts of Langshisha glacier as defined in the former study by Pellicciotti et al. (2015) belong to a different catchment. This is very clear if a high resolution DEM is used to delineate the upper boundaries of glacier but is not evident from optical images, since the ice divides are often entirely snow covered. We have clarified the sentence in the manuscript ("We also re-delineated the catchment boundaries using the SRTM 30 m DEM and an automated flow accumulation process to accurately *delineate the ice divides between neighboring catchments*", p. 12, lines 27-29).

We now use the Cartosat-1 2006 ortho-image as a background image in Figure 1. The shading on north-aspect slopes slightly facilitates the visual identification of ice divides.

**Page 9, lines 2-5. This is a huge difference and is bound to have great effect on the result. You compere these two studies later on for this glacier, without even mentioning this important difference.**

We agree the differences in area are large and that this should be mentioned in the comparison. We have added a sentence on p. 27, lines 21-23. See also the new Figure S2 which compares the glacier outlines for Langshisha Glacier used by the present study and by Pellicciotti et al. (2015). p.

**Page 9, line 12. Standard deviation of deltah/deltat at given point calculated for the up 28 difference maps or is this calculated over a given window?**

The standard deviations of Δh/Δt (σ Δh/Δt) values were calculated for each difference map and each 50 m elevation band of each debris-covered glacier tongue. However, in the revised manuscript σ Δh/Δt is not be used anymore but we directly use the information of the cliff and lake inventories to identify cliff/lake areas (Section 3.3).

**Page 9, line 16. Well here is the answer to the question above. Personally I don't find this a good way writing, when something is only partly explained in a sentence and the same sentence and the following sentence does not indicate that further explanations will be given, but then later on the missing puzzle suddenly pups up. When I read such text, I am always asking myself "did I miss something?"**

We agree that the two sentences (one starting on line 13 and one on line 16) should have been presented in reverse order. Both sentences have been removed from the manuscript due to the change of methods (see comment above).

**Page 10, lines 12-15. Here a justification why this should be errors but not actual elevation changes are completely missing. The span of elevation change rate over an entire glacier can easily be greater than the DEM errors but this depends on the time span, DEM quality, glacier type, etc.**

We agree that a justification was missing. We have corrected this in the revised manuscript (paragraph on 'outlier removal' in Section 3.2.3). 3σ levels are selected for outlier definitions outside the

accumulation areas following e.g. Gardelle et al. (2013). 3σ error levels are less strict that the 2σ levels that were used in the original manuscript and therefore the risk of misclassifying actual elevation changes as errors decreases. On the other hand, stricter 1σ error levels are applied in the accumulation areas since here outliers are more likely to occur and since in the accumulation areas only narrow ranges of values are plausible over periods of several years.

**Page 11, line 4. Outlier correction uncertainty? Do you maybe rather mean sensitivity to outlier removal?**

In the revised manuscript the term is renamed 'sensitivity to outlier correction' (Section 4.2.1). We think the two terms are mostly equivalent, since sensitivity to outlier removal leads to uncertainty in the geodetic estimates, given that perfectly objective and unambiguous threshold criteria for outlier detection do not exist.

**Page 11, lines 5-17. This is very confusing text. I don't really understand what you are doing including why the thinning rate 2006-2015 is appropriate proxy for the outlier removal (of all data sets or just the 2006-2015 difference map?).**

Here we assess if the mean Δh/Δt values are sensitive to outlier definitions. Note that 'Outlier correction uncertainties' (i.e. uncertainties associated with the correction of outliers) are not used anymore for the detection of Δh/Δt map outliers to simplify our procedures. The paragraph to which the reviewer is pointing here has therefore been removed from the manuscript.

The thinning rate 2006-2015 was the value used as threshold to identify outliers, in order to keep the level of noise below the level of signal (where the thinning rate 2006-2015 is the signal and the outlier correction uncertainty is the noise). However, we agree that we could as well have chosen a different thinning rate from the ensemble as threshold. In this respect the 2006-2015 thinning rate was not a good choice, because our decision lacked objectivity.

**Page 11, line 18. DEM adjustment uncertainty? Is the term uncertainty appropriate here? I do not see that the parameter explained in this section is really used in your uncertainty assessment.**

We agree that the term 'DEM adjustment uncertainty' was not a good choice since in the literature it is known as 'triangulation residual' (e.g. Paul et al., 2015). We agree that the determined triangulation residuals were not abundantly discussed in the original manuscript (we only referred to Figure S2 once).

'Triangulation residuals' are not used anymore as outlier criteria in the revised manuscript. We considered discussing triangulation residuals in the new Section 4.2.1 and provide the values in the new Table 6. However, we noticed that none of the triangulation residuals exceeds the elevation change uncertainties as provided by Table 5. To shorten the manuscript we have therefore removed all content on triangulation residuals.

**Page 11, line 29. I have problem obtaining the same results as the authors from this equation. If n=8 making Ndeltat=28 and k=3, I get C_2= (8 over 3)/(2*28)=(8!/(3!*5!))/(2*28)=56/56=1, not 6 as authors say one should get.**

The equation as shown in the paper was wrong but our calculations were correct. The denominator should return the number of permutations for a given k-element subset that can be selected from a number of n objects. The correct expression is 'n!/(n-k)!)' instead of 'n over k'.

The equation has been removed from the main manuscript and in general all content on triangulation residuals (see comment above)..

**Page 12, line 7-14. It took me quite a bit of time to actually understand what you are doing. I think I do now. Again I can't see what is logical about using the thinning rate from October 2006 to October 2015 as a threshold value. Can you explain that?**

The thinning rate 2006-2015 was used as a threshold to guarantee that the signal to noise ratio is higher than 1:1, which indicates more signal than noise (same explanation as above regarding the comment on p. 11, lines 5-17). However, the outlier criterion discussed here is not used anymore in the revised manuscript and the selection of Δh/Δt maps for the ensemble is based on simple and straightforward criteria, following the reviewers comments.

**Page 12. Do I understand you right that the last outlier detection you do is the catchment scale outlier detection? Wouldn't be more appropriate to do that before you the do glacier scale outlier detection?**

The order of steps is not significant, since the results of the glacier scale outlier detection do not depend on the catchment scale outlier correction. Note that the outlier criteria discussed here is not used anymore in the revised manuscript.

**Page 12, lines 23-24. Here we are left with the question "how?" until half a page later. Again, this is not a good way of writing, it makes the paper hard to read.**

We apologize for the writing style here. The outlier criteria discussed is not used anymore in the revised manuscript and therefore the sentences "half a page later" has been removed.

**Section 3.4.3. Here you come up with three outlier criterion. Why this complexity? It is not really justified in the paper.**

The three criteria look at mean off-glacier elevation differences (MED) and the role of slope and snow cover for MED. These criteria are not used anymore for outlier detection in the revised manuscript. We have moved the revised Figure 3 of the original manuscript to the Supplement (Figure S1).

**Sections 3.4.2-3. It seems to me that the glacier catchment scale outlier removals are not likely to function appropriately when the time interval between DEMs is so variable and you do the outlier detection on deltah/deltat. deltat is ranging from < 1 year up to 32 year. This means e.g. for the last criteria that the DEM error for the 1974 DEM causing the 1974-2006 deltah/deltat to be considered as an outlier would need to be 32 times larger than the error in 2009 DEM causing the 2009-2010 deltah/deltat to be considered an outlier. DEMs over short interval off course need to be very accurate to have informative value for volume change estimates, hence this is logical from that perspective. If my understanding of the outlier removal procedure is correct it does however result in very weak outlier criterion for the 1974-2006 interval. If the authors rely entirely on this automatic outlier removal, it may result in erroneous result for this period, which to me, seems to be the case when looking at Figure 6 a. This is very unfortunate given that the main focus of your results and discussion is on the difference between the periods 1974-2006 and 2006-2015.**

At the catchment scale we used the distributions of the mean off-glacier elevation differences (MED) to identify outliers. We agree that the DEM errors are more likely to be classified as outliers if the intervals between DEMs are short. This is certainly one of the reasons why the DEM differencing maps involving the Hexagon 1974 DEM were not identified as outliers. However, we think it was

justified not to compare absolute values (due to DEM errors, units in m) for outlier detection, since throughout the manuscript we are using units of m/a to discuss elevation changes.

To prevent erroneous results for the period 1974-2006 we now apply stricter outlier definitions for the accumulation areas, since only narrow ranges of Δh/Δt values are realistic in the accumulation areas. Note that also before the revisions of our methods; it would not have been justified to reject the entire 1974-2006 Δh/Δt map, since the quality at debris-covered areas is very good. Figure 4a confirms that the off-glacier elevation differences at lower elevations (where the image contrast is high and the terrain is less steep) are very small.

**Page 14, line 5. I find the problem with your bias or trend correction approach manifest in this equation (I guess you are not the only one doing this). If this study had been only on one of these glaciers the data used for trend or bias correction would (presumably) only have been from the neighbouring area of this glacier resulting in MED~=0. But since you do the trend correction for the catchment as a whole (which I think is fine if you are studying the catchment but not individual glacier), MED~=0 is often not true for individual glacier, hence you will get different value for a given glacier than if you had focused the study only on that glacier. You are trying to compensate for this by adding this effect here into the uncertainty, but you are still left with the fact that the probabilistic mean of the actual average elevation change is likely not well represented by the centre of the given error bars. This becomes particularly awkward since your discussion of the results almost neglect the derived uncertainty limits and focuses on the centre of the error bars.**

We agree with the reviewer that the center of the error bars does not well represent the probabilistic mean of the actual elevation change if MED~=0, and we agree that our discussions should have better reflected this. However, MED can only be different from zero at the very high elevations, where the presence of snow does not allow using off-glacier terrain for bias correction. In the revised paper we now use a new approach to deal with Δh/Δt errors in the accumulation areas of glaciers and consider only a narrow range of plausible values close to zero (Section 3.2.3). As a consequence of this new approach the center of the error bars now agrees with the probabilistic mean of the actual elevation change, and it is also not necessary anymore to consider MED for the uncertainty calculations (hence we use only the standard error following Gardelle et al. 2013).

**Page 14, lines 8-10. Are you saying that you use n=1? If so state it clearly, you could add to the sentence (i.e. n=1). Your usage of i.e. is not appropriate here (if I understand the sentence correctly). The fact that you use n=1 implies only that all pixels within the elevation band are fully dependent on one another (which truly is a conservative estimate). It does not however implies that there is no dependence between elevation bands. Since no attempts has really made to quantify the effect of the spatial correlation of your data (see e.g. Rolstad et al., 2009 or Magnússon et al., 2016, for further info) we don't really know if your assumption of no error compensation across elevation band is likely to lead to a conservative estimate of the uncertainty.**

We now use the standard error for uncertainty estimations, where n is equal to the number of *independent* measurements per altitude band, which means that the spatial correlation of the data is now taken into account. However, we would like to clarify that in the original manuscript n was not equal to 1, but n was the number of pixels per elevation band. We apologize if our usage of i.e. was not clear ("*i.e. assuming no error compensation across elevation bands*") and we have rewritten the sentence (p. 10, lines 26-28). It really meant that we weight the uncertainties identified per elevation band according to elevation distributions to calculate weighted averages per glacier. This implies that

we are assuming no error compensation across elevation bands (and thus 100% dependency between elevation bands). This is indeed a conservative estimate.

**Section 3.6. It seems to me that your surface velocity could do with some more masking of errors and outliers e.g. with correlation threshold. The masking that you are carrying leaves almost the entire velocity field intact as revealed by Figure 11 even though it is clear that much of it is just errors. The level of errors seen outside the glaciers is such that it is not clear if the signals on the glaciers are real or just errors as well. The figure itself is very hard view.**

We agree that Figure 11 (now Figure 13) needed to be improved, so that signal from glaciers can be better distinguished from errors outside the glacier area. Note that on the relatively flat debris covered areas errors are much less likely to occur. Errors occur where the terrain is steep or where image contrasts are low.

In the revised Figure 13 we masked out areas with slopes that are not representative for glacier area. We used a threshold of 45°, which corresponds to the 95th percentile of the slope of all glacier grid cells. Off-glacier velocity data are shown in transparent color so that signal from glaciers can be better distinguished.

The velocity profiles of debris-covered tongues (and error bars) are now shown in the revised Figure 12. The error bars represent the standard deviation in pixel values per elevation band and do not suggest that additional outlier correction is necessary

**Page 15, line 9. Outlier and uncertainty assessment? Confusing. Wasn't this already done?**

Here we presented results and not methods. As such, this section could have been part of the result section, but we decided to discuss uncertainties and outliers in a separate section given their importance. This section has been removed to shorten the paper, since most of its content became redundant after the simplifications in the data selection procedure.

**Section 4. This is all rather confusing. You calculate a lot of quality proxies used for outlier detection, mostly to convince yourself that the data that you derive your results from is of good quality. This is all good if one also reviews critically the outcome, which seem to be lacking in this study. A lot of these proxies are referred to as uncertainties apparently without being used to estimate uncertainty of the presented geodetic results. It is also not clear if all the DEM available during the period 2006-2015, apart from the initial and the final DEM, were really used to narrow down the uncertainty of volume change during this period. If not it seems to me that this paper would be much clearer if the focus of this paper were only on three DEMs, the ones from 1974, October 2006 and February 2015.**

We are not convincing ourselves of a good data quality. This section simply presented an honest assessment of uncertainties and outliers. The quality proxies that were chosen have all been already applied in previous studies (although mostly for quality assessments and not for outlier removal). We agree however that outlier removal algorithm obviously failed in the accumulation areas. In this respect we should have reviewed the outcome more critically. In the revised manuscript we correct this by improving the outlier removal at the grid scale (Section 3.2.3). Those instances of failure are no longer there.

It is true that the proxies were not used to estimate uncertainty of the presented geodetic results but only for quality assessments and for outlier removal. This made the paper lengthy and difficult to read. In the revised manuscript the quality proxies outlier correction uncertainty (or 'sensitivity to outlier correction') and mean elevation differences (MED) are now only be presented for quality assessment.

Contents regarding DEM adjustment uncertainty (or 'triangulation residual') have been removed from the manuscript to shorten the paper since we noticed that triangulation residuals are within the uncertainty bounds as provided by the revised Table 5. Our main criteria to select Δh/Δt maps for the ensemble is now the time interval between DEMs, since we can show that the uncertainties decrease with period length (see new Figure 5). In this respect the uncertainty estimates are now used directly for outlier detection. Section 4 has been entirely removed to shorten the paper.

We are sorry if it is not clear that we used all DEMs available for 2006-2015 to narrow down the uncertainty. We thought it was (e.g. p. 18, lines 1-3; p. 18, lines 10-12; p. 18, lines 25-26, p. 24, lines 30-33, p. 30, lines 9-13). We did use all the DEMs and are convinced that there are clear advantages in doing so, since several independent measurements (differential DEMs for the period 2006-2015) lead to higher confidence in detected signals, even if the uncertainty of each measurement is high. For this reason, we do not see the point of using only three DEMs (the alternative suggestion of the reviewer). However, we have made an effort to more clearly explain the advantages of the ensemble approach throughout the revised paper.

**Page 18, line 18. Well if you think this is due to remaining systematic error, did you consider that your outlier removal is maybe not functioning so well?**

The outlier removal functioned excellently for ablation areas, considering for instance Figure 8, which clearly shows which thinning patterns are consistent across different dataset. If we considered all 28 differential DEMs for Figure 8 it would not have been possible to clearly identify patterns which are consistent across datasets, because the ability to identify these patterns requires a level of accuracy which is not granted per se. However, in the accumulation areas we agree that the outlier removal did not help to clarify thinning/thickening changes. This is why we decided to choose a different approach for grid scale outlier correction here (Section 3.2.3). The outlier removal the catchment scale has also been completely revised (Section 3.2.5).

**Page 18, line 23-24. This is very true. Unfortunately you seem to forget it repeatedly in your discussion. Given that your uncertainties will be the same after revision of this work, much of the discussion on the results can be omitted because it is meaningless due to the large uncertainties.**

After revision of the methods the uncertainty estimates are different, especially for the periods 1974-2006 and 1974-2009. Regarding the particular example of Kimoshung Glacier, to which the reviewer is pointing here, the uncertainty could be constrained by only accepting realistic Δh/Δt values in the accumulation areas (Figure 9g). We are puzzled by the dismissive tone of the reviewer.

**Page 19, line 6. This is a good example of what I am talking about regarding the author neglecting the uncertainty in their discussion of the results. You cannot state here that the thinning rate increased by more than 100 %. If we know that John owns between 0 and 4 cars and Mike owns between 2 and 6 cars can you state that Mike owns at least twice as many cars as John? No and the probability of such statement being true is only 14/25=0.56 (given even probability distributions for the car ownership in both cases).**

The reviewer's schoolmasterly example is not appropriate here. We are not comparing only two estimates, but thinning rates of two periods (1974-2006, and 2006-2015), where for the second period we use several independent datasets. This changes the situation as a whole. To take up the reviewers' example, John belongs to an automobile club and we are estimating the average number of cars owned by the members of the club. If each member owns between 2 and 6 cars, then the standard deviation of the sample mean is

$$\mathrm{SD}_{\bar{x}} \ = \ \frac{\sigma}{\sqrt{n}}$$ (https://en.wikipedia.org/wiki/Standard_error).

whereas σ is the standard deviation of each single estimate and *n* the sample size (number of club members). Therefore the standard deviation converges towards zero for a large ensemble. Already with only four ensemble members the standard deviation of the sample mean decreases by 50%. In other words, the error might be too large when comparing only two periods, but when comparing two groups of values for the two periods the differences between the values become significant. We show this clearly in the revised manuscript (see figures 9 and 10).

Throughout the revised manuscript we now emphasize better the value of the ensemble approach by reporting ensemble mean and ensemble uncertainty values for the period 2006-2015.

We would also like to strongly rebut the reviewer's statement that we are neglecting the uncertainties in the discussion of our results. In the particular sentence to which the reviewer is pointing here the uncertainties were provided in brackets. We think this was an honest way of presenting the results. Uncertainties were discussed abundantly throughout the manuscript (with Section 4 we dedicated a whole section to the discussion of outliers and uncertainties). We have however revised the manuscript to discuss uncertainties now together with the results (e.g. by stating the confidence level in detected thinning accelerations, Section 4.2). Accordingly, we have removed Section 4 of the original manuscript).

**Section 5.2.1. This comparison between debris covered and debris free glacier looking at "Explanatory variables" is rather primitive. For one thing it is rather inappropriate to refer to some of them as variables. I would rather refer to outcome of processes, which in some cases probably show correlation since they are dependent on the same physical variables. It is also strange that only Yala is included as candidate for the debris free glacier. The behaviour of Yala is then compared with 5 other debris covered glaciers. Even though the difference between Yala and each of the other 5 glacier is sometimes visually clear it is misleading to calculate the r-value for all the 5 debris covered glacier at ones and compare with a value calculated for a single glacier. When using data from several glaciers, various variables which effect the glaciers in different manner is bound to reduce the studied correlation compared to having data from just a single glacier.**

This section has been thoroughly revised, also based on comments by reviewer 2. We now use a new dataset of cliff and lake areas, which substitutes the cliff/lake proxies (detailed explanations above under 'New Methods' and 'Explanatory variables'). The section has been renamed to "Section 4.5: Surface velocities and supraglacial cliff/lake areas". The old Figure 10 has been removed and replaced with the new Figure 12. With the scatterplots and r values (old Figure 10) we attempted to explain the variance in thinning rate changes of all debris-covered glaciers at the same time. However, single variables cannot possibly explain all the spatial variation in thinning rates, and therefore the r values were generally low (max. 0.55 for debris-covered glaciers). We agree that the sample size affects the correlation coefficients, and therefore the r values calculated separately for debris-covered terrain and Yala Glacier were not comparable.

In the revised Section 4.5 we discuss the spatial variability in thinning rates at each debris-covered glacier separately (see revised Figure 12). The differences between debris-free and debris-covered glaciers are not presented anymore in this section, but are addressed briefly in Section 4.3 ("Altitudinal distributions of elevation changes"). It is sufficient to state that at debris-free glaciers

thinning rate changes are elevation dependent while this is not the case at debris-covered glaciers (p. 18, lines 23-29).

Note that we think it is justified to consider Yala Glacier as the main reference for debris-free glacier characteristics. Kimoshung Glacier has an unusually high accumulation area ratio , is very dynamic and is affected by higher DEM uncertainties due to the steepness of the tongue. Yala Glacier has an AAR of approximately 40% (Table 1) which is identical to the mean AAR indicated by Kääb et al. (2012) for the entire HKH region and is therefore suitable as a reference. In the revised paper this is now mentioned (p. 25, lines 9-11).

It is not clear to us which of the variables discussed in the original Section 5.2.1 we cannot refer as "variables". In Section 5.2.1 we considered elevation, slope, surface velocity and the variability of local thinning rates (as cliff/lake proxies). The reviewer is right that e.g. the variability of thinning rates (σ dt/dh) is an outcome of processes. All the others are variables. Still, in our opinion it was correct to refer to them as explanatory variables (σ dt/dh explains the role of heterogeneous surface properties for local changes in thinning rates).

**Page 22, line 1. You mean April 2015.**

Yes, thank you for noticing. We have corrected this.

**Section 5.3. It is probably of interest for some to know the volume of these enormous avalanches. It however seems clear that avalanche falling on debris covered glacier (particularly the low insulated part of the glacier) is only going have minor and short last effect on the mass balance since it is going to melt much faster than the debris covered ice beneath it.**

It is not clear per se that the avalanche cones disappear quickly. It is documented and described in our manuscript that a new debris layer appears on top of the avalanche material (as snow/ice melts out from the debris; p. 15, lines 4-7). Avalanche accumulation is one of the most important processes for debris-covered glacier formation (Scherler et al., 2011a). Likely the tongue of Lirung Glacier would not exist without accumulation through avalanches (Ragettli et al. 2015). We therefore do not agree that the long term effect on mass balance of the enormous post-earthquake avalanches is clear. In the revised manuscript we provide more background information about debris-covered glacier formation (Section 5.1.1).

**Page 24, line 25. What is numerical evidence?**

We have removed "numerical" and just left the "evidence".

**Page 25, line 14. Is there no uncertainty in the area change?**

We have added uncertainty estimates to the revised Table 7 and to area changes reported in the text (assuming a 0.5 pixel buffer around the tongues; p.12, lines 22-23). We now dedicate a separate results section on glacier area changes (Section 4.4), following a suggestion by reviewer 2. **Page 25, lines 16-17. I don't understand what you specifically mean by correlation between areal changes and surface elevation height.**

What we meant here is that one can expect glacier mass balances near steady state if the glacier area remains nearly constant, as it is the case for Kimoshung Glacier. We have removed the corresponding sentence to shorten the paper.

**Page 25, lines 23-26. Again not promising for your outlier removal, even though it is better to admit it does not work to well. It would however be even better to justify why you think this is an error, e.g. by pointing out that local lowering of 60-100 m over 32 years (as indicated by figure 6a) on such small glacier at such high altitude is very unlikely to say the least.**

We agree with the reviewer that the values were unrealistic. We correct this with a new outlier removal approach for accumulation areas (see response in our general statement above).

**Page 25, line 28-29. Even though it is likely that hypsometry plays a crucial role here this statement is far too bold given that it is based on very limited and apparently erroneous data (according to Figure 6a).**

After replacing the erroneous data we now get results which are trustworthy. The data now unambiguously reveal the differences in elevation change rates between Kimoshung and Yala Glacier (revised Figure 8f and g). Those differences can only be explained by the very different altitudinal distributions of the two glaciers (see new Figure S5). The results now clearly suggest that the hypsometry plays a crucial role.

**Page 25, line 31. See my previous comment regarding the AAR.**

We think the AARs in our study are realistic estimates. Especially at Kimoshung Glacier an uncertainty in the ELA by ±100 m would not lead to very different AARs because the glacier is very steep near the ELA (AARs given an ELA uncertainty of ±100 m: 80%-88%, see new Table 6). The uncertainty about the ELA does not change the fact that the AAR of Kimoshung Glacier is high, and this is what really counts for the discussion here.

**Page 26, line 26. Here and at other places in this paper, some temperature data (if available) would support your discussion.**

It is not the purpose of this study to document warming trends in the Nepalese Himalaya. It is difficult to relate spatial changes in elevation over different glaciers to remote point observations of temperature trends and the meaning of this would be very limited. In fact, no study of geodetic mass balance has used this coarse information to explain some of the observed thinning patterns. We therefore refrain from doing this here. References on warming rates in this part of the Himalaya are provided on page 23, line 23.

**Page 27, lines 16-18. You have far too little data with far too great uncertainty to make such statement.**

We agree with the reviewer that the wording needed to be improved here. However, we stated that our observations do not support the findings of previous studies and that is true. We find this an important result especially because it is the first time that this is proven with detailed, multi-ensemble data for a specific catchment using high resolution DEMs (in contrast to large scale regional studies). We agree that we cannot extrapolate this finding to larger glacier samples and we reworded the sentence to make this clear ("Our observations do not support the *findings* of previous studies about *similar present-day lowering rates of debris-covered and debris-free glacier areas* at the same elevation (Kääb et al., 2012; Nuimura et al., 2012; Gardelle et al., 2013)", p. 25, line 31).

**Page 27, lines 20-23. You can say that Kimoshung glacier has higher hypsometry than Yala. The staggering difference between AAR values (which here are treated as some kind of truth but not as estimates based on the assumption of fixed ELA=5400 m a.s.l. for the whole catchment) is however misleading.**

We do not agree with the reviewer, as explained above. Previous studies have found similar ranges of AARs in the HKH (e.g. Khan et al. 2015) and the AAR differences between individual glaciers (e.g. Yala and Kimoshung Glacier) can be explained by topographic differences (new Figure S5). We have added the reference to Khan et al. (2015) to the paper (p. 24, line 6).

**Page 28, lines 16-17. I am confused, is this in accordance with previous statement in this section (page 27, lines 16-18).**

We apologize if the wording was not clear here. Our observations tell us that the thinning rates at debris-covered tongues are lower than at debris-free Yala Glacier AT THE SAME ELEVATION (this relates to the discussion on page 27, lines 16-18, in the original manuscript). However, there are examples for both types of glaciers where AVERAGE thinning has increased significantly or where thinning remained approximately constant (p. 26, lines 7-10). Differences in thinning rates at the same elevation do not allow concluding about differences in average thinning. When it comes to glacier mass balances or glacier-wide average thinning rates, the elevation distribution of glaciers plays a crucial role. We have almost entirely rewritten section 5.3 ("Differences between debris-free and debris-covered glaciers") to be more clear and to base the discussion also on observations from debris-free Kimoshung Glacier (the new Figure S5).

**Page 28, lines 27-28. There is completely insignificant difference between these values. There is no point in trying to explain the "difference" between them.**

We agree that the differences are not significant, but we think it is justified to point to a possible overestimation of thinning rates identified by Pellicciotti et al. (2015) due to an underestimation of the SRTM radar penetration depth.

**Page 29, lines 6-10. I am very puzzled here. You need to justify here why this data is now suddenly considered as usable data, when the one processing the data rejected it in recently published paper. Why has he/she as the third author of this paper changed his/her mind?**

The two studies (Pellicciotti et al. 2015 and the present study) use different outlier correction approaches, but both studies tried to avoid arbitrary or subjective criteria. Pellicciotti et al. 2015 used 2σ thresholds to define outliers at the grid scale, but those thresholds are calculated only once for all altitude ranges together. In the original manuscript we used a slightly less restrictive outlier definition (thresholds calculated separately for accumulation and ablation areas), with the consequence that at some places (e.g at Langshisha Glacier) erroneous data remained in the dataset, while at other places (e.g. at the tongues of debris-free glaciers) the application of a less strict criteria lowered the risk of classifying correct data as outliers. This choice was justified by the fact that the quality of the new DEMs (2006-2015) is generally much higher than the quality of the Hexagon 1974 and the SRTM DEMs used in Pellicciotti et al. (2015). The third author therefore did not change her mind. The choice of the method depended on data quality. We also notice that there is progress in scientific research, and that improvements of revision of methods are common in papers by the same authors when data of better quality are available or new approaches are deemed more suitable. We do not see anything wrong here.

However, the 1974 DEM is the same for both studies, and therefore the less restrictive outlier definition failed to identify erroneous data in the 1974 difference maps. We have corrected this with the revised outlier correction procedure (see 'New Methods' above).

**Page 29, line 13. How can you state this? Does including apparently erroneous data make the uncertainty estimate more realistic?**

Yes, we think that the uncertainty estimates should reflect the quality of the data. On the other hand it is also justifiable to exclude erroneous data from the start and use uncertainty estimates that reflect the quality of the data without considering those outliers. We think this is what the reviewer suggests and this is what we have done in the revised manuscript. In general, please see our response on the new outlier removal and uncertainty estimates.

**Page 29, lines 15-16. What other data did they use? They were hardly using GPS in 1982.**

The reviewer is right, they did not use GPS in 1982. To cite Sugiyama et al. (2013): "The surface elevation in 1982 was surveyed by ground photogrammetry (Yokoyama, 1984) and later digitized into a 10m resolution digital elevation model (DEM) (Fujita and Nuimura, 2011)." This is now stated correctly in the revised manuscript (p. 28, lines 8-9).

**Page 30, lines 21-23. Sorry, I don't think many will agree on this statement.**

We think that many will agree on this statement – especially after the revision of this paper. However, we have omitted this sentence and leave it to others to judge if our study is one of the most solid ones.

**Page 32, lines 5-9. This text does not fit into conclusion. If the authors think this text should be in the paper, it would be more appropriate to include it in the introduction.**

We agree that the text does not fit here. We have removed those lines and added the reference to Kargel et al. (2015) to the introduction (p. 4, line 17).

**Page 40, Table 1. See previous comments regarding the AAR.**

The reviewer's comments regarding AAR are addressed above.

**Page 42, Table 5. It is not clear how the uncertainty of the average elevation change over the entire Langtang glacier catchment is calculated. Given its value it seems close to being basically (-)/(A*2), which basically corresponds to assuming that errors between glacier are completely dependent. Such assumption gives really conservative estimate, even too conservative causing the results to be downgraded. I also recommend that you stick to the same order of glaciers in the table as given in Table 7 with the glacier id.**

We calculate uncertainties of the entire glacierized terrain identically as for any other given area: we consider the standard error per elevation band (eq. 2) and the altitudinal distribution to calculate weighted averages. We explain this more clearly in the revised manuscript (page 10, lines 26-28).

It is not clear to us what the reviewer means by (-)/(A*2). It is true that errors between glaciers are dependent, since always the same off-glacier data are used. Only the altitudinal distributions are different (and therefore the uncertainty estimates). We have stated this clearly in the revised manuscript (p. 11, line 1).

We have changed the order of glaciers in Table 5 as suggested.

**Page 43, Table 7. Why no uncertainties? Are they within the digit of the given value, or did you simple not think about it? You are discussing these area changes in the paper without giving the reader any confirmation that these changes are significant.**

We have added uncertainty estimates to the revised Table 7 (assuming a 0.5 pixel buffer around the tongues; p.12, lines 22-23). The uncertainties are small given that we used high resolution (1.5 m to 4 m) optical satellite imagery to delineate the glaciers. Detected area changes are significant.

**Page 43, Figure 1. See my previous comments regarding this figure.**

The reviewer's comments regarding this figure are addressed above.

**Page 44, Figure 2. The data on the debris covered glaciers is the most convincing part of this manuscript.**

We thank the reviewer for this assessment and we agree with his evaluation. Indeed, the uncertainties are low over debris because of good image contrast and shallow slopes. We agree that here the data are least ambiguous, and allow to unambiguously infer some very interesting results on thinning patterns.

**Page 45, Figure 3. It seems to me that using all the 6 proxies result in the same outlier removal as when you just use med2 and sigma2.**

The reviewer is right. As explained above, none of these criteria are used anymore for outlier removal in the revised manuscript. This figure has been moved to the Supplement (Figure S1) and now only shows the stable terrain uncertainties of the $\Delta h/\Delta t$ maps in the final selected ensemble.

**Page 46, Figure 4. 50% confidence level? What would the error bars be for a reasonably strict confidence level like 95%?**

In the revised manuscript we provide the error bars with a confidence level of 95% but only considering the $\Delta h/\Delta t$ maps of the final selected ensemble (Figure 3 in the revised manuscript).

**Page 47, Figure 5. Do you mean: a) A whiskers plot showing the distribution of uncertainties for the (up to?) 28 deltah/deltat maps. What do the red crosses indicate?**

The boxplots show the distribution of uncertainties for non-rejected $\Delta h/\Delta t$ maps. This is now specified in the caption text of the revised manuscript (now Figure 2). The red crosses indicated outliers according to the standard Matlab boxplot function. In the revised figure we removed the red crosses and whiskers extend to the most extreme data points. Please note that we have replaced the old Figure 5b by the previous Figure S1 (and removed the latter from the Supplement). We think that the stereo matching scores are more directly linked to calculated uncertainties (new Figure 2a) than the fraction of pixels remaining after removing low stereo matching scores and outliers. If the reviewer prefers, we can still add the old Figure 5b to the Supplement. **Page 47, Figure 6. See various previous comments on this figure. Should also be enlarged for better readability.**

The reviewer's comments on this figure (now Figure 4) are addressed above. We will decide later (in the proof stage) if this figure will be sufficiently large if printed as a two-column paper figure. Please note that we have added a new figure to the Supplement (Figure S3) showing the elevation change rates of all $\Delta h/\Delta t$ maps in the 2006-2015 ensemble.

**Page 48, Figure 7. The order of panels for the glaciers should be kept the same as the numbering of the glaciers in Table 7.**

Ok, we have reordered the panels as suggested (see revised Figure 9).

**Page 50, Figure 9. Something went wrong with the altitudinal distribution for Yala.**

The altitudinal distribution for Yala Glacier is shown correctly. The altitudinal distributions of all glaciers are shown for 50 m elevation bands. The reviewer might not have noticed that the x-axis ranges are different for each sub-figure. We now point to this in the revised caption text (see revised Figure 11).

**Page 51, Figure 10. See previous comment regarding this figure.**

This figure has been removed from the paper and replaced by a new figure (now Figure 12) that better shows the relationship between glacier thinning and glacier characteristics and supraglacial features (see previous comment on this).

**Page 52, Figure 11. Very hard to read. The arrows are e.g. very hard to detect. Results do not appear very reliable (see previous comment).**

The reviewer's comments on the reliability of the velocity data are addressed above. We have improved the quality of this figure (larger arrows, focus on debris-covered areas, see revised Figure 13).

**IN RESPONSE TO REVIEWER 2**

**General comments:**

**This paper presents glacier surface elevation change in Langtan Himal from 1974 to 2015 based on DEMs generated from satellite images. The authors analyzed temporal and spatial patterns of glacier thinning over the studied seven glaciers. Focuses of the discussion are spatial heterogeneity in the thinning rate, comparison of debris-covered and debris-free glaciers, changes in the thinning rate after 2006. The data are also used to quantify the impact of the earthquake in 2015.**

**Despite the increasing importance and interests on the Himalayan glaciers, long-term data on glacier changes are few in the region. Considering intensive research activities in the Langtang region in the past and recent periods, the presented data set is valuable. Nevertheless, uncertainty is rather large particularly in higher elevation areas.**

**This is very common in photogrammetric elevation analysis because snow covered surface loses surface features required for this method. Judging from the unrealistic thickening and thinning patterns in Figure 6a, it is questionable whether the DEM analysis is applicable in the accumulation areas. Moreover, estimated uncertainties are based on very complex outlier rejection criteria, which sometimes appear to be subjective and unconvincing. These problems result in limited reliability in the conclusions. Overall impression on the manuscript is that conclusions are too conclusive as compared to what are shown by the data.**

**I encourage the authors to thoroughly revise the manuscript (1) by using only reliable data, (2) with well focused objectives, (3) to draw only convincing conclusions. For example, omitting data from the accumulation reduces total uncertainties in Figure 7, which leads to more reliable discussion on recent increase in the thinning rate. Among others, elevation change over the debris-covered regions and impact of the earthquake are promising subjects.**

We thank the reviewer for his/her very useful comments and appreciating that the data set is valuable. We agree with his/her comment on the accumulation area values and have changed for this our outlier removal procedure (following also suggestions from Reviewer 1). As a result, erroneous data from the accumulation areas are now corrected. In this way, all conclusions presented can be solidly justified.

We also agree that the paper should have well focused objectives. We have simplified the procedure for selecting maps for the 2006-2015 ensemble (see explanations above under 'New Methods' and 'Data Selection' and the new Section 3.2.5), which allow to focus better on the three paper objectives as defined in the Introduction (p.4, lines 9-14).

We agree with the reviewer that the results on the debris covered sections and impact of the earthquake are relevant and interesting results of our work, and thank him/her for noticing this.

We kindly disagree that overall "the conclusions are too conclusive as compared to what are shown by the data"; since also with our original methods and data we were able to present unambiguous evidence for heterogeneous thinning patterns (e.g. spatially variable thinning trends at debris-covered tongues). We think this result alone deserves publication in TC, but we agree that the manuscript required revision. After following both reviewers' suggestions for improvement, we are now able to draw more convincing conclusions regarding all addressed main points.

To demonstrate that with the revised methods we are able to draw relevant conclusions that are supported by data we shortly summarize the main quantitative outputs of our study regarding the three main goals:

1.) Assess if overall thinning of glaciers in the region has accelerated in recent years.

Thinning rates have increased, from -0.24 ± 0.08 m a$^{-1}$ (1974-2006) to -0.45 ± 0.18 m a$^{-1}$ (2006-2015, ensemble mean). The uncertainty bounds are overlapping at the ends. However, the probability that thinning rates have not increased is less than 1% (estimated confidence levels are now reported in the revised manuscript on p. 16, lines 1-5).

2.) Determine if spatial thinning patterns have changed over time.

We can now conclude that spatial thinning patterns have changed over time, since thinning accelerations at the debris-covered tongues are highly non-uniform in space. Local changes in thinning rates (comparing the periods 1974-2006 and 2006-2015, Figure 8) range from -80% (at Ghanna Glacier, 4800 -4850 m a.s.l.) to +150% (at Shalbachum Glacier, 4650-4700 m a.s.l.). The uncertainty in identified thinning accelerations is only about ±10% (p. 29, line 15).

3.) Assess if there are major differences between the response of debris-covered and debris-free glaciers in the sample.

Here we partly agree that we were too conclusive in the original manuscript regarding this point. In the revised manuscript we state clearly that our observations need to be confirmed by studies using larger glacier samples (p. 27, lines 1-2). However, considering that the elevation distribution of Yala Glacier is common for the HKH (an AAR of 40% is common in the HKH, see Kääb et al. 2012), this glacier can be used as a reference. There are indeed *major differences* between debris-covered glaciers and Yala Glacier: Within the same altitudinal range, thinning rates of debris-covered glaciers do not exceed 35% - 75% of the thinning rates at Yala Glacier (p. 26, line 6-8). Considering the changes in mean thinning rates, we identified a strong thinning acceleration at Yala Glacier from -0.33 ± 0.06 m a$^{-1}$ (1974-2006) to -0.89 ± 0.23 m a$^{-1}$ (2006-2015, ensemble mean) (p. 16, lines 25-26). Debris-free Yala Glacier is currently downwasting at 60%-100% higher rates than the large debris-covered glaciers in the valley (p. 26, lines 29-31). Our work is one of the first to assess differences between debris covered and debris-free glaciers at this level of details and high spatial resolution.

**Major concerns:**

**1. Reliability of the DEM in the accumulation area Figure 6a shows unusually large thickening and thinning patterns in the accumulation areas. The regions of the suspicious elevation change agree with the frequently snow covered regions shown in Figure 1. Most likely, photogrammetric analysis is hampered by featureless snow surfaces. Because such data from the accumulation areas are used for the mean thinning rate over each glacier, conclusions on the recent thinning acceleration and comparison between debris-covered and debris-free glaciers are unreliable.**

We have carefully revised our outlier removal procedure to address the reviewers' concern. It is true that outliers in the accumulation areas remained in the 1974-2006 map and therefore the outlier removal procedure failed for these areas. Our new approach for outlier correction is described at the beginning of this document under 'General Revisions' (and in Section 3.2.3). The revised outlier detection algorithm now identifies unrealistic patterns in the accumulation areas and removes them. Missing data in the accumulation areas are replaced by plausible values (see Figure 4a which corresponds to old Figure 6a).

Although erroneous data over featureless snow surfaces in the Hexagon 1974 DEM are evident, we are convinced that with the revised outlier correction and gap filling procedure now allows for convincing conclusions regarding recent thinning accelerations. As the reviewer states above it is common in photogrammetric elevation analysis that uncertainties are high over featureless snow surfaces. Many previous studies addressed the same problem. Errors in the accumulation areas do not require rejecting the whole dataset, since it known that over long time periods only narrow ranges of $\Delta h/\Delta t$ values close to zero are realistic in the accumulation areas (e.g. Schwitter and Raymond, 1993; Huss et al., 2010). Previous geodetic studies have thus assumed no elevation changes in the accumulation areas (Pieczonka et al.,2013) or have used glaciological expert knowledge to define acceptable $\Delta h/\Delta t$ ranges in the accumulation areas (Pieczonka and Bolch, 2015). In our study we benefit of a large dataset of several independent $\Delta h/\Delta t$ maps. We now use the available information to narrow down the uncertainty and replace missing data in the accumulation area with data from the same glaciers (Section 3.2.3).

We now also use a more established method for uncertainty quantification (Gardelle et al. 2013). The new approach results in substantially lower uncertainty estimates for the 1974-2006 scene (see revised Figure 9). Our previous approach clearly overestimated the uncertainties (see our response to the second main comment by reviewer 1). We are therefore convinced that the revision of the outlier correction and uncertainty estimation procedures allows now for substantially more convincing conclusions regarding recent thinning accelerations and differences between debris-covered and debris-free glaciers.

**2. Data and Method section. The authors spend more than 1/3 of the manuscript for Data and Method section. This section is suffered from too much detailed explanations on how to reject outliers and estimate uncertainty. All details are given, but hard to understand the reasoning of each process. First, I suggest the author to move these details to the supplement, and describe in the main text only essence of the techniques in an understandable way. Second, the structure of the section should be reconsidered. It can be something like, 3.1. Satellite data, 3.2. DEM (generation, differencing, processing, uncertainty), 3.3. delineation, 3.4. velocity.**

We agree with the reviewer that the outlier removal procedure was complex and the presentation of the method took substantial manuscript space. The selection of maps for the ensemble was based on four different data quality proxies (% data available after outlier correction at the grid scale, sensitivity to outlier correction, triangulation residuals, and mean off-glacier elevation differences). Although all these proxies have already been applied in previous studies for quality assessments, perfectly objective criteria were not available to decide whether a map should be included in the ensemble or not. We assume this is the reason why the reviewer states that it is "hard to understand the reasoning of each process".

We have therefore decided to considerably simplify the procedure for the selection of maps. We now simply select all $\Delta h/\Delta t$ maps from the period 2006-2015 that cover periods of four years or longer (Section 3.2.5). Short periods are discarded from the beginning, since uncertainties increase with shorter time intervals (due to lower signal to noise ratios, see also new Figure 5). We have restructured the Methods section (Section 3) as suggested by the reviewer.

Since the quality proxies are now not used anymore as criteria for outlier detection we have proceeded as follows: Sensitivity to outlier correction is now assessed in a short separate sensitivity section (Section 4.2.1, new Table 6). The figure about mean off-glacier elevation differences has been transferred to the Supplement (Figures S1). The fraction of glacier pixels remaining after removing outliers and low stereo matching scores (old Figure 5b) and triangulation residuals (old Figure S2) is

not presented anymore in the paper to shorten the manuscript. If the reviewer finds this important we can add the old Figure 5b to the Supplement and/or discuss triangulation residuals in Section 4.2.1, but would prefer not for reasons of shortness. None of the triangulation residuals exceeds the elevation change uncertainties as provided in Table 5. This means that the potential co-registration errors are within our uncertainty estimates.

**3. Influence of the earthquake It is interesting and important to evaluate the impact of the earthquake on the glacier surface elevation. However, the elevation change due to the earthquake in 2015 is essentially different from those occurred from 1974 to 2014. Accordingly, elevation change from 1974 to 2015 (e.g. Table 5 and Figure 6b) is not suitable to discuss recent glacier changes in general. Therefore, I suggest the author to separate the elevation change after the earthquake from the rest of the study period.**

We agree that elevation changes after the earthquake are locally very different from those during the period before the earthquake. For this reason, the May 2015 SPOT7 DEM has not been used to assess long-term glacier changes. However, we also concluded from our analysis that "Over periods of several years, the effect of the post-earthquake avalanches on the altitudinal thinning profiles such as presented in Figure 8 is only minor." (p. 23, lines 6-8, original manuscript). We now assess more carefully in Section 4.1 ("Impacts of the April 2015 earthquake") which of the post-earthquake DEMs can be considered to discuss recent glacier changes. We do this by comparing the avalanche impact to long term glacier changes and to uncertainties associated to elevation changes.

Almost 90% of the avalanche debris remaining in October 2015 accumulated on Lirung and Langtang glacier tongues. However, neither at Lirung tongue nor at Langtang tongue post-earthquake elevation changes represent outliers with respect to other 2006-2015 multi-annual periods (p. 17, lines 7-10).

The short-term effects of the post-earthquake avalanches (April 2014 – Oct 2015 elevation changes) are now shown in the revised Figure 8 and can be compared to the elevation changes 2006 - Oct 2015 and 2009 - Oct 2015. It becomes clear that in contrary to Apr 2014 – Oct 2015, the elevation change profiles 2006 - Oct 2015 and 2009 - Oct 2015 show very similar characteristics in comparison to other recent periods.

For the reasons stated above we therefore still use the October 2015 SPOT7 scene to discuss glacier changes in general (e.g. Figures 9-11). However, we replaced the differential DEM on Figure 6b (now Figure 4b) by a map showing only pre-earthquake elevation changes (but all other $\Delta h/\Delta t$ maps of the 2006-2015 ensemble are now presented in the new Figure S3).

**4. Text. I understand that the author tried to be careful and accurate in the text. However, the manuscript is lengthy, redundant and diffuse at many places. This hinders reader's understanding of the methodology, important results and conclusions. Please consider to shorten and simplifies sentences throughout the manuscript.**

We agree that the text needed to be improved by shortening and simplifying sentences throughout the manuscript. We have revised the text as requested by the reviewer.

**Specific comments:**

**page 1, line 15: we present volume and mass changes of . . . (omit "glacier")**

Ok.

**page 1, line 22: "mass balance trends" sounds to me "surface mass balance trends". What about "mass loss trends" or "thinning trends"?**

We indeed refer to "surface mass balance trends" (p.1, line 26).

**page 1, line 22: "highly non-linear" to what? elevation? time?**

We will add "spatially non-linear thinning profiles" (p.1, line 27).

**page 3, line 4: What do you mean by "downslope condition"?**

We have removed the corresponding sentence to shorten the manuscript. What we meant is that the surface mass balance at the debris-covered tongues may be influenced more quickly by changes in high-altitude precipitation due to avalanche nourishing. **page 3, line 8: . . . present-day "surface" lowering rates. . .**

We have changed the sentence as suggested (p. 2, line 31) .

**page 3, line 16: What is "melt due to glacier emergence velocity"?**

The sentence was not clear and we apologize for this. What we meant is that glacier emergence velocity has to be accounted for when comparing thinning rates to melt rates. We have rewritten the sentence ("*Models can also provide actual melt rates while geodetic studies only provide glacier thinning rates, which are affected by glacier emergence velocity*", p. 3, lines 9-10).

**page 4, line 26: . . ., Kimoshung Glaciers. . .**

We have corrected this.

**page 4, line 31-32: Please consider to shorten this kind of sentences. It should be OK to write ". . . are exceeded most part of the debris-covered area (Ragettle et al., 2015). Relatively thin debris layer appears only near the equilibrium line."**

We have revised the sentences as suggested (p. 4, lines 30-32).

**page 5, line 6: a.s.l.**

Ok. We now consistently use a.s.l. instead of asl

**page 7, line 9: ALOS PRISM**

Ok.

**page 8, line 5: What is "correlation score"?**

During the automatic DEM extraction, image correlation is used to extract matching pixels in two overlapping images. The correlation score indicates if pixels have been matched successfully. We have changed the sentence as follows: "The correlation score maps, *indicating which pixels have been matched successfully during the DEM extraction process*, are used to exclude all DEM grid cells with a correlation score below 0.5." (p. 8, lines 9-11).

**page 8, line 9: Either of "older" or "earlier acquisition date" is fine.**

Ok. We now simply state 'the older DEM' (p. 8, line 20).

**page 9, line 12-13: I understand that these parameters are useful to measure spatialnon-uniformity in the melt rate. However, I do not understand why you use both of them. Particularly, the second one needs a reason why you take 50% and 10%. Moreover, why not using the information on cliffs and lakes delineated from the satellite image (Figure 2b)?**

This was a good comment and we have revised our approach accordingly. Quality-checked cliff and lake inventories have been set up based on the available satellite imagery for the period 2006-2015 (Section 3.3). Those inventories are now used in the revised manuscript to directly relate cliff/lake area to local thinning rates. The two cliff proxies are not used anymore.

**page 10, line 16: I wonder why "higher accuracy" can be the reason to apply the higher threshold.**

We show that the standard deviations ($\sigma$) are lower over flat and non-snow covered terrain (Figure 3 of the original manuscript and Figure S1 in the Supplement of the revised manuscript). In the accumulation areas, on the other hand, the presence of many outliers leads to higher standard deviations in the elevation differences. Accordingly, with a $3\sigma$-threshold we identify outliers at debris-covered terrain, whereas over featureless and steep terrain a lower threshold is necessary for efficient outlier detection. This is now better explained in the revised manuscript ("*Above the ELA, steep terrain or featureless snow surfaces lead to low DEM accuracy and therefore the outlier criteria should be more restrictive (e.g. Pieczonka et al., 2013; Pieczonka and Bolch, 2015).*" (p. 9, lines 20-22).

**page 11, line 11: Why do you use the thinning rate from 2006 to 2015 as a threshold?**

The thinning rate 2006-2015 was used as a threshold to guarantee that the signal to noise ratio is higher than 1:1, which indicates more signal than noise (whereas the thinning rate 2006-2015 is the signal and the DEM adjustment uncertainty is the noise). However, the outlier criterion discussed here is not used anymore in the revised manuscript and the selection of $\Delta h/\Delta t$ maps for the ensemble is now based on simple and straightforward criteria (see 'General Revisions' above).

**page 11, line 18-page 12, line 18: It is hard to understand the concept and the procedure to obtain U_cadj. If this is a commonly used parameter, please provide a good reference. I recommend the author to describe this kind of details in supplement.**

To compute triangulation residuals is a quite common approach for DEM co-registration quality assessment (e.g. Paul et al. 2015). However, we agree that the presentation of the method is lengthy and the results are not much discussed in the paper. Since we are not using this quality proxy anymore for outlier detection and potential errors due to co-registration are covered by our uncertainty estimates (see our response above to the reviewers $2^{nd}$ main comment) we removed all figures and text regarding triangulation residuals from the manuscript.

**page 13, line 12-13: I wonder how these thresholds were chosen and why they "effectively minimize the uncertainty".**

These thresholds minimized uncertainty because they allowed detecting outliers regarding mean elevation differences (MED). However, these thresholds and MED are not used anymore for outlier detection in the revised manuscript.

**page 14, line 5: Using three characters as a symbol is not common. By the way, do you need to define this symbol "unc"?**

We now use $E_{\Delta h}$ following Gardelle et al. (2013)

**page 14, line 14: Do you use the same density in the accumulation area?**

Yes. This is a common assumption in geodetic mass balance studies (e.g. Bolch et al., 2011, Gardelle et al. 2013).

**page 15, line 12: Should be "92 maps were removed because they FULFIL outlier criteria"?**

Ok. This text is now redundant and has been removed from the manuscript since we do not perform an outlier correction at the glacier scale anymore.

**page 15, line 22: Define the acronym "RPC".**

The acronym indicates Rational Polynomial Coefficients. However, we have removed the corresponding sentence to shorten the manuscript.

**page 17, line 5: Please be consistent with the unit, m/a or m a-1.**

Agree. We now consisently use m a$^{-1}$.

**page 17, line 29-30: "but a majority of values suggest that . . .." » This is not very sure from the data. It appears to me that the thinning rate is decreasing recently**

After applying the new outlier correction and uncertainty estimation procedures the data is now less ambiguous about mean thinning rates at Shalbachum Glacier (where indeed the thinning rates seem to have increased, see new Figure 8c and p. 16, lines 8-10). At Ghanna Glacier the data are still unclear. We now state that the ensemble uncertainty is too high to draw any conclusions regarding thinning trends at Ghanna Glacier (p. 16, lines 15-20).

**page 18, line 1: What do you mean by "ensemble of values"?**

This is an important point and we explain this clearly in the revised manuscript (Section 3.2.5). By 'ensemble of values' we mean the ensemble of observations available for overlapping periods. In the revised manuscript we use the data ensemble available for the period 2006-2015 more systematically to identify a sound signal and narrow down uncertainty. See our general response at the beginning of this document.

**page 18, line 30: What about simplifies the sentence to "The most negative elevation change for 1974-2006 was observed at Shalbachum . . ...".**

We have revised the sentence as suggested (p. 16, line 21).

**page 19: line 2-3: It makes more sense to compare 1974-2006 and 2006-2014 to eliminate the influence of the earthquake.**

Here we referenced to the values reported in Table 5 where the pre-earthquake February 2015 DEM was considered. There was a mistake in the manuscript text (it should have been "Comparing the two periods 1974-2006 and 2006-*Feb* 2015"). However, we think the October 2015 DEM is valuable to discuss multi-annual ($\Delta h > 4$ years) elevation changes and should not be excluded from the ensemble. See our comment to the reviewers' main comment 3 above.

**page 19, line 1: m a-1 » You need a space between m and a-1.**

Agree.

**page 19, line 9: "The most important differences in mean Δh/Δt values. . ." » "The greatest increase in thinning rate . . ."?**

We have removed this sentence and entirely rewritten the corresponding paragraph (from p. 16, line 28, until the end of Section 4.2) .

**page 19, line 15-21: I find this paragraph is not necessary here. Because Figure 8 clearly shows the thinning patters, you do not need to give questionable comment on Figure 6.**

Ok. We agree and we have removed the paragraph as suggested.

**page 19, line 32-page 20, line 3: This sentence is very hard to read. Please consider to rewrite it.**

We agree that the sentence was too long. We have rewriten the paragraph as follows:

*"On Langshisha Glacier (Figure 8b) near the terminus, the comparability of 1974-2006 thinning rates with the 2006-2015 ensemble is limited. Here, the glacier tongue became very narrow in the last decade and ultimately a small part below 4500 m a.s.l. disconnected from the main tongue (Figure 1) between 2010 and 2014. The fragmentation of the tongue leads to mean thinning rates close to zero at elevation bands where a substantial part of the glacier area disappears during a given time interval."* (p. 18, lines 12-17). **page 21, line 30: "Δh/Δt_1974-06-Δh/Δt_2006-15<-0.2 m/a" » Is this correct? Isn't the left side positive if the thinning is accelerated?**

The reviewer is right here. It should have been "Δh/Δt_2006-15 - Δh/Δt_1974-06". We have rewritten the paragraph (p. 21, lines 22-27). In the caption text of new Figure 12 we now state "Negative Δ(Δh/Δt) values represent thinning accelerations".

**page 22, 5.3. Impacts of the April 2014 earthquake: This is an interesting analysis. I suggest the author to use the DEM after the earthquake only for this purpose. In other words, elevation change from 1974 to 2014 should be used for the rest part of the discussion.**

We agree that the May 2015 DEM should be used only for this purpose. However, the long-term impacts of the post-earthquake avalanches (Δh > 4 years) are already negligible in October 2015 and we thus use the October 2015 DEM also for the long-term comparison. See our statements above.

**page 22, line 2-8: This should be explained in the introduction section.**

We have removed those lines from the results section. We have added the reference to Kargel et al. (2015) to the introduction (p. 4, line 17).

**page 22, line8-21: This should be explained in the method section.**

Ok. We have followed the reviewers' advice (see new Section 3.5).

**page 23, line 5: "compensated by about 50%" » What density do you assume for the avalanche debris deposition?**

Here we only discussed volume changes and no mass changes. We therefore did not make any assumption about density. However, we have added to the paper an estimation of mass change due to the post-earthquake avalanches. According to Scally and Gardner (1989) avalanche deposit density

increases until the end of the ablation season to about 720 kg/m$^3$. Considering this value and a density of ice of 900 kg/m$^3$, the mass deposits compensate by about 40% for glacier mass loss during an average year (p. 25, lines 1-4).

**page 23, line 12: "Elevation changes in the debris-covered area are primarily independent of elevation (Figures 8 and 10c) as previously identified in Langtang catchment (Pellicciotti et al., 2015) and elsewhere . . ..."**

Thank you. We have revise the sentence as suggested (p. 22, lines 3-6).

**page 23, line 16: "downward-" » downglacier?**

Ok. We have replaced 'downward' by 'downglacier' (p. 22, line 7).

**page 24, line 1-2: Not clear where and how water pressure is elevated.**

The delivery of surface-generated meltwater to the en- and subglacial environment is the driver of raising water pressure. Likely, lake formation itself can be attributed to enhanced englacial water pressure. It is therefore sufficient to state: "Such stresses are usually not large enough to initiate open surface crevasses, but in combination with elevated water pressure *due to local water inputs* lead to hydrologically driven fracture propagation (hydrofracturing) and englacial conduit formation (Benn et al., 2009)" (p. 22, lines 22-27).**page 24, line 5-6: Do you mean that thinning accelerated where ice motion is active because cliffs and lakes develops? It contradicts to my experience to observe cliffs and lakes formation on debris-covered stagnant ice.**

Yes, with the revised Figure 12 we can show clearly that especially cliffs appear more frequently where the glacier is not stagnant. This is not contradictory to previous studies. Several studies have shown that ice cliffs on depris-covered glaciers in the Himalaya appear most frequently in the transition zone between the active and inactive glacier parts (Sakai et al., 2002; Bolch et al., 2008; Thompson et al., 2016). The appearance of supraglacial lakes, on the other hand, is strongly related to the surface gradient. Large supraglacial lakes can only form where the slope is less than 2° (Reynolds, 2000), and the largest supraglacial lakes in the Himalaya form near the terminus of glaciers where a terminal moraine prevents free drainage of meltwater (Benn et al., 2012). The large debris-covered glaciers in the Upper Langtang catchment, however, have not reached this regime yet. We have added a paragraph to discuss the conditions that lead to supraglacial lake appearance (starting from p. 22, line 28).

**page 24, line 20: "glacier uplift" » do you mean "ice thickening due to compressive flow regime"?**

Yes. Uplift of ice occurs by convergence of the ice flux. However, we now state that "it can be assumed that *a slowdown of the compressive flow regime* is not the primary factor that *causes* the observed thinning accelerations" (p. 23, lines 20-21), which is more precise.

**page 23, line 27-page 25, line 11: The goal of this section is not clear. It appears that this section discusses the mechanism of surface elevation change on debris-covered ice. However, the thinning rate is highly variable in space and time, and there is no general trend in the observed glaciers. What kind of results does the author try to explain here? Many processes related to surface elevation change of debris-covered glaciers are described, but none of them are connected to reliable interpretation of the data. Describe first an observational fact that you want to discuss, and interpret the observation in a logical manner.**

This is a good comment, and we have substantially revised this section on the basis of the reviewers' comment. This entire section was based on the results presented in the old Figure 10. With the scatterplots and correlation coefficients we attempted to explain the variance in thinning rate changes of all debris-covered glaciers at the same time. However, single variables cannot explain all the spatial variation in thinning rates, and therefore the r values were generally low (max. 0.55 for debris-covered glaciers). We therefore removed the old Figure 10 and now show the most important variables (cliff area, lake area, surface velocity and changes in thinning) separately for each glacier (new Figure 12).

We now use a *new dataset of cliff and lake areas*, which substitutes the cliff/lake proxies. The new Figure 12 and the new Table 8 allow for a more reliable interpretation of the data. The main 'observational facts' that are presented  are i) the relation between thinning acceleration and active glacier dynamics (given by the glacier velocity) at all debris-covered glaciers except Lirung, and ii) the relation between supraglacial cliff area and thinning acceleration (see new Section 4.5).

**page 25, line 16-17: What do you mean by "correlate with"? Which data show this?**

What we meant here is that one can expect glacier mass balances near steady state if the glacier area remains nearly constant, as it is the case for Kimoshung Glacier. We have removed the sentence to shorten the manuscript. Area changes are now presented in the new section 4.4.

**page 25, line 13-page 26, line 9: The first part of this section 7.1.1. explains that thinning accelerated at Yala Glacier, whereas it appears to be at a similar level at Kimoshung Glacier. This kind of explanation should be completed in Result section. Interpretation on the difference begins at page 25, line 28, but not convincing because there is no qualitative discussion. For example, hypsometry is not shown for Kimoshung Glacier, and no information about the 0 degree C isotherm altitude.**

The first part of section 7.1.1 (now section 5.2) has been completely removed from the discussion section. Glacier area changes are now presented in a separate results section (Section 4.4).

We guess the reviewer wanted to say that a quantitative discussion was missing (since the discussion was essentially qualitative). In the revised manuscript we thus show that variations in the estimated equilibrium line altitude (5400 m a.s.l.) of $\pm$ 100 m lead to an AAR variation of 13%-70% at Yala Glacier, but at Kimoshung Glacier only to a variation of 80%-88% (Table 6). This explains why Yala Glacier is much more sensitive to warming than Kimoshung Glacier. Consequently, thinning at Yala Glacier accelerated from -0.33 $\pm$ 0.06 m a$^{-1}$ (1974-2006) to -0.89 $\pm$ 0.23 m a$^{-1}$ (ensemble mean 2006-2015), while it accelerated only insignificantly at Kimoshung Glacier from +0.07 $\pm$ 0.13 m a$^{-1}$ to -0.02 $\pm$ 0.17 m a$^{-1}$ (Table 5). Hypsometry of Kimoshung Glacier is now shown by the new Figure S5 in the Supplement. We think that the large differences between Yala and Kimoshung Glaciers are now presented clearly and that the interpretation of the differences is now convincing.

**page 26, line 11-page 27, line 11: This section has the same problem as section 7.1.1. The first paragraph describes several different observations. These details should be explained in Result section, and here the focus of the discussion should be stated briefly. In the second paragraph, speculative conclusions are given without detailed/quantitative comparison with the modeling work.**

We have removed this section (section 7.1.2. in the original manuscript) and merged some of its content with section 5.1 ('Elevation changes of debris-covered glaciers'). We partly agree that some of our previous conclusions were speculative, since the differences in retreat rates between Ghanna Glacier and other debris-covered glaciers are not significant. We therefore removed this comparison

from the manuscript. The remaining text (moved to section 5.1; p. 23, line 26, until p. 24, line 9) summarizes the current theoretical knowledge about the dynamical response of debris-covered glaciers to rising air-temperatures. Our conclusions are firmly supported by our data (thinning rates near the fronts of the large debris-covered glaciers in the valley indeed have not yet started to significantly decrease, Figure 12a-c, and the glacier tongues are indeed still dynamically active, Figure 13).

**page 27, line 18-20: Not clear what are compared in Figure 6b.**

We referred to Figure 6b because the figure shows also the thinning rates of Kimoshung Glacier. We have removed those lines and now base our discussion on the new Figure S5 (p. 26, lines 9-17). The figure directly compares thinning profiles at Kimoshung and Yala Glaciers.

**page 27, line 24-25: Not clear why you compare the elevation of Yala terminus and that of maximum thinning on Langtang.**

We apologize if this was not clear. The main point here was to compare lowering rates of debris-free and debris-covered glacier area at the same altitudinal range. Since debris-covered glaciers reach much lower elevations, a comparison is only possible from 5150 m a.s.l. upwards. The elevation of maximum thinning on Langtang Glacier coincides with Yala terminus elevation, but this is likely just a coincidence and should not be our main point here. We have thoroughly revised this paragraph to be clear (p. 26, lines 19-29).

**page 27, line 26-28: It is not clear which part of the elevation range is compared here. If you discuss elevation change of debris-covered and debris-free glaciers at same elevation range, why not preparing a plot for this purpose?**

Here we compared the thinning rates near the terminus of Yala Glacier (5150-5200 m a.s.l.) to the thinning rates at Langtang Glacier at the same elevation. As stated in our comment above, we have thoroughly revised this paragraph to better explain the differences between debris-covered and debris-free glaciers (p. 25, line 19, until p. 26, line 8). If the reviewer asks for it we can also add Figure R2 below to the Supplement (comparing thinning profiles of Yala Glacier to thinning profiles of Langtang Glacier tongue). However, the same information is provided by Figure 8. We therefore do not think an additional figure is required here.

[Figure]

**Figure R1. Elevation change profiles of Langtang Glacier tongue (debris-covered) and Yala Glacier (debris-free). The figure shows the ensemble-median results for the period 2006-2015, and error bars represent the ensemble-uncertainty.**

**page 28, line 3-15: The point of the discussion is unclear.**

We apologize for the lack of clarity. We have improved the clarity of this paragraph by rewriting some of the sentences. The first important point here is that the magnitude of this thinning increase is very different from glacier to glacier ("*there are examples ... of glaciers where thinning has increased significantly or where thinning remains approximately constant*", p. 26, lines 20-21). Then we state that "*A significant difference in thinning trends between debris-free and debris-covered glaciers in our sample cannot be identified*". This is why we conclude that "*our observations reveal a heterogeneous response to climate of both the debris-free and the debris-covered glaciers*" (p. 26, lines 18-19). Finally, we suggest that the altitude distribution of glaciers likely plays a more important role for average thinning rates than debris-cover alone (p. 26, lines 23-29). In the second part of the paragraph we have added quantitative details about the differences in glacier characteristics to make sure our conclusions are firmly supported by our data.

**page 28, line 16-17: Not clear what you mean. Do you mean that your result support the studies by Kaab, Nuimura and Gardelle?**

No, our results do **not** support the observations by Kääb, Nuimura and Gardelle, since our observations do not show that the lowering rates of debris-covered glacier area are similar to those of debris-free areas at the same elevation. We have stated this clearly on Page 25, line 31, until p. 26, line 8. On the other hand, we do not observe a significant difference in the overall mass balance trends of debris-free and debris-covered glaciers in our sample (see our previous comment above). Here we wanted to say that these two observations are not contradicting. A comparison of thinning rates at the same elevation is not representative of average thinning rates due to the differences in altitude distribution. However, since this is rather obvious we have removed those lines to shorten the manuscript.

**page 30, line 17-page 32, line 14: Only a few data appear in Conclusion section, which results in very qualitative descriptions. This represents the weakness of the paper. Please draw your conclusions which are supported by data.**

The reviewer is right that some more quantitative measures should be provided in the conclusion section. We have done this in the revised manuscript (p. 29, lines 11-12; p. 29, lines 29-31). A summary of quantitative outputs with respect to the three main goals is provided at the beginning of the revision statements to this reviewer. The summary and the revised conclusion section show that we are able to draw relevant conclusions firmly supported by our data.

**page 52, Figure 11: This velocity map is not much used for the study. Judging from the vectors on the plot, it is not sure how much this analysis is reliable.**

Velocity is a key variable to discuss thinning patterns on debris-covered glaciers (see revised Figure 12). We think the data are reliable, especially since the vectors consistently point in down-glacier direction. It is not clear to us how the reviewer came to a different conclusion regarding the vectors. The revised Figure 13 shows larger vectors, so we hope this is now clearer.

[revised manuscript text omitted]
}$ provides an estimate of the co-registration uncertainty. Having $n$ DEMs available, area average elevation differences can be calculated $C_{k-1}$ times by employing $k$ DEMs (and therefore $k-1$ DEM differencing steps):

$$C_{k-1} = \frac{\binom{n}{k}}{2} N_{\Delta t}^{-1} \tag{2}$$

$N_{\Delta t}$ is the number of possible two-fold combinations of $n$ DEMs (equation 1). According to equation (2), having $n=8$ DEMs available, the difference between two DEMs can be determined six times by adding or subtracting the $\Delta h$ values using a third DEM ($k=3$). Using $k$ equal to 2, $C_1$ is equal to 1 according to equation (2). The DEM adjustment uncertainty ($U_{adj}$) is then calculated as follows:

[revised manuscript text omitted]

---

## Author Response (AR2)

**REVISION STATEMENT**

**"Heterogeneous glacier thinning patterns over the last 40 years in Langtang Himal"**

**by S. Ragettli, T. Bolch and F. Pellicciotti**

**IN RESPONSE TO THE EDITOR**

**The methodology and the description of the results have been improved. The weakest part of the paper remains the discussion. It is long and contains some statements unsupported by the data itself.**

**One striking example of this lack of coherence is:**

**P26 L22 "A significant difference in thinning trends between debris-free and debris-covered glaciers in our sample cannot be identified" and then**

**P26 L32 "our results indeed point to a difference in current volume loss of debris-free and debris-covered glaciers"**

**With two contradictory statements ten lines apart, the reader is left without any clear message.**

**There are also several occurrences where the authors discuss what "glacier could have done but did not do". One example is P24 L16-20: discussion of a terminus advance that has not been observed! Authors could discuss so many things that their study glaciers did not do… It makes the discussion long and lost the reader.**

We thank the editor for his detailed comments. We have followed the editors' advice to increase the coherence of the discussion and to remove unsupported statements. Our comment in response to the editors' examples above is provided in our detailed responses below. We have removed the statement that our "results indeed point to a difference in current volume loss of debris-free and debris-covered glaciers". We agree that this statement could not be well supported by our data.

**Abstract. TC has no specific requirement for the length of the abstract. A good target (set by JGlac) is 200 words. 250 words is a maximum I think.**

We have shortened the abstract and it includes now less than 250 words.

**P4 L32. Why "However". Not real opposition.**

We have removed 'however'.

**P7 L13. DEMs already defined in abstract. Maybe define DEM for its first occurrence in the main text?**

We have removed the definition of DEM here and define DEM now at its first occurrence in the main text (P.3, line 14).

**P8 L25. Avoid long parenthesis. Make a separate sentence instead. Parenthesis break the flow of reading.**

Ok. We now make a separate sentence instead.

**P9 L26. "A stricter criterion for the accumulation area is also justified by the fact that it can be assumed that elevation changes in the accumulation areas over periods of several years are small". The fact that elevation changes are small is not a good reason. Small is not synonym of homogeneous.**

This statement is needed. Elevation changes are usually small in the upper accumulation areas. This finding is published and well justified (Schwitter and Raymond, 1993; Huss et al., 2010). We are not sure we understand the editor's comment, as we nowhere suggest that they are homogenous. In addition, restrictive outlier criteria are justified by low DEM accuracy due to steep terrain and featureless snow surfaces.

**P11. L26ff Presentation of the "ensemble mean" could be improved. Why counting all dh/dt maps (including the one using the 1974 DEM) if in the end only one is used for 1974-2006 and only a few of them for 2006-2015? Describe right away (i) the choice for the 1974-2006 time period (but then in some figures 1974-2009 is sometime shown…) and (ii) that the redundancy of information between 2006 and 2015 allow extracting N maps of dh/dt with a time interval larger than 4 years**

We have revised section 3.2.5 as suggested. We are not counting anymore all dh/dt maps and consequently have also removed equation 1 from section 3.2.2.

The period 1974-2009 is shown in figures 9 and 10 (thinning profiles) just to illustrate that the differences between 1974-2006 and 1974-2009 are small and that it is therefore not necessary to consider an ensemble of dh/dt maps for the long period. We think it is therefore useful to show the 1974-2009 results in figures 9 and 10 but if the Editor still thinks this is confusing for a reader we can still remove them.

**P13 L5. Outlines**

Thank you for noticing. We have corrected this.

**P14. Section 3.5. The threshold of 5 m seems high such that some regions of moderate elevation gain may not be accounted for. Sensitivity? How did the authors choose 5 m? What happen if 2 m is used instead of 5 m?**

We agree that a threshold of 5m is rather conservative. However, elevation changes on debris-covered glacier area are highly heterogeneous. Positive elevation changes of 2-5 m are not uncommon due to lake filling, cliff movement, uncertainties in the DEM etc. Figure 2 below illustrates that a 2 m threshold identifies many pixels as avalanche affected that are isolated and not adjacent to the avalanche cones. This is why we chose 5 m.

We carried out the suggested sensitivity test and found that the differences in elevation changes provided by Table 8 in the paper are mostly within the indicated uncertainty ranges (see table below).

The calculated avalanche volume on 7 May 2015 increases by 34% when using a threshold of 2 m instead of 5 m. Avalanche volume on 6 Oct 2015 increases by 31%. Approximately 40% of avalanche material remains until the end of the ablation season according to these numbers, regardless of the threshold used (see table below). This means that 40% of the material from large avalanche cones and 40% of 2-5 m initial avalanche material remains until 6 October 2015. However, one can assume that shallow layers of avalanche material should disappear almost completely over one ablation season. Since this is not the case according to the numbers above, the results strongly suggest that a large fraction of pixels with 2-5 m elevation gain would be misclassified if counted as avalanche area. We therefore prefer to consider only a threshold of 5 m which considers "*all glacier grid cells with significant positive elevation changes*", as stated in the paper.

Reviewer 2 states that the paper already "contains too much detail". However, if the editor finds it appropriate we can provide some of the explanations above in the paper.

**Table 1. The same variables as in Table 8 of the manuscript but showing the range of values obtained by using a threshold between 2 and 5 m to identify avalanche affected pixels in May 2015.**

|  | 21 Apr 2014-25 Apr 2015* (m) | 25 Apr 2015-7 May 2015 (m) | 25 Apr 2015-6 Oct 2015 (m) | 6 Oct 2015, volume remaining (%) |
|---|---|---|---|---|
| Langtang** | -0.2 - -0.1 ± 0.05 | 1.3-1.8 ± 0.4 | 0.4-0.6 ± 0.2 | 31% - 33% |
| Langshisha | -0.04 - -0.1 ± 0.05 | 0.3-0.7 ± 0.4 | 0.1-0.3 ± 0.2 | 32% - 38% |
| Shalbachum | -0.1 - -0.2 ± 0.05 | 0.7-1.1 ± 0.3 | 0.3-0.4 ± 0.2 | 36% - 42% |
| Lirung | -0.9 - -1.1 ± 0.06 | 6.8-7.3 ± 0.4 | 3.9-4.0 ± 0.2 | 54% - 57% |
| Average | -0.1 - 0.2 ± 0.05 | 1.3-1.8 ± 0.3 | 0.5-0.7 ± 0.2 | 39% - 40% |

*Estimation based on average annual melt Oct 2006 – Apr 2014

**Only lower part (south of 28°19'N), upper part not on April 2014 scene

[Figure]

**Figure 1. Red pixels identify debris covered areas with positive elevation changes (Δh) exceeding 5m between April 2014 and May 2015.**

[Figure]

**Figure 2. Red pixels identify debris covered areas with positive elevation changes (Δh) exceeding 2 m between April 2014 and May 2015.**

**P14 L24. "the Apr 2014 differential DEMs". Unclear! Diff DEM with only one date.**

We have changed this to "differential DEMs involving the Apr 2014 scene". We hope this is now clear.

**P16 L10. Mixed of thinning rates and volume loss rate in the same sentence. They cannot be compared they do not have the same unit.**

We are only comparing thinning rates here and therefore have changed "volume loss rates" to "thinning rates".

**P16 L28. "Volume change rate" but your figure show dh/dt. More rigor is needed in the use of the terminology**

We have changed "volume change rate" to "elevation change rate".

**P17 L14 Section 4.2.1 but no section 4.2.2. Not logical**

We have changed sub-subsection titles to subsection titles. The section has been renumbered to 4.2.

**P18 L23-L29. Unclear structure. Authors start with Yala then "This is in clear contrast to the much less uniform patterns at debris-covered glaciers" and then they come back to describe the change for Yala. Try to make the structure of the paragraphs more logical.**

We have merged this paragraph with the following paragraph to make the structure more logical and we have shortened the text. We now state that "*On Yala Glacier there has been a three-fold increase in thinning rates below 5400 m a.s.l, comparing 1974-2006 to the 2006-2015 ensemble results (Figure*

*10d). Maximal thinning takes place at the terminus and then decreases nearly linearly with altitude until it reaches values close to zer). This is in clear contrast to the much less uniform patterns on debris-covered glaciers (Figure 10a-c)."* and then continue to describe the changes for debris-covered glaciers.

**P19 L8. I have a hard time reconciling the figures 8a and 11a for Lantang. Increase in dh/dt with time is much more clear in 11a than in 8a.**

We agree that the thinning accelerations are clearer in Figure 11a than in Figure 8a (in the revised manuscript now figures 10a and 9a). This is because the elevation profiles shown in Figure 11 also consider debris-covered tributary branches. For Figure 9 we only consider the main tongues excluding tributary branches (as stated in the caption text and as shown in Figure 6). The difference between the two figures shows that thinning accelerations are important on the tributary branches, probably because debris is much thinner there. We can comment on this also in the manuscript if the editor suggests so.

**P20 P29 long parenthesis**

We have added a sentence and removed the parenthesis.

**P24 L16-20. Same as above: "Authors could discuss so many things that their study glaciers did not do"**

We think that the possibility of terminus advances of debris-covered glaciers during periods with higher temperature is a fascinating fact but the editor is right that our glaciers do not show such a behavior. We have therefore removed those lines.

**Section 5.1.1 but no section 5.1.2. Not logical**

We have changed sub-subsection titles to subsection titles. The section has been renumbered to 5.2.

**P25 L13. Delete "to"**

Ok. We have deleted "to".

**P25 L18. Why "global" warming? A glacier respond to the local climate not the global one**

We have replaced "global warming" by "atmospheric warming".

**P25 L19. "The balanced conditions of Kimoshung Glacier therefore indicate that precipitation in recent decades remained approximately stable". This is a really, really indirect indication. Can you rule out that the effect of warming has been offset by an increase in precipitation? Only a modelling exercise could prove the differential sensitivity to temperature/precipitation and make the attribution to one factor credible. Again an occurrence where authors conclude too much out of their data…**

Our sensitivity test shows that AARs of Kimoshung Glacier are not sensitive to ELA changes, due to its very steep tongue. The effect of warming has therefore not been offset by a precipitation increase as the editor states, but this effect of warming is less strong on Kimoshung Glacier. Changes in glacier mass balance over time at Kimoshung Glacier therefore have to be attributed mainly to precipitation changes in our opinion. We therefore do not agree to remove the sentence entirely as this is an interesting and relevant possibility should be included as possible explanation based on expert judgment. However, we have rewritten the sentence and now state that *"One possible explanation for*

the balanced conditions of Kimoshung Glacier *could therefore be* that precipitation in recent decades remained approximately stable, which agrees with the findings of studies on precipitation trends in this part of the Himalaya (Shrestha et al., 2000; Immerzeel, 2008; Singh et al., 2008). *However, further analysis is required for justification.*"

**P25 L25-27. Again a lot of speculation in this statement**

We have changed the sentence as follows: "*Due to the common AAR of Yala Glacier and the extreme topography of Kimoshung Glacier it can be assumed that other debris-free glaciers in the region are also thinning and that balanced conditions such as observed on Kimoshung Glacier are exceptional*." We think this sentence does not involve too much speculation, since we show that Kimoshung Glacier is indeed an extreme case in terms of AAR.

**P26 L3 "of" missing**

Ok. Thank you for noticing.

**P26 L9-11. Authors need to clarify what they want to show here**

Here we discussed the thinning rates on Kimoshung Glacier in comparison to debris-covered glaciers. We have removed this paragraph to simplify and shorten the paper. The editor was right that the point of the discussion was not perfectly clear.

**P26 L14-17. If a glacier is in equilibrium with the climate, with no elevation change in the accumulation/ablation area, the emergence velocity (of divergence of the flux, I guess this is what the authors mean by compressive flow) compensate for the surface mass balance WHATEVER the surface slope is. So I do not follow your reasoning.**

We agree with the editor that the balanced conditions should be regarded as the main reason why thinning rates on Kimoshung Glacier are low across all altitude bands. We apologize for having forgotten to mention this. We have removed this paragraph (see our answer to the comment above), but have added two sentences to section 5.2 about the elevation range and thinning profile of Kimoshung Glacier ("*Kimoshung Glacier has a very steep tongue that reaches similarly low elevations as the debris-covered glacier tongues (Table 1). The glacier is nearly in equilibrium with the climate (Table 4), which explains the low thinning rates at low elevations (Figure S5).*")

**P26 L21-22 and then L32ff. Two contradictory statements in the same paragraph.**

We agree that this paragraph needed to be improved. While the first statement is based on our data, the second statement was speculative. We have thus removed the statement that our "results indeed point to a difference in current volume loss of debris-free and debris-covered glaciers". Still, we think it is important to mention that the response to climate of Yala Glacier might be indicative of larger samples of debris-free glaciers, due to its common characteristics, and that future studies should follow-up on this. We have therefore revised the second part of the paragraph as follows: "*Considering the common characteristics of Yala Glacier and given that this glacier has been denominated as a benchmark glacier for the Nepal Himalayas (Fujita and Nuimura, 2011) it seems important that future geodetic or field based studies extend our analysis to larger glacier samples.*"

**P27 L13. Lot of speculation. A better reasoning would be to compute the penetration depth needed to reconcile the two estimates and then discuss if this is in agreement with values proposed in Kääb et al.**

It is probably naïve to believe that only the uncertainty in the penetration depth explains the difference in the values obtained by our study and by Pellicciotti et al. What matters here is that larger penetration depths as suggested by Kääb et al. would correct the values obtained by Pellicciotti towards less negative mass balances, in agreement with our findings. However, the uncertainty about penetration depth is only one of the many potential error sources whose correction could permit to reconcile the two estimates. In our opinion it would not be meaningful to compute the penetration depth as suggested above, since it implies the wrong assumption that the penetration depth is the only source of error. To simplify and shorten the manuscript (following the advice of reviewer 2) we have removed the two sentences about reconciling the estimates with the penetration depth and have revised the following sentence as follows: "Differences in the mass balance of Langtang, Lirung and Shalbachum Glacier are within uncertainty bounds and can be attributed to differences in used glacier masks, study period, outlier correction approaches, density assumptions *and uncertainties regarding the penetration depth of the SRTM radar signal (Kääb et al., 2015).*"

**P27 L25. What are -0.4 to 0.4 m? A range of extreme values?**

Yes, this is the range of obtained values that can be compared to the value of -2 m a$^{-1}$ obtained by Pellicciotti et al. We have clarified this in the text (P. 26, lines 7-8).

**P28 L4ff. This comparison is not really useful because the compared time periods is so different. In the end what do we learnt/conclude? Not much.**

We agree that only very limited conclusions can be drawn from this comparison and we have therefore removed the paragraph to shorten the manuscript.

**P28 L14. Repeated from the introduction. Keep only for the discussion I think.**

We agree that it is not necessary to provide the values obtained by Bolch et al. (2011) and Nuimura et al. (2012) both in the introduction and in the discussion. To avoid repetition we have removed this part from the introduction.

**P29 L13-16. 2 methodological conclusions in the middle of the glaciological results. I suggest skipping or organizing differently.**

We have removed the two sentences with methodological conclusions as suggested.

**P29. L25. Authors draw some conclusions from a sample of two debris free glaciers. The linearity of dh vs. altitude was shown for half of their sample (1 glacier). What about for Kimoshing? Not shown I think.**

The dh altitude profile of Kimoshung Glacier is shown in Figure S5. We have slightly revised the sentence to make clear that the conclusions are valid only for our relatively small glacier sample ("Debris-free glaciers *in our sample* present thinning rates that are linearly dependent on elevation, while debris-covered glaciers have highly non-linear altitudinal elevation change profiles."). It is stated at the beginning of the introduction that our study region only includes two debris-free glaciers. We therefore think that all our conclusions here are justified by our data and are presented adequately.

**IN RESPONSE TO REVIEWER 2**

**This paper is substantially improved from the previous version. I concerned about too much complicated data screening process in the previous manuscript, but now the methodology is simplified and presented in a more understandable way. Uncertainties in the results are very carefully analyzed, which enabled quantitative discussion on the three main research questions, acceleration in glacier thinning, spatial patterns and influence of debris cover, and the impact of the 2015 earthquake. The results are described in detail with carefully prepared plots and tables. The discussion and conclusion are useful not only for researchers on Himalayan glaciers, but also those who uses same techniques for glacier surface elevation change.**

**I understand that the authors tried to describe the work very carefully. However, in my opinion, it contains too much detail. For example, ALOS PRISM image is listed and described as satellite data used in this study, but actually they did not use it in the analysis and explain why they were excluded. Detailed and frequent descriptions on the uncertainty often get into the way of understanding of more important subject. It is tough to read and try to understand every detail, which are not necessary to catch the main points of the paper. I also think the text is lengthy in general. The manuscript will be improved by using simpler and more straightforward expression. I also find several paragraphs are not in a right position. For example, starting Result section with the impact of 2015 earthquake is odd, because it forms the third main research question and occurred at the end of the study period.**

**Below, I list my comments and suggestions for consideration by the authors to improve the paper before publication.**

We thank the reviewer for his useful comments. We agree that the quality of the manuscript could be improved by using simpler and more straightforward expressions. We also agree that several details could be excluded from the text since they were not necessary to catch the main points of the paper. In this respect both the editor and the reviewer have provided valuable recommendations where the paper could be shortened or simplified.

Regarding the ALOS PRISM scene, however, we do not agree that it was not used in the analysis and that it was not explained why it was excluded. Like all other scenes, the ALOS PRISM scene was used to calculate elevation changes. The ALOS PRISM DEM was then excluded from the 2006-2015 ensemble of elevation changes because of 30-100% higher uncertainties in dh/dt values. This is stated clearly in the text (P. 11, line 19). We think this is an important result that should be shown in the paper, especially since this study is one of the first for the HKH that presents a large set of DEMs extracted from different sources of optical stereo imagery. The reviewer states himself that our discussions are useful for "those who uses same techniques for glacier surface elevation change". In this respect, we do not agree that the description of the ALOS PRISM scene is not necessary to catch the main points of the paper or that it gets into the way of understanding of more important subjects. The ALOS PRISM scene is also still used in the paper to calculate surface velocities (section 3.4).

We have however revised the method section by providing a more concise and complete description of which elevation change maps were selected for the ensemble, which were discarded and for which reason (section 3.2.5). We are also not counting anymore all dh/dt maps since this is indeed not necessary to catch the main points of the paper (consequently we have removed equation 1 from section 3.2.2). We have also followed all the reviewers' advices regarding reordering of the paragraphs.

**page 1 Abstract:**

**>> Abstract is too long. Please focus on the main points.**

We have shortened the abstract as suggested.

**page 3, line 29 – page 4, line 6:**

**>> This paragraph should be merged to the next paragraph. Instead, here you can describe more about the advantages and problems of satellite derived DEMs and mass change measurements by DEM differentiation. This is because substantial portion of the text, figures and tables are dedicated to uncertainty estimation.**

We have merged the paragraphs as suggested. We have also added a few sentences about the uncertainties in satellite derived DEMs and mass change measurements by DEM differentiation in the HKH (P.3, lines 13-19)

**page 5, line 27:**

**>> Please consider to exclude ALOS PRISM from the text because it is not used for the analysis.**

We prefer not to exclude the ALOS PRISM 2010 scene from the text here. It is true that the DEM is excluded from the analysis of elevation changes, but the scene is still used to calculate surface velocities. We also carefully assess the uncertainties associated to all sets of elevation change maps, including those which are later rejected from the analysis. We think that the evaluation of uncertainties is a valuable result of our study, and the reviewer states himself that our discussions are useful for "those who uses same techniques for glacier surface elevation change".

**page 6, line 24: dGPS**

**>> Please make sure that the term "differential GPS" is correctly used. It is something different from static or kinematic survey with two receivers.**

We can confirm that the term differential GPS is correctly used. The dGPS measurements are described in detail in Brun et al. (2016).

**page 7, line 2: 17 GCPs**

**>> 17 GCPs off the glaciers?**

Yes. We now specify that these are 17 *off-glacier* GCPs.

**page 11, line 22: 850 kg m-3**

**>> Do you use this density in the ablation area and debris covered regions?**

Yes, this density is used throughout all glacier area as this volume to mass conversion number is well accepted in the international peer reviewed literature (but we consider also an ice density uncertainty of 60 kg/m3).

**page 12, line 28: an automated flow accumulation process**

**>> What is this?**

We have simplified the expression by just writing "flow accumulation" instead of "automated flow accumulation process". Flow accumulation is used to identify all cells flowing into each downslope cell, starting from the outlet of the catchment.

**page 14, line 2: using a window size of 128 down to 32 pixels,**

**>> What do you mean? "between 128 and 32 pixels"?**

"Multiscale window sizes of 128 down to 32 pixels" is the correct expression (Scherler et al., 2008, 2011; Ruiz et al., 2015). Multiscale window sizes allow the measurement of large displacements without losing resolution.

**page 14, line 16-21: To calculate the deposited volumes ….**

**>> Do you mean that "Volume change due to the earthquake by subtracting mean thinning rate between Oct 2006 and Feb 2015 from the volume loss between April 2014 and April 2015."?**

We have rewritten these lines to be clear: "*To account for pre-earthquake volume losses we first subtract from the DEM differences the elevation changes between April 2014 and April 2015 determined on the basis of the Oct 2006 - Feb 2015 mean thinning rates.*" (P. 14, lines 5-8)

**4.1. Impacts of the April 2015 earthquake:**

**>> It is odd to have this subsection in the beginning of this section. It is better described at the end of Results section as 4.5, because it addresses the third main research question (page 4, 12-14) and it occurred at the end of the observation period.**

We have moved this subsection to the end of the Results section as suggested (now section 4.6).

**page 15, line 4: Field visits**

**>> Field visits to which glacier?**

Field visits to Lirung and Langtang Glaciers. This is now specified in the text.

**page 15, 8-19:**

**>> I do not understand why October 2015 DEM was used for the 2006-2015 ensemble but May 2015 DEM was not considered. DEMs in May and October 2015 are both influenced by the avalanche by the same amount of 1.31 m. The deposition in October is less than that in May. This is just because it spent one ablation season, and it does not mean the October 2015 DEM is less influenced by the avalanche. I suggest the author to exclude all the DEMs after the earthquake to compute the 2006-2015 ensemble.**

The May 2015 and the October DEMs are both influenced by the earthquake, but not to the same degree. Volume loss from glacier area where avalanche material accumulated between May and October 2015 was 30 times higher than during an average year (May 2015 – Oct 2015: $-1.5*10^7$ m$^3$, Oct 2006 – Apr 2014:$-5*10^5$ m$^3$a$^{-1}$). After this rapid initial downwasting the avalanche material diminished to a volume that does not influence the DEM to a degree which would justify its rejection from the 2006-2015 ensemble. This is why we state that that the avalanche impacts six months after the earthquake are not significant in comparison to the 2006-2015 ensemble uncertainty and this is why we think the October 2015 DEM should be used for the 2006-2015 ensemble.

To explain this better we have thoroughly revised the second part of section 4.6 (P. 20, lines 12 – 25) and we now provide the details given above:

*"Volume loss from glacier area where avalanche material accumulated between May and October 2015 was 30 times higher than during an average ablation season (May 2015 – Oct 2015: -1.5\*10$^7$ m$^3$, Oct 2006 – Apr 2014: -5\*10$^5$ m$^3$ a$^{-1}$). After this rapid initial downwasting the avalanche deposits diminished to a volume of 10$^7$ m$^3$, equivalent to an average positive surface elevation change over all debris-covered glacier area of 0.52 ± 0.19 m (or 0.06 - 0.09 m a$^{-1}$ if divided over six to nine years). The avalanche impact on the Oct 2015 DEM is thus within the uncertainty range associated to multi-annual Δh/Δt values (±0.12 m a$^{-1}$, Table 3) and justifies why the October 2015 DEM is considered for the 2006-2015 ensemble.*

*The avalanche traces are still visible six months after the earthquake at Lirung Glacier (4350-4400 m a.s.l.), at Langtang Glacier (4500-4900 m a.s.l.), at Langshisha Glacier (4800 m a.s.l.) and at Shalbachum Glacier (4750 m a.s.l.) (Figure 9). Except for Lirung Glacier at 4350 m a.s.l. the 2006-Oct 2015 and 2009-Oct 2015 thinning profiles are within the error bounds associated to other multi-annual periods."*

**page 15, line 30 – page 16 line 6: The error bounds….**

**>> This is an example that too much detailed description on the uncertainty gets into the understanding of more important points. Please consider to move this kind of details to supplementary information if necessary.**

We have simplified and shortened the text as follows: "*However, error bounds are not overlapping at 80% confidence levels (assuming normal distribution). Given the probability of less than 10% for 1974-2006 and 2006-2015 thinning rates for being above or below this confidence interval, the estimated confidence level of accelerated thinning rates is higher than 99%.*" (P. 14, lines 20-23)

**page 16, line 8: At Shalbachum Glacier the error bounds are overlapping but the estimated probability that 1974-2006 thinning rates are higher than 2006-2015 volume loss rates is less than 10%.**

**>> "At Shalbachum Glacier the error bounds are overlapping but the estimated probability of thinning acceleration is more than 90%."**

Thank you. We have revised the sentence as suggested.

**page 16, line 13: The estimated probability that at one of these glaciers mean thinning rates changed by less than ± 0.15 m a-1 between the two periods is higher than 90%.**

**>> "Mean thinning rates of these glaciers changed by less than \pm 0.15 m a-1 between the two periods (90% confidence)."**

We have revised and simplified this sentence: "*At Lirung and Kimoshung Glaciers the mean thinning rates have likely remained approximately constant (Table 4). Mean thinning rates of these glaciers increased by less than 0.10 m a$^{-1}$ between 1974-2006 and 2006-2015.*" (P. 14, lines 27-29)

**page 16, line 17-20: "The ensemble uncertainty is ± 0.43 m a-1, which …"**

**>> Please consider to omit these sentences because they are not essential to draw your main conclusions.**

We have omitted this sentence as suggested.

**page 16, page 29: An increase in identified mean volume loss rates…**

**>> An increase in volume loss rates …**

Revised as suggested.

**page 16, line 31 - page 17, line 2: For Ghanna tongue the identified changes in thinning rates are not significant given the uncertainties, …**

**>> "Changes in thinning rates are not significant for Ghanna tongue, but five out of six members of the 2006-2015 decreased as compared to the previous period. "**

Revised as suggested.

**page 17, line 11-13: Here, the 2006-2015 ensemble mean value (-0.50 ± 0.20 m a-1) indicates more than three times lower thinning rates than at Lirung tongue.**

**>> " Here, the 2006-2015 ensemble mean value (-0.50 ± 0.20 m a-1) is 30% of the thinning rate at Lirung tongue."**

Revised as suggested.

**page 17, line 25-28: Our analysis thus shows that elevation change estimates are in most cases not significantly different if we assume different thresholds for outlier definition or if we consider the uncertainty in our ELA estimate.**

**>> "Our analysis thus shows that elevation change estimates are in most cases not significantly influenced by outlier definition and ELA estimate."**

Revised as suggested.

**page 17, line 28-30: Significant sensitivity values …**

**>> Please rewrite. e.g. "Erroneous patterns in the accumulation areas (> 1 sigma) cause significant influence on the results."**

We have omitted this sentence as it is a repetition of what is stated above in the same section.

**page 18, line 10: This pattern of decreasing thinning rates contrasts…**

**>> "This pattern constrasts…"**

Revised as suggested.

**page 18, line 12-13: "… the comparability of 1974-2006 12 thinning rates with the 2006-2015 ensemble is limited."**

**>> What do you mean? "thinning patterns in 1974-2006 and 2006-2015 are different"?**

We have clarified the sentence as suggested: "*On Langshisha Glacier thinning patterns in 1974-2006 and 2006-2015 are different near the terminus (Figure 9b).*"

**page 18, line 23: To compare the thinning patterns of debris-covered glaciers to the thinning patterns of debris-free glaciers, …**

**>> "To compare the thinning patterns of debris-covered and debris-free glaciers, …"**

This sentence has been omitted. We have shortened the text and merged the paragraph with the following paragraph to make the structure more logical (based on the editor's comment on P18 L23-L29).

**page 18, line 25-26: Yala Glacier experiences more rapid thinning over almost its entire elevation range in recent periods.**

**>> " Yala Glacier experiences more rapid thinning in recent periods over almost its entire elevation range."**

This sentence has been omitted. We have shortened the text and merged the paragraph with the following paragraph to make the structure more logical (based on the editors' comment on P18 L23-L29).

**page 19, line 5-11:**

**>> Please consider to rewrite this paragraph.**

Some of the details provided here could be omitted (see our answer to the comment below) and we have thus revised this paragraph as follows: "*On the large debris-covered glaciers, areas of maximum thinning seem to have shifted and extended to higher elevations only at Langtang Glacier, where during the period 1974-2006 maximum thinning occurred between 4850 and 4950 m a.s.l. (Figure 9a). On Shalbachum Glacier maximum thinning during the period 1974-2006 occurred slightly higher up at 4750 – 4800 m a.s.l. (Figure 9c).*"

**page 19, line 8-9: "the difference between thinning near the terminus and maximum thinning"**

**>> It is not clear what is meant by "difference".**

What we meant here was that the difference between thinning rates near the terminus and the upper part of the tongues (where maximum thinning occurs) increased. We have omitted this part of the sentence, since heterogeneous thinning accelerations on debris-covered glacier area are discussed in section 5.1.

**page 19, line 10: "but"**

**>> I do not understand why these two clauses are connected by "but".**

The reviewer is right, the connection of the clauses by "but" was not justified here. We have revised the paragraph as stated above.

**page 20, line 18 – page 21, line 27:**

**>> Please consider to move these paragraphs to Discussion section. You describe more than "Results".**

We prefer to keep these paragraphs in the Results section, as the text here is also not a "discussion" of results, but an in-depth description of observations.

**page 21, line 19: Lirung tongue also shows an opposite behavior, except for the lowest elevation band.**

**>> It is not clear what is meant by "opposite behavior".**

What we want to say here is that the observation of high cliff area fractions where thinning rates did not change significantly does not apply for Lirung tongue. We have replaced "opposite" by "different" (P. 19, line 23).

**page 21, line 25: "not stagnating"**

**>> Please consider to reword it.**

We have reworded the sentence as followed: "*Across all debris-covered glacier tongues, 77% of all elevation bands where thinning accelerated ($\Delta(\Delta h/\Delta t) < -0.2$ m a$^{-1}$) are not stagnating (velocities above 2.5 m a$^{-1}$), and in 72% of all elevation bands where thinning rates remained constant or declined ($\Delta(\Delta h/\Delta t) \geq -0.2$ m a$^{-1}$) we observe velocities below 2.5 m a$^{-1}$.*"

**page 21, line 27: …we observe stagnant conditions with velocities below 2.5 m a-1.**

**>> "… we observe velocity below 2.5 m a-1."**

We have reworded the sentence as explained above.

**page 22, line 16-19: "Accelerated thinning …"**

**>> It is difficult to understand this sentence.**

We have shortened this sentence and moved it to the end of the paragraph ("*Accordingly, accelerated thinning of debris-covered area in the Upper Langtang catchment does not take place on stagnating parts of the tongues, but where the transition between the active and the stagnant ice can be expected (Figure 11).*"). We hope that the sentence is now clear.

**page 23, line 18-20: "However, given the usually very slow dynamical response of debris-covered glaciers to changes in the local temperature (Banerjee and Shankar, 2013),"**

**>> I do not understand why this can be a reason why "slowdown of the compressive flow regime is not the primary factor".**

We have removed the sentence, since we agree that also a slow dynamical response could potentially affect glacier thinning significantly over the time scales discussed in our study.

**page 23, line 30: loose**

**>> lose?**

Yes. Thank you for noticing. We have corrected this.

**page 24, line 6-7: "It can thus be assumed that they become less abundant with decreasing flow."**

**>> Are you sure that this is true on debris covered glaciers in general?**

Our results can probably not be generalized to all debris-covered glaciers. We have thus replaced "*it can be assumed that*" by "*our results suggest that*" (P. 22, line 30).

**page 24, line 21: 5.1.1 Post-earthquake avalanche impacts**

**>> I suggest the author to move this subsection to the last in Discussion.**

This is now the second last subsection of the Discussion section (now section 5.4). We think that subsection "Comparison to other studies" should come last.

**page 26, line 32: 2011), " our results indeed point to a difference in current volume loss of debris-free and debris-covered glaciers."**

**>> It is not clear what you mean.**

We have removed this sentence and revised the paragraph on the basis of the editors comment on P26 L21-22 and L32ff.

**Table 4 caption: " … due to avalanches triggered by the Nepal earthquake on 25 April 2015."**

**>> It is not accurate to attribute all the changes to the earthquake because they include the effects of mass balance and ice flow for a certain amount. What about writing as "… after the earhquake on 25 April 2015"?**

The sentence suggested by the reviewer would not be correct. The table (now Table 8) indeed only reports the elevation changes due to avalanches, since the values represent only elevation changes of avalanche affected area ($\Delta h > 5$ m in May 2015) divided by the total debris-cover area, and not the changes of all debris-covered glacier tongues after the Nepal earthquake on 25 April 2015. We hope this is now clear (we have added "*$\Delta h > 5$ m in May 2015*" in brackets to the caption text).

**Figure 2:**

**>> I suggest the author to move this plot to supplementary information. This is too much detail to show in the main text.**

We think this figure is useful to summarize the performance of the DEM extraction process and the determined uncertainties. We agree these are relatively detailed results. However, the detailed description of differences in uncertainty between glaciers is a novel aspect of our study and we would like to present this figure therefore in the main text. As the reviewer states in his general comment, the careful descriptions are useful to those who use the same techniques for assessing glacier surface elevation changes.

**Figure 3:**

**>> Please explain in the caption what do the color bars represent.**

We have added a sentence as suggested ("*The color bars represent hypsometries of glacier area, off-glacier area and debris-covered glacier areas, respectively.*").

**Figure 13: "Off-glacier velocities are shown in transparent color."**

**>> I do not think this information is necessary on the plot.**

We do not agree and think the information is useful, as it allows a reader to evaluate the quality of the data.

[revised manuscript text omitted]

---

## Author Response (AR3)

**"Heterogeneous glacier thinning patterns over the last 40 years in Langtang Himal, Nepal"**

**by S. Ragettli, T. Bolch and F. Pellicciotti**

Dear Editor,

Thank you for the careful reading of our manuscript and for the comments. Our point-by-point responses are provided below.

With kind regards,

Silvan Ragettli,

**Editor's technical comments:**

**Title. Add that this is in Nepal?**

It is now specified in the title that Langtang Himal is in Nepal.

**Abstract L20. Is "identify" the best verb? "measure"? "compute"?**

We have replaced "identify" by "compute" as suggested.

**Effect of the earthquake. It has also been studied using SPOT6/7 data by Lacroix, 2016. Probably worth mentioning this study together with Kargel et al., 2015.**
**Lacroix, P.: Landslides triggered by the Gorkha earthquake in the Langtang valley, volumes and initiation processes, Earth, Planets and Space, 68(1), 1–10, doi:10.1186/s40623-016-0423-3, 2016.**

This is a good suggestion. We now also cite Lacroix 2016.

**P8 L11. Space missing**

Ok. Thank you for noticing.

**P11 L27. I think "until" can be deleted**

We agree and have removed "until".

**Everywhere: when you have two or more "glaciers", I do not think a cap letter is needed for "glaciers"**

Both capital letters or small letters after naming two or more "glaciers" are used in the literature. We prefer to consistently use capital letters.

**P17 L25. suggestion: "During the same two time intervals debris-free Kimoshung Glacier" ("debris-free" added for the sake of parallelism with Yala Glacier and to remind this fact to the reader)**

Ok. We have added "debris-free" here.

**P18 L25, L29. Not so easy for the reader to understand what you mean by "here" (the present study? the glacier described?). Maybe you could clarify a bit.**

We have clarified this and removed "here" in both cases.

**P20. L17. Maybe you could add that this is because you are only considering time period of at least 4 years in the ensemble? Then the impact is at most 0.52/4=0.16 just above 0.12 m/yr? Otherwise it is not really clear how 0.52 m is compared to 0.12 m/yr.**

This is now clarified. We now state "(… if divided over *six years, which is the shortest time interval of any Δh/Δt map in the ensemble involving the October 2015 DEM*)."

**P21. L10. (and elsewhere) "volume loss rate". You are often using both "thinning rate" and "volume change rate" as if they were the same. I think in the context of your paper/results, it would be easier for the reader if you could stick to a single terminology (thinning rate). You could legitimately answer me that this is just a matter of dividing by the area and you would be mathematically right. But unifying your terminology (i.e. text = figure/table) will, I think, help the reader to really extract the real message.**

Ok. We have replaced "volume loss rate" consistently by "thinning rate" everywhere in the text.

**P26 L26. I think it would be more accurate to say here "for the same 10 glaciers as Bolch et al. (2011) in the Everest region" (because the Gardelle et al., 2013 mass balance for the whole Khumbu area is less negative than for these 10 glaciers)**

We have clarified the sentence as suggested.

**P27 L3. I do not think you need to define DEM again in the conclusion.**

We agree. We have changed this.

**Figure 2 legend. "The central marks correspond to the median" is repeated twice**

We have rephrased the legend to prevent repetitions.

**Figure 7. The dark blue line for the 1974-2006 dh/dt rate is not continuous. Probably because the time scale is not linear. To be explained in the legend.**

We have added a sentence to clarify this (also in the legend of Figure 8): "*Note that the 1974-2006 time scale is not linear (dashed dark blue line).*"

**P45 L4. " uncertainty in function" --> " uncertainty as a function" (I think). Same for figure 10.**

We have corrected this as suggested.

**P47. L6. data from Figure 11. Do you mean Figure 13?**

We have corrected this. Thank you for noticing.

[revised manuscript text omitted]